# New insight into the significance of KLF4 PARylation in genome stability, carcinogenesis, and therapy

Zhuan Zhou[1,†] ![ID], Furong Huang[2,†], Indira Shrivastava[3,†], Rui Zhu[2], Aiping Luo[2], Michael Hottiger[4] ![ID], Ivet Bahar[3], Zhihua Liu[2], Massimo Cristofanilli[5] & Yong Wan[1,*]

## Abstract

KLF4 plays a critical role in determining cell fate responding to various stresses or oncogenic signaling. Here, we demonstrated that KLF4 is tightly regulated by poly(ADP-ribosyl)ation (PARylation). We revealed the subcellular compartment for KLF4 is orchestrated by PARP1-mediated PARylation. We identified that PARylation of KLF4 is critical to govern KLF4 transcriptional activity through recruiting KLF4 from soluble nucleus to the chromatin. We mapped molecular motifs on KLF4 and PARP1 that facilitate their interaction and unveiled the pivotal role of the PBZ domain YYR motif (Y430, Y451 and R452) on KLF4 in enabling PARP1-mediated PARylation of KLF4. Disruption of KLF4 PARylation results in failure in DNA damage response. Depletion of KLF4 by RNA interference or interference with PARP1 function by KLF4[YYR/AAA] (a PARylation-deficient mutant) significantly sensitizes breast cancer cells to PARP inhibitors. We further demonstrated the role of KLF4 in modulating homologous recombination through regulating BRCA1 transcription. Our work points to the synergism between KLF4 and PARP1 in tumorigenesis and cancer therapy, which provides a potential new therapeutic strategy for killing BRCA1-proficient triple-negative breast cancer cells.

**Keywords** DNA damage response; DNA repair; KLF4; PARP1; tumorigenesis and therapeutics

**Subject Categories** Cancer; Molecular Biology of Disease

## Introduction

Krüppel-like factor 4 (KLF4, GKLF) plays a pivotal role in orchestrating a variety of cellular processes, including cell cycle control, genome stability, signal transduction, stem cell expansion, and immune response (Rowland & Peeper, 2006; Tetreault *et al*, 2013; Ghaleb & Yang, 2017). Results from recent pathophysiological and "The Cancer Genome Atlas (TCGA)" studies have unveiled an oncogenic property for KLF4 in breast carcinogenesis (Fletcher *et al*, 2011; Li *et al*, 2013; Hu *et al*, 2015), although its underlying mechanism in breast tumor initiation and invasion remains unclear.

KLF4 acts as a transcriptional factor and regulates various biological functions by either activating or inhibiting a network of genes involved in developmental and etiological events (Gamper *et al*, 2012; Tetreault *et al*, 2013). To our surprise, recent physiological studies provide an ambivalent view of KLF4 with regard to oncogenesis as either a tissue-specific tumor suppressor or an oncogenic factor with an unknown mechanism (Rowland & Peeper, 2006; Tetreault *et al*, 2013). Depending on tumor type, KLF4 is thought to be a tumor suppressor in gastrointestinal, esophageal, lung, and pancreatic cancers (Evans & Liu, 2008; Hung *et al*, 2013; Wei *et al*, 2016), while it acts as an oncogenic player in breast and squamous cell carcinoma (Foster *et al*, 1999; Foster *et al*, 2000; Pandya *et al*, 2004; Foster *et al*, 2005; Rowland *et al*, 2005; Dong *et al*, 2014). Despite the knowledge about its role in gastrointestinal and pancreatic cancers, the process through which abnormal accumulation of KLF4 that promotes malignant transformation in mammary glands and skin remains unclear (Evans & Liu, 2008; Tetreault *et al*, 2013).

Our recent work has demonstrated that in response to oncogenic signaling as well as genotoxic stress, KLF4 undergoes posttranslationally modified such as in ubiquitination and methylation (Evans *et al*, 2007; Meng *et al*, 2009; Hu & Wan, 2011; Hu *et al*, 2015; Tian

1  Department of Obstetrics and Gynecology, Department of Pharmacology, The Robert H. Lurie Comprehensive Cancer Center, Northwestern University Feinberg School of Medicine, Chicago, IL, USA
2  State Key Laboratory of Molecular Oncology, National Cancer Center/National Clinical Research Center for Cancer/Cancer Hospital, Chinese Academy of Medical Sciences and Peking Union Medical College, Beijing, China
3  Department of Computational and Systems Biology, University of Pittsburgh School of Medicine, Pittsburgh, PA, USA
4  Department of Molecular Mechanisms of Disease, University of Zurich, Zurich, Switzerland
5  Lynn Sage Breast Cancer Program, Department of Medicine-Hematology and Oncology, Robert H. Lurie Comprehensive Cancer Center, Northwestern University Feinberg School of Medicine, Chicago, IL, USA
   *Corresponding author. Tel: +312-503-2769; Fax: +312-503-0095; E-mail: yong.wan@northwestern.edu
   †These authors contributed equally to this work

et al, 2015; Zhou et al, 2017a). We have discovered that the interplay between ubiquitin ligase VHL/VBC and arginine methyltransferase PRMT5 governs KLF4 protein stability (Hu et al, 2015). To further explore whether posttranslational modifications orchestrate KLF4 protein trafficking, in particular within the cytosol, from the cytosol to the nucleus, and from the nucleus to the chromatin, we have searched for new posttranslational modifiers that are involved in the above critical cellular processes. Here, we report the novel finding that (ADP-ribosyl)ation (PARylation) of KLF4 by Poly [ADP-ribose] polymerase 1 (PARP1) regulates KLF4 chromatin recruitment following DNA damage response, which provides potential novel strategies for targeted cancer therapy.

PARP1 is the enzyme responsible for PARylation, which catalyzes the covalent transfer of mono- or oligomeric ADP-ribose groups from NAD$^+$ to target proteins (Kim et al, 2005). Among the seven members of the PARP family, PARP1 plays a key role in multiple DNA damage response pathways and governs genome stability. Upon DNA damage, PARP1 is rapidly recruited to the altered DNA damage lesion sites, where its catalytic activity increases by a hundred-fold, resulting in the conjugation of long branched PAR chains (Jackson & Bartek, 2009; De Vos et al, 2012; Wang et al, 2012; Yazinski et al, 2017). Recent works have shown a series of critical DNA damage-related and DNA-repair proteins that are modified by PARylation, including histones, topoisomerase, DNA protein kinase (DNA-PK), XRCC1, mitotic recombination 11 (MRE11), and ataxia telangiectasia-mutated (ATM) (Wang et al, 2012; Wei & Yu, 2016). Similar to protein phosphorylation and ubiquitylation, PARylation is a reversible process, wherein the conjugation of the PAR polymer can be countered by two enzymes, including poly(ADP-ribose) glycohydrolase (PARG) and ADP-ribose hydrolase ARH3 through hydrolyzing the PAR chain (Leung, 2014). For mouse genetic studies on the impact of PARP1 in both tumor suppression and oncogenesis, different methodologies and physiological context have shown differing results (Yelamos et al, 2011; Schiewer & Knudsen, 2014). While ablation of PARP1 linking to mammary tumor formation (de Murcia et al, 1997; Tong et al, 2007) suggests its tumor suppressing effect, the oncogenic role of PARP1 was implied due to the correlation between its abnormal accumulation and poor prognoses in various types of cancers, including breast, uterine, lung, ovarian, colorectal, and skin cancers (Ossovskaya et al, 2010).

Although considerable attention has been paid to elucidate the biochemical mechanism by which functional proteins are modulated by PARylation, how exactly the impaired PARP1 drives oncogenesis remains unclear. Specifically, while the overlapping physiological impact for both PARP1 and KLF4 is observed in many aspects, such as developmental control, stem cell self-renewal, DNA damage response/DNA repair, and tumorigenesis, the functional mechanism between PARP1 and KLF4 and its physiological consequences have not yet been adequately explored. The present identification of DNA damage-induced PARylation of KLF4 fills a critical gap in the knowledge of the impact of PARP1 and KLF4 in oncogenesis and on therapeutics.

Development of PARP inhibitors, including olaparib, rucaparib, niraparib, and talazoparib, has provided a method to treat triple-negative breast cancer (TNBC) patients with BRCAness, who mimic BRCA1 or BRCA2 loss and are deficient with regard to homologous recombination (HR), based on their synthetic lethal effect (McCann & Hurvitz, 2018; Papadimitriou et al, 2018). However, the application of the PARP inhibitors is limited to a small fraction of TNBC patients who bear the BRCA1/2 mutation (Papadimitriou et al, 2018; Sulai & Tan, 2018). Unfortunately, approximately 80% of TNBC patients who have normal BRCA function are not responsive to PARP inhibitors due to their normal HR function (Hartman et al, 2012; Greenup et al, 2013; Papadimitriou et al, 2018). Thus, a new strategy that blocks BRCA-mediated HR could potentially expand the therapeutic value of PARP inhibitors to benefit BRCA-proficient TNBC patients (Johnson et al, 2016). Here, we identified PARP1 as a previously undocumented posttranslational modifier of KLF4 in genome stability through regulating KLF4 transcriptional function by dictating its recruitment to the chromatin, ensuring DNA damage response. We further unveiled a newly discovered function of KLF4 in modulating HR through regulating BRCA1 in a PARylation-independent manner.

Finding a functional interaction between KLF4 and PARP1 in DNA damage response and DNA repair unveils a novel approach to induce synthetic lethality that further leverages PARP inhibitors specifically to benefit BRCA-proficient TNBC patients. Indeed, inactivation of KLF4 or depletion of KLF4 in preclinical models significantly sensitizes BRCA-proficient TNBC breast cancer tumor to PARP inhibitors. Therefore, our findings demonstrate a potential novel therapeutic strategy to target BRCA-proficient TNBC breast cancer by exploiting the synergism of KLF4 and PARP1.

## Results

### Establishment of KLF4$^{loxp/loxp}$ coupled AAV7-Cre inducible mouse to determine the impact of KLF4 in governing genome stability and tumorigenesis

Previous results using a cultured-cell model obtained by us and others demonstrated the critical role of KLF4 in governing DNA damage response and DNA repair; otherwise, deregulation could lead to genome instability and tumorigenesis (Yoon et al, 2003; Yoon & Yang, 2004; Yoon et al, 2005; Ghaleb et al, 2007). It has been a technical challenge in the field to determine the relevance of KLF4-mediated genome stability in cancer formation due to KLF4's ambivalent role in tumorigenesis as either a tissue-specific tumor suppressor or an oncogene. In order to dissect the impact of KLF4 in genome stability, carcinogenesis, and therapeutics, we have established KLF4$^{loxp/loxp}$ mice followed by engineering conditional knockout of KLF4 in intestinal tissue by utilizing an adeno-associated virus-Cre inducible system (Polyak et al, 2008; Zincarelli et al, 2008) (Fig 1A). We then exposed mice with conditional knockout of KLF4 to γ-radiation (Fig 1A) and subsequently detected the relevance of KLF4 ablation to genome stability by measuring morphological damage of the intestine, mouse tolerance to genotoxic stress, and DNA damage status in the intestine (Talmasov et al, 2015).

We have tested various AAV serotypes with green fluorescent protein (GFP) reporter in vivo by intraperitoneal injection and found that AAV7 exhibits the strongest intestinal tissue tropism among other AAV serotypes (AAV1, AAV2, AAV5, AAV6, AAV8, and AAV9) (Appendix Fig S1A and B). The intraperitoneal injection of AAV7 could deliver GFP reporter to the gastroenterology system, including the stomach, duodenum, and jejunum/ileum (Appendix Fig S1A and C).

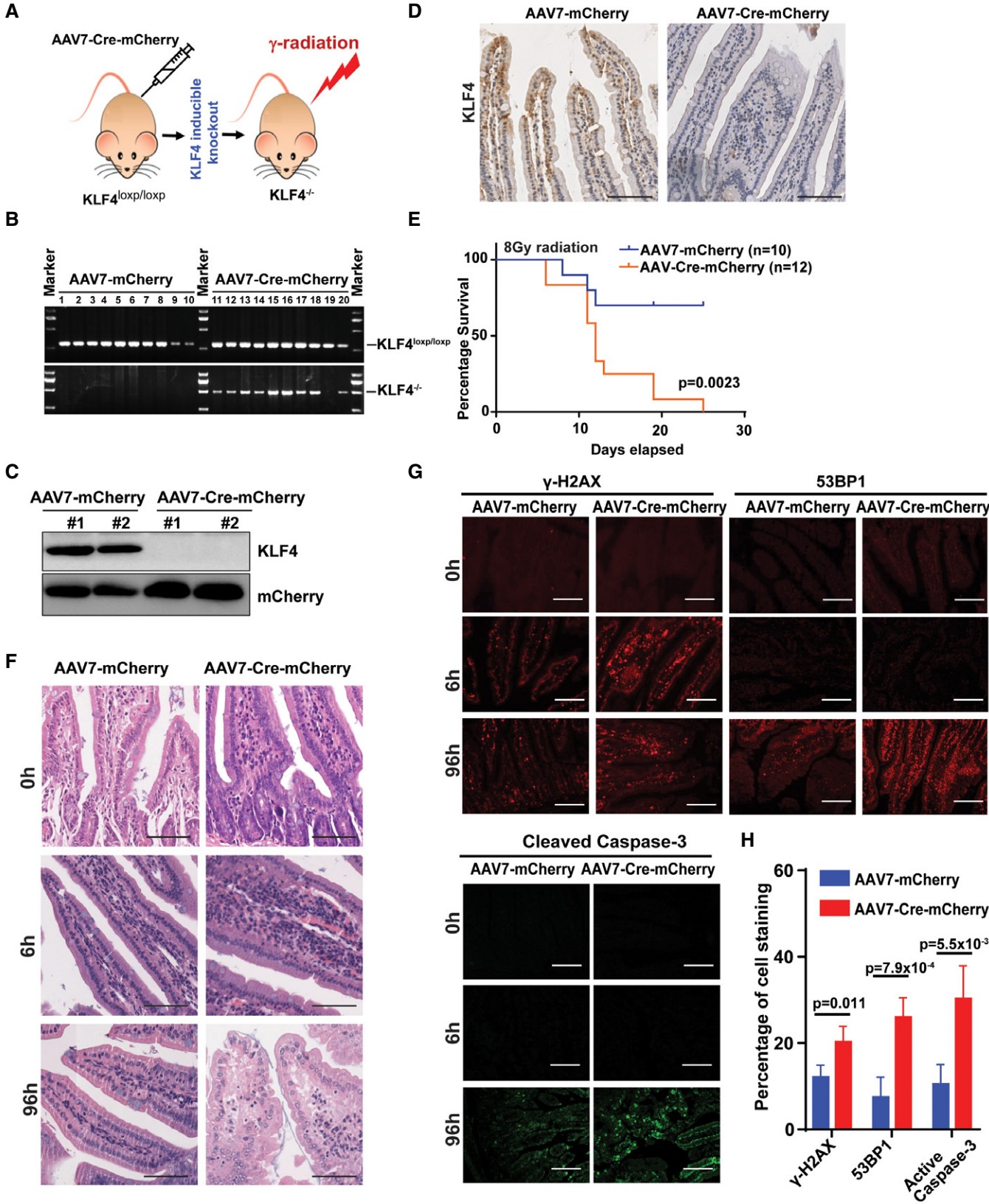

**Figure 1.**

◄ **Figure 1.** **Establishment of KLF4$^{loxp/loxp}$ coupled AAV-Cre inducible mouse to determine the impact of KLF4 on governing genome stability and tumorigenesis.**

A Schematic diagram of establishment of KLF4$^{loxp/loxp}$ coupled AAV7-Cre inducible mouse.

B Genotyping of KLF4$^{loxp/loxp}$ coupled AAV-Cre inducible mouse. $2 \times 10^{11}$ particles of AAV7-Cre-mCherry or AAV7-mCherry were intraperitoneally injected into KLF4$^{loxp/loxp}$ mice (6–8 weeks). Five weeks later, mice tails were cut and collected for DNA extraction and PCR analysis. PCR results were analyzed by 1% agarose gel.

C, D Validation of inducible knockout of KLF4 in mouse intestine. Five weeks after injection of AAV7-Cre-mCherry, mouse intestine was removed and followed by the preparation of tissue section. The KLF4 expression in the intestine was then detected by Western blot (C) and immunohistochemistry (D). Scale bars, 100 μm.

E Kaplan–Meier survival curves of KLF4$^{loxp/loxp}$ mice with intraperitoneal injection of AAV7-Cre-mCherry or AAV7-mCherry followed by the treatment with 8-Gy (total-body) γ-irradiation 5 weeks after then. AAV7-mCherry, $n = 10$ per group; AAV7-Cre-mCherry, $n = 12$ per group. $P = 0.0023$, log-rank test.

F Histological analysis of intestinal epithelium of KLF4$^{loxp/loxp}$ mice with AAV7-mCherry or AAV7-Cre-mCherry intraperitoneal injection followed by 8-Gy (total-body) γ-irradiation 5 weeks after then. Tissues were collected from the sham mice and mice at different time after exposure to γ-irradiation. Scale bars, 100 μm.

G, H Immunofluorescent staining of γ-H2AX, 53BP1, and cleaved caspase-3 in the intestinal epithelium of KLF4 $^{loxp/loxp}$ mice with injection of AAV7-mCherry or AAV7-Cre-mCherry followed by treatment of γ-irradiation. Tissues were collected from sham mice and mice at different time after exposure to γ-irradiation and then staining with indicated antibodies. (G) Quantification of γ-H2AX, 53BP1 and active caspase-3-positive cells based on the Immunofluorescent staining results presented in (H). Data are mean ± SEM; $n = 4$ per group; $P = 0.0023$ (r-H2AX), $P = 7.9 \times 10^{-4}$ (53BP1) $P = 5.5 \times 10^{-3}$ (active caspase-3). One-way ANOVA was used for the statistical analysis. Scale bars, 60 μm.

Therefore, we applied the AAV7-Cre-mCherry to deliver Cre expressed in the gastroenterology system to establish adult inducible KLF4 knockout in KLF4$^{loxp/loxp}$ mice. As shown in Fig 1B–D, intraperitoneal injection of AAV7-Cre-mCherry into KLF4$^{loxp/loxp}$ mice induces significant local KLF4 knockout in intestinal tissue. We have observed that, after mice were subjected to 8 Gy total-body γ-radiation, ablation of KLF4 causes shorter survival times as compared to KLF4 $^{loxp/loxp}$ mice (Fig 1E). Results from histological analysis of intestinal epithelium of KLF4 $^{loxp/loxp}$ mice with either AAV7-mCherry or AAV7-Cre-mCherry delivery showed profound intestinal tissue damage at 96 h postirradiation after KLF4 deletion (Fig 1F), while no significant distinction is observed pre-irradiation or 6 h postirradiation. Furthermore, at 96 h postirradiation, the intestinal epithelium in KLF4 $^{loxp/loxp}$/ AAV7-Cre-mCherry mice developed deep-set crypts and damaged intestinal mucosal structures with focal villus edema, indicating severe damage of the intestinal epithelium (Fig 1F) (Potten *et al*, 1990; Talmasov *et al*, 2015). Moreover, immunofluorescent staining of the intestinal epithelium with critical DNA damage/DNA repair and apoptosis markers, including p53, p-CHK2/Thr68, γ-H2AX, 53BP1, and cleaved caspase-3, indicated increased DNA damage (γ-H2AX, 53BP1) and apoptosis (cleaved caspase-3) in the duodenum of KLF4 $^{loxp/loxp}$ mice with AAV7-Cre-mCherry in comparison with AAV7-mCherry injection (Fig 1G and H, and Appendix Fig S1D and E). Taken together, our results of intestine-specific inducible ablation of KLF4 showed the important role of KLF4 in genome stability through modulating DNA damage response and DNA repair.

## KLF4 orchestrates DNA damage response and DNA repair

To further study the role of KLF4 in DNA damage response and DNA repair, we have measured the effect of KLF4 knockout on chromosomal instability, DNA-double strain break, aneuploidy, alteration of mitotic index, and HR and non-homologous end joining using KLF4 knockout mouse embryonic fibroblasts (MEFs) as well as U2OS cells (Hagos *et al*, 2009; El-Karim *et al*, 2013). As shown in Appendix Fig S2A–E, genetic ablation of KLF4 leads to significant chromosomal breaking, as measured by metaphase karyotype analysis (Elenbaas *et al*, 2001). KLF4 deletion causes obvious accumulation of 53BP1 and γ-H2AX foci compared with wild-type MEF cells, which indicated a failure in DNA damage repair/response (Appendix Fig S2F–H; Harper & Elledge, 2007). Moreover, we observed that loss of KLF4 in MEFs resulted in increased aneuploidy

cells (> 4n) and pH3-positive cells (Appendix Fig S2I and J; Hu *et al*, 2015). To assess the role of KLF4 in HR and non-homologous end joining, we engineered stable KLF4 knockdown based on U2OS cells, followed by performing HR and non-homologous end joining (NHEJ) assays in U2OS-GFP-EJ5 and U2OS-DR-GFP cells (Wei *et al*, 2015). As shown in Appendix Fig S2K and L, deletion of KLF4 led to significant defects in DNA repair of HR, while no significant difference was observed in NHEJ between KLF4 knockdown and wild-type cells. Taken together, our data suggest that KLF4 is a critical player, and its dysfunction affects both DNA damage response and DNA repair.

## Identification of KLF4 PARylation by PARP1 in KLF4-mediated DNA damage response

The above genetic and physiological analyses suggest a pivotal role for KLF4 in governing genome stability by regulating cellular response to DNA damage and damage lesion repair. While the fact that KLF4 governs transactivation of its downstream targeting genes has been partially explained, not much is known about how KLF4 is regulated in response to genotoxic stress. We recently reported that KLF4 is a fast turnover protein with its protein stability governed by an interplay between VHL-mediated ubiquitylation and PRMT5-mediated methylation (Hu *et al*, 2015).

To search for other proteins that might regulate KLF4 function in genome stability and carcinogenesis, we took a biochemical approach to identify new KLF4-interacting proteins, especially for interactions enhanced in response to DNA damage signal, using cells expressing tagged KLF4 to isolate KLF4 protein complexes by tandem immune-purification (Gamper *et al*, 2012; Hu *et al*, 2015). Our efforts led to the identification of interaction between KLF4 and PARP1 using mass spectrometric analyses (Fig 2A–C and Appendix Fig S3A and Table S1). Our purification indicates, while basal interaction between KLF4 and PARP1 is detected, the capacity of PARP1 to bind to KLF4 increases several-fold after cellular exposure to γ-radiation, suggesting the potentially critical role of PARP1 in regulating KLF4 in genome stability.

The interaction of KLF4 and PARP1 was further confirmed by immunoprecipitation of endogenous KLF4 complex followed by immune-blotting of PARP1 (Fig 2D) or by determining complexes of overexpressed tagged KLF4 precipitated with antibody against FLAG tag followed by probing for PARP1 (Appendix Fig S3A). Co-immunoprecipitation (co-IP) of PARP1 and KLF4 was detected in

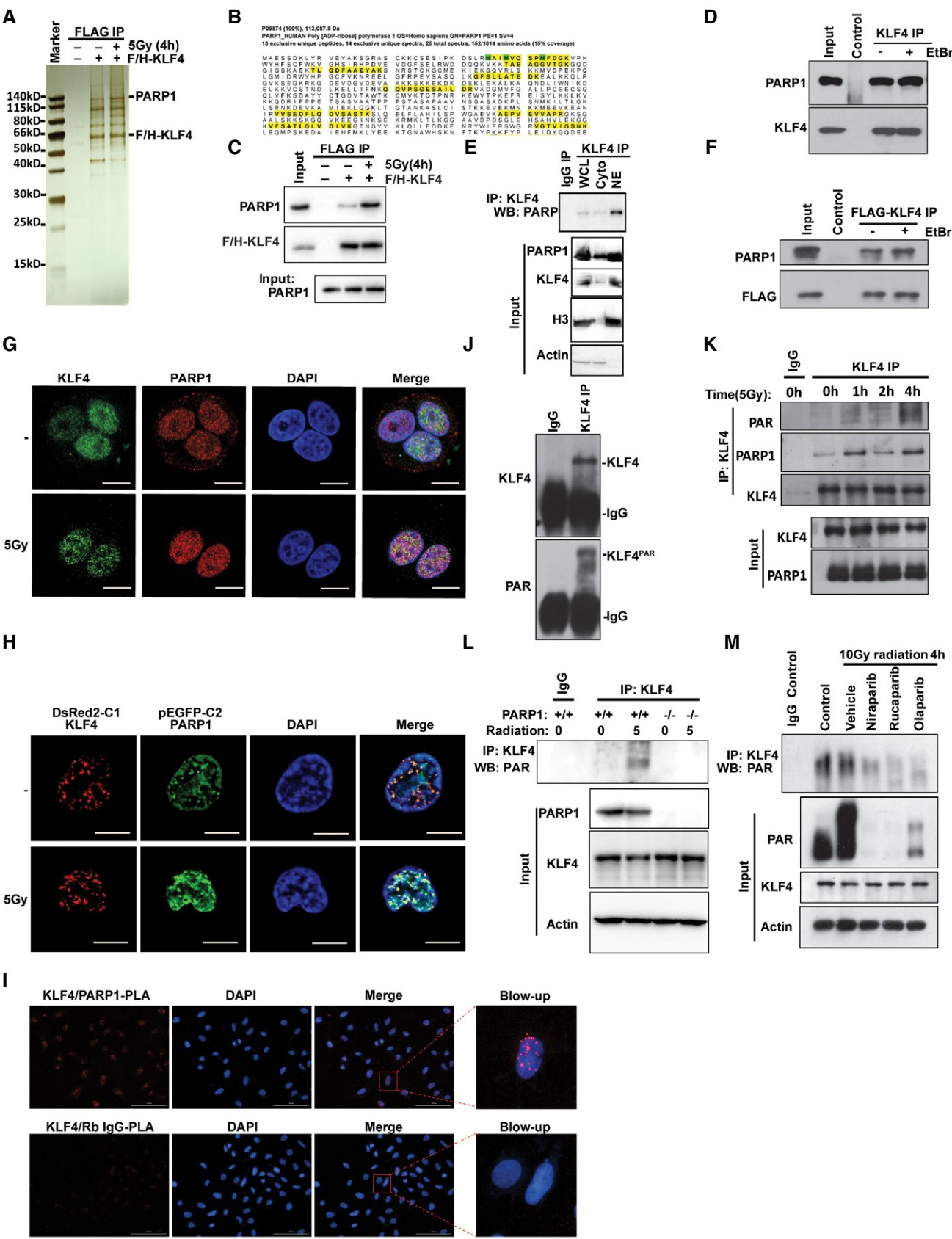

**Figure 2.**

**Figure 2. Identification of KLF4 PARylation by PARP1 in KLF4-mediated DNA damage response.**

A   Purification of KLF4 protein complex in the presence and absence of DNA damage based on TAP-KLF4 stable expression cells (U2OS). Proteins that interacted with KLF4 were purified from U2OS cells expressing FLAG and HA-tagged KLF4 in the absence and presence of 5 Gy radiation at 4h after the treatment. The accumulated bind induced in response to radiation was isolated for mass spectrometry analysis. PARP1 was identified as a binding partner for KLF4.

B   The sequences of mass spectrometry analysis for identification of PARP1 (P09874) to be an interacting partner of KLF4. The identified peptides were labeled in yellow.

C   The purified complex was further confirmed by Western blot detected by FLAG-KLF4, PARP1(lane 1). The interaction between KLF4 and PARP1 was significantly increased in response to γ-radiation detected by pulldown experiment (lanes 3 and 4).

D   Validation of interaction between ectopically expressed KLF4 and PARP1. 293T cells were transfected with FLAG-KLF4 and Myc-PARP. Whole cell lysates or IP complex pulled down by anti-FLAG antibody were analyzed by Western blotting.

E   Validation of interaction between endogenous KLF4 and PARP1 in cytosolic lysate and nuclear lysate using immunoprecipitation and Western blotting in MDA-MB-231 cells. The Histone 3 (H3) is the control for nuclear portion, and actin is the control for cytosol portion.

F   Co-immunoprecipitation of PARP1 with endogenous KLF4 is independent on DNA. The DNA binding inhibitor EtBr was added to the MDA-MB-231 cell lyses followed by immunoprecipitation of KLF4 complex.

G, H   Co-localization analysis for endogenous PARP1 and KLF4 (G), or ectopic expressed GFP-PARP1 and DsRed-KLF4 (H) in MDA-MB-231 cells. PARP1 and KLF4 are co-localized in the nucleus, and this co-localization is increased by in response to radiation. Scale bars, 5 μm.

I   Validation of the interaction between endogenous KLF4 and PARP1 by in situ proximity ligation assay (PLA). No positive staining in KLF4/Rabbit IgG antibody PLA assay. Scale bar, 100 μm. The right panel shows the blow-up.

J   KLF4 was poly(ADP-ribosyl)ated. KLF4 was immunoprecipitated and detected by anti-PAR and anti-KLF4 antibodies in MDA-MB-231 cells.

K   The PARylation of KLF4 was increased after exposure to DNA damage. KLF4 complex was purified using immunoprecipitation from MDA-MB-231 cells that were treated with 5 μM doxorubicin and were then collected at different times (1, 2, 4 and 8 h). PARylation was detected by using antibody against PAR.

L   Loss of PARP1 attenuates KLF4 PARylation. PARP1$^{+/+}$ and PARP1$^{-/-}$ MEFs cells were transfected with FLAG-KLF4 followed by 4h after 5 Gy radiation, and then the PARylation of KLF4 was detected by pulldown.

M   PARP1 inhibitors decrease KLF4 PARylation. U2OS cells were pretreated with 10 μM various PARP1 inhibitor niraparib, olaparib, and rucaparib for 1hr followed by exposure to 10 Gy radiation for 4 h. KLF4 was pulled down, and the PARylation was detected.

both cytosol and nuclear fractions (Fig 2E), and their binding shows genuine DNA-independent protein associations since the interaction was not affected by ethidium bromide (EtBr) (Fig 2F) (Lai & Herr, 1992). Consistent with the co-IP validation, we also observed increased co-localization and foci formation between KLF4 and PARP1 in response to DNA damage signal, as measured by immunostaining coupled with examination by confocal microscope (Fig 2G and H). We further confirmed the interaction between KLF4 and PARP1 mainly in the nuclear in MDA-MB-231 cells by in situ proximity ligation assay (PLA) (Fig 2I). In addition, using immuno-precipitation of endogenous KLF4 followed by immunoblotting with antibody against PAR (poly ADP-ribose chain), we observed that KLF4 is tightly regulated by PARylation that in turn facilitates DNA damage response (Fig 2J and K). Finally, we observed that disruption of PARP1 by PARP1 knockout or blockade of PARP1 by PARP inhibitors abolished KLF4 PARylation, suggesting that PARP1 is a physiological binding partner and regulator for KLF4, and this interaction is largely enhanced in response to DNA damage signal (Fig 2L and M).

## Abnormal KLF4-PARP1 axis correlates with poor prognosis for breast cancer

Impaired KLF4 or PARP1 has been previously linked to breast cancer formation (Foster et al, 2000; Pandya et al, 2004; Rowland et al, 2005; Fletcher et al, 2011; Rojo et al, 2012; Li et al, 2013; Green et al, 2015; Orr et al, 2015). To examine the clinical relevance of the KLF4-PARP1 axis to breast cancer development, we have measured the protein expression levels of KLF4 and PARP1 in various types of breast cancer cells as well as human breast tumor tissue specimens in comparison with adjacent normal tissues, by using Western blotting and immunohistochemistry (IHC). As shown in Fig 3A, compared to normal mammary gland epithelial cells, MCF10A and MCF12A, KLF4 accumulation was detected in various TNBC cell

**Figure 3. Abnormal KLF4-PARP1 axis correlates with breast cancer poor prognosis.**

A   Expression of PARP1 and KLF4 in mammary gland epithelial cell and various types of breast cancer cell lines.

B   Accumulation of PARP1 and KLF4 protein was detected in human breast cancer tissues with 183 breast invasive ductal carcinoma and 10 pairs of primary cancer and adjacent normal tissue specimens by IHC in comparison with adjacent normal breast tissues. Scale bars, 100 μm.

C, D   Statistic results of PARP1 and KLF4 IHC staining. Normal (n = 10), tumor (n = 183); P = 4.36 × 10$^{-7}$ (KLF4), P = 0.0473 (PARP1). One-way ANOVA was used for the statistical analysis.

E   Statistical analysis of PARP1 staining among normal (n = 10), ER/PR-positive (ER/PR) (n = 68), HER2-positive (n = 62), and triple-negative breast cancer (TNBC) (n = 48). The P values were labeled in figure, and one-way ANOVA was used for the statistical analysis.

F   Statistical analysis of KLF4 protein staining among normal (n = 10), ER/PR-positive (n = 68), HER2-positive (n = 62), and TNBC (n = 47). The P values were labeled in figure, and one-way ANOVA was used for the statistical analysis.

G, H   Elevated expression of PARP1 and KLF4 is significantly correlated in 183 breast invasive ductal carcinoma human breast cancer tissue specimens (same batch of staining as (B)). Represents of paired IHC staining of PARP1 and KLF4 (case 1–3) are shown (G). Scale bars, 100 μm. (H) Statistic analysis of IHC staining indicates that PARP1 expression is positively correlated with KLF4 expression in breast cancer. n = 183, r = 0.434, P = 8.68 × 10$^{-10}$, Pearson correlation coefficients.

I   Survival analysis of KLF4 protein expression in 117 breast cancer patients. Compared to patients with low KLF4 protein expression, patients with high KLF4 levels had an inferior cumulative survival rate (Kaplan–Meier assay, LogRank P = 0.012); low, staining weak and moderate; and high, staining strong.

J   Survival analysis of both KLF4 and PARP1 protein expression in 66 breast cancer patients. Compared to patients with both low KLF4 and PARP1 protein expression levels, patients with both high KLF4 and PARP1 levels had lower cumulative survival rate (Kaplan–Meier assay, LogRank P = 0.022).

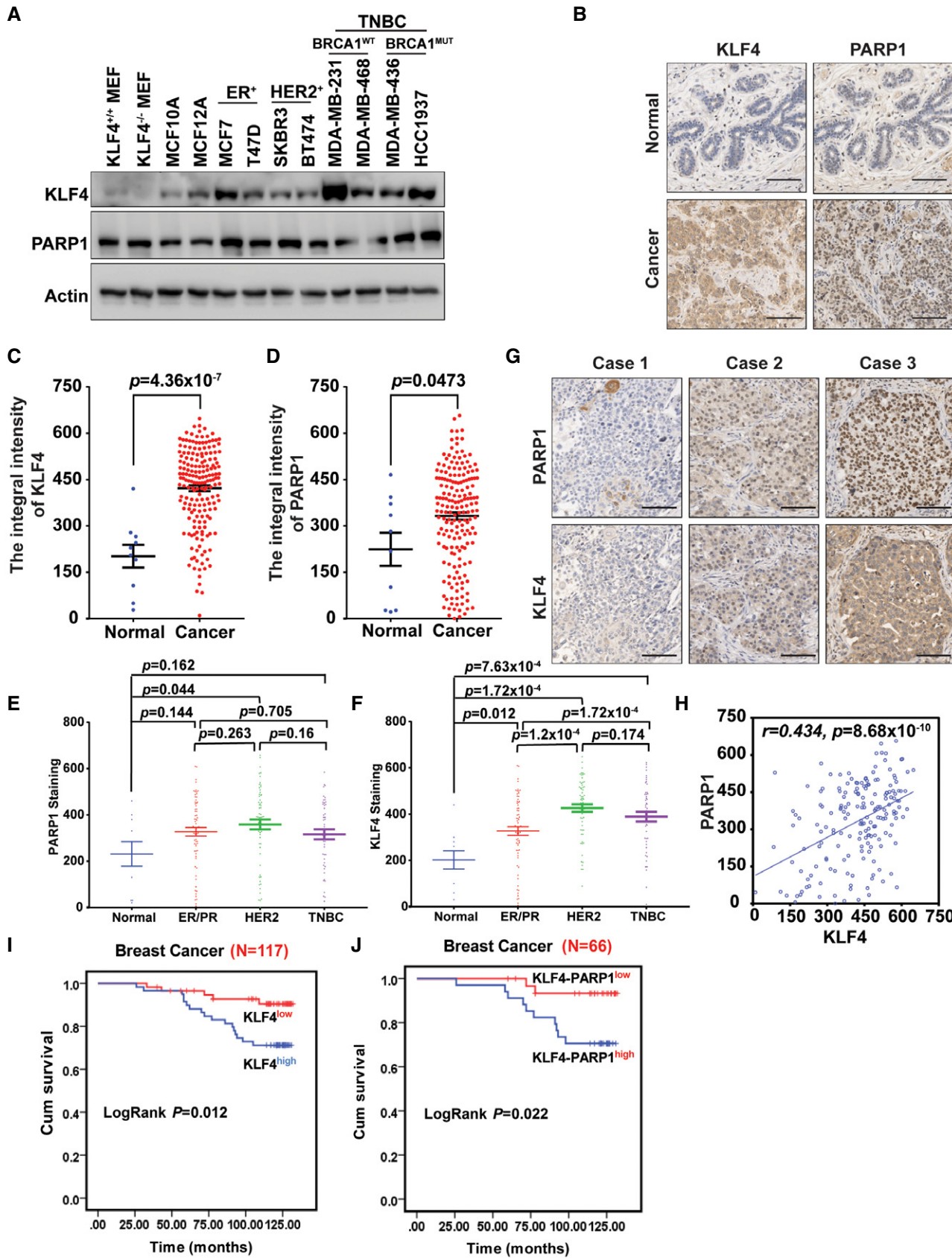

Figure 3.

lines such as MDA-MB-231, MDA-MB-468, HCC1937, and MDA-MB-436. Elevated PARP1 levels were also observed in HCC1937 and MDA-MB-436 cell lines. Moreover, significant KLF4 accumulation was detected in TNBC cells such as MDA-MB-231 and HCC1937. Furthermore, tissue arrays of 183 breast invasive ductal carcinoma (Appendix Table S2) and 10 adjacent normal tissue specimens were examined by IHC with anti-KLF4 and anti-PARP1, and visualized by DAB staining. As shown in Fig 3B–D, both KLF4 and PARP1 protein levels were significantly higher in breast tumor tissues than in adjacent normal tissues. Surprisingly, the expression of KLF4 was significantly correlated with the expression of PARP1 and vice versa in breast cancer tissues, which indicates that KLF4 and PARP1 are co-accumulated in most breast cancer tissue including TNBC, HER2-positive, and ER/PR-positive breast cancer (Fig 3C–H).

In order to analyzing the correlation of the KLF4-PARP1 axis with poor breast cancer prognosis, we have conducted additional immunohistochemistry measuring the expression levels of KLF4 and PARP1 based on an array of 117 human breast cancer tissue samples followed by Kaplan–Meier analysis. As shown in Fig 3I and J, and Appendix Fig S3B–D and Table S3, we observed that patients with high KLF4 expression levels have shorter cumulative survival, while patients with lower KLF4 expression levels have relatively long cumulative survival. Furthermore, we observed that the subset with high expression for both KLF4 and PARP1 has a shorter cumulative survival window than the population with lower expression of both KLF4 and PARP1 (Fig 3I). The expression of KLF4 and PARP1 is also significantly correlated with each other in these 117 cases of breast cancer tissues (Appendix Fig S3E and F). Furthermore, the co-expression of KLF4 and PARP1 in breast cancer cell lines and breast cancer tissues was detected with accumulation of KLF4 PARylation. Co-localizations of both PARP1 and PAR with KLF4 were measured in breast cancer tissue specimens (Appendix Fig S3G and H). In addition, we also observed that KLF4 PARylation tends to be present in greater numbers in TNBC cells such as MDA-MB-231 and MDA-MB-468 as compared to normal mammary gland epithelial cell lines MCF10A and MCF12A (Appendix Fig S3I). Thus, abnormal accumulation of KLF4 and PARP1 protein levels in breast cancer tissues is associated with poor prognosis.

### Identification of molecular motifs on KLF4 and PARP1 that mediate KLF4 PARylation

To characterize the mechanism of recognition of KLF4 by PARP1, we examined the molecular motifs that facilitate their interaction on KLF4 as well as PARP1 (Zhou *et al*, 2017b) (Fig 4A–G). To this end, a series of deletion mutants for both KLF4 and PARP1 were engineered as illustrated in Fig 4A, C and F. Co-transfection of a set of KLF4 deletion mutants and PARP1 deletion mutants, respectively, followed by co-immunoprecipitation led to an initial assessment of the binding region between KLF4 and PARP1 (Zhou *et al*, 2017b). As shown in Fig 4B, the mapping result indicates that the amino acid 411–441 stretch (Zinc finger 2) on KLF4 is critical to mediating its interaction with PARP1. Furthermore, this region (411–441) is also involved in mediating the PARylation of KLF4 catalyzed by PARP1 (Fig 4D and E). As for PARP1, the mapping results indicate that the amino acid stretch of 829–1,014 on the C-terminal region is involved in mediating the interaction with KLF4, thereby facilitating KLF4 PARylation (Fig 4F and G).

### Mechanistic role of DNA damage-induced KLF4 PARylation in orchestrating the recruitment of KLF4 to chromatin

We next asked what the molecular mechanism and physiological consequence of PARP1-mediated KLF4 PARylation is. To this end, we first investigated the exact site on KLF4 that is targeted by PARP1 for PARylation. We noted that a structurally resolved histone chaperone, aprataxin PNK-like factor (APLF), known to act as a modulator for PARylation, could serve as the template for modeling KLF4-PARP1 interactions. Accordingly, we have performed homology modeling and molecular docking simulations to identify the possible binding site for the PARylation chain on KLF4. In accordance with the known binding pose of RFA (2′-O-α-D-ribofuranosyl adenosine) onto the protein-binding zinc (PBZ) domain of APLF (PDB id: 2KQE) (58), the Y430/Y451/R452 motif (YYR) on the zinc finger II of KLF4 (PDB id: 2WBS) PBZ domain has been identified as the potential PARylation site (Fig 5A and Appendix Fig S4A–C). To test the possible role of the YYR motif (on zinc II domain) on KLF4 PARylation compared to that of the YKH motif (on zinc I domain), we have engineered a set of KLF4 mutants, including triple mutants at the YYR (Y430A/Y451A/R452A) or YKH (Y411A/K413A/ H416A) motifs, as well as a single Y451A mutation on KLF4 and then transfected to 293T cells. As shown in Fig 5B and Appendix Fig S5A, while KLF4 PARylation dramatically decreased for the KLF4-YYR mutant Y430A/Y451A/R452A and single Y451A mutation, as measured by KLF4 immunoprecipitation followed by Western blotting using anti-PAR antibody, no significant effect was observed for wild-type or the KLF4 YKH mutant (Y411A/K413A/ H416A), indicating that the YYR motif on KLF4 plays a critical role in mediating KLF4 PARylation. According to computation modeling, the tyrosine 451 site was expected to be the RFA docking site (Fig 5A and Appendix Fig S4A–C). The mutant of tyrosine 451 to alanine almost abrogated the PARylation modification as the KLF4-YYR mutant Y430A/Y451A/R452A, which further implicated tyrosine 451 is the site for the PAR chain to attach (Fig 5B).

To test whether the replacement of Y430/Y451/R452 (YYR motif) by alanines affects other types of posttranslational modifications previously reported, we have examined KLF4 methylation, sumoylation, acetylation, and ubiquitylation using the KLF4-Y430A/Y451A/ R452A mutant in 293T cells (Evans *et al*, 2007; Meng *et al*, 2009; Hu & Wan, 2011; Hu *et al*, 2015; Tian *et al*, 2015; Zhou *et al*, 2017a). As shown in Fig 5C, the results indicated that the YYR motif mutation only attenuates KLF4 PARylation but not other types of posttranslational modifications, suggesting YYR's exclusive role is mediating KLF4 PARylation.

To determine the physiological relevance of KLF4 PARylation to DNA damage response, we have examined how KLF4 PARylation affects several features of KLF4, including its stability, cytosol/nuclear translocation, and chromatin recruitment in KLF4 overexpressed cell line MDA-MB-231. While no effects on protein stability or cytosol/nuclear translocation were observed, to our surprise, the fractionation analyses indicated that doxorubicin dramatically induced chromatin accumulation of KLF4. Furthermore, this DNA damage-induced chromatin accumulation of KLF4 was attenuated by inhibiting PARP1 expression and activity (Fig 5D and Appendix Fig S5B–D). Similarly, dramatically induced chromatin accumulation of KLF4 was observed by using γ-radiation (Fig 5E and F and Appendix Fig S5E). Interestingly, interference by

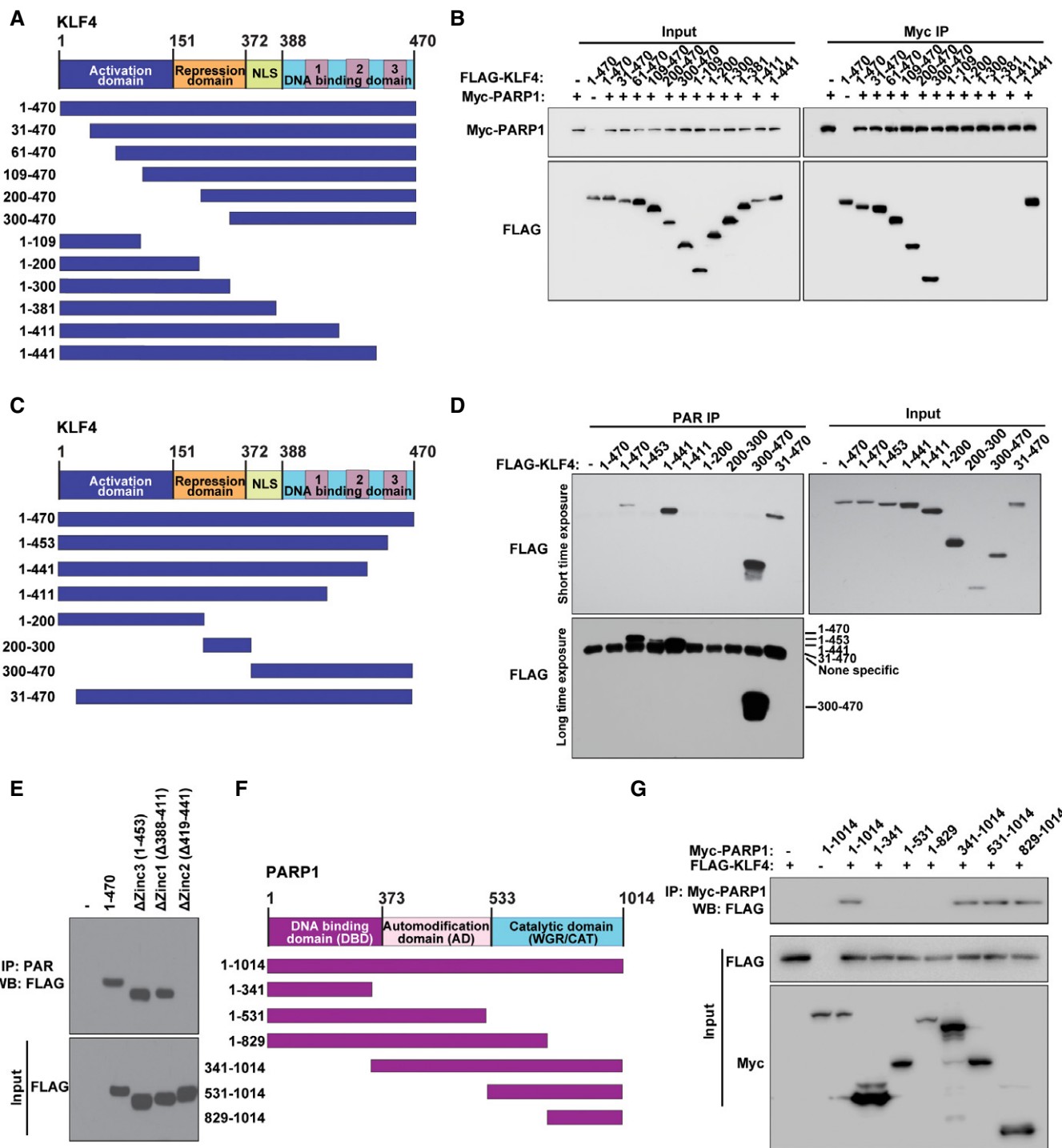

**Figure 4. Identification of molecular motifs on KLF4 and PARP1 that mediate KLF4 PARylation.**

A   Schematic diagram of human KLF4 functional domains and strategy for engineering a series of KLF4 deletion mutants.

B   Identification of amino acid stretch 411–441 on the repression domain of KLF4 as a segment mediating the interaction with PARP1 in 293T cells.

C, D   Mapping of poly(ADP-ribosyl)ation domain on KLF4. (C) Schematic diagram of human KLF4 functional domains and strategy for engineering a set of KLF4 deletion mutants. (D) Identification of amino acid stretch 411- 441 on the Zinc 2 domain of KLF4 involved in mediating its PARylation modification in 293T cells. PARylation for FLAG-tagged KLF4 deletion mutants was analyzed, respectively, by pulldown using PAR antibody followed by Western blotting using antibody against FLAG.

E   Dissection of the PARylation region of KLF4 into three zinc finger motif-containing domains, Zinc 1, Zinc 2, and Zinc 3. While no effect was observed on Zinc 1 and Zinc 3 domains, deletion of Zinc 2 domain led to significant attenuation of KLF4 PARylation.

F   Schematic diagram of human PARP1 functional domains and strategy for engineering a series of PARP1 deletion mutants.

G   Identification of amino acid stretch 829–1,014 on the C-terminal segment of PARP1 to be involved in mediating its interaction with KLF4 in 293T cells.

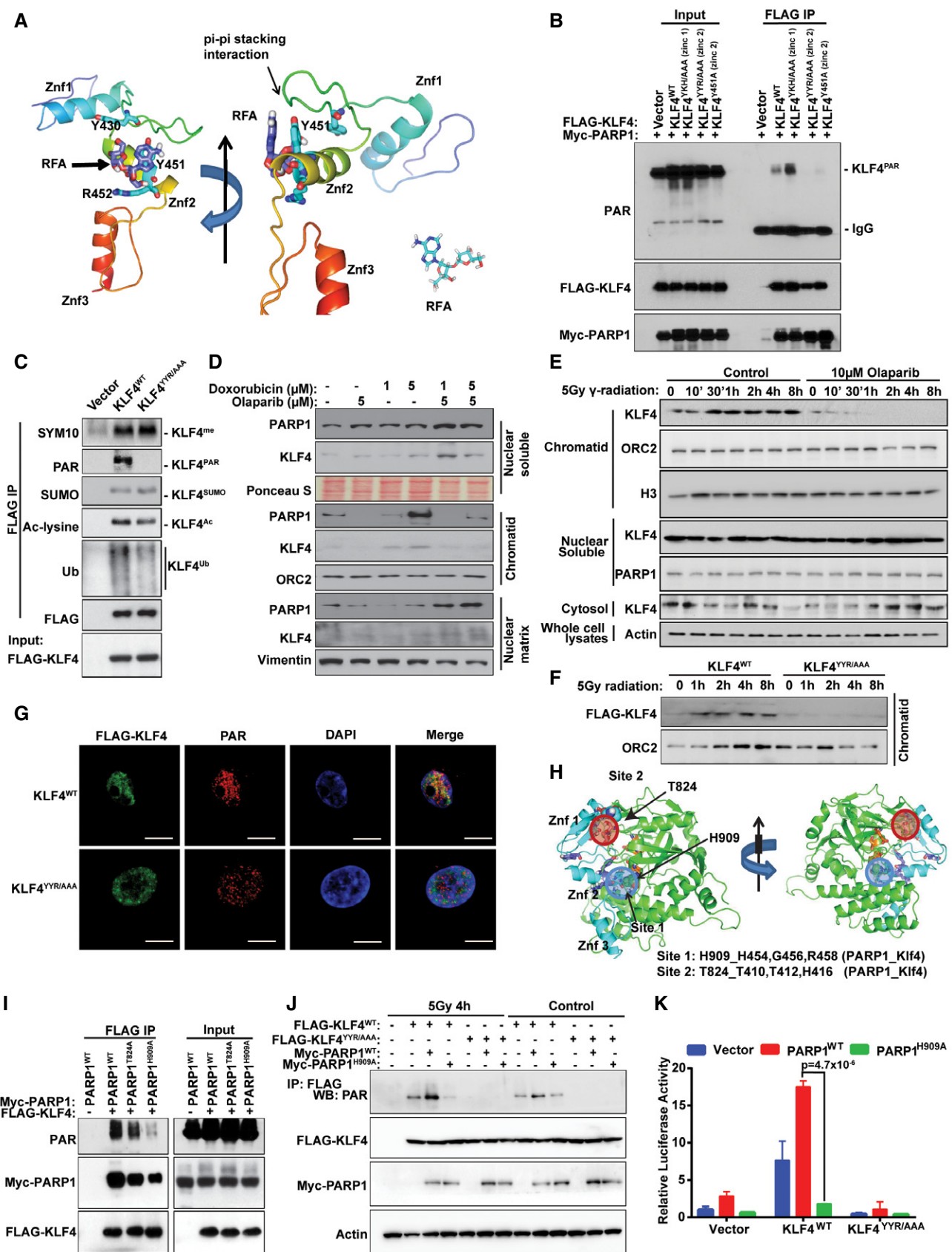

**Figure 5.**

**Figure 5.  Mechanistic role of DNA damage-induced KLF4 PARylation in orchestrating the recruitment of KLF4 to chromatin.**

A   Simulation analysis of the potential KLF4 PARylation site (PBZ domain) with or without DNA binding. The triple amino acids Y430, Y451, and R452 on KLF4 zinc finger 2 (Znf2) are identified as the potential YYR motif involved in mediating PARP1 modification.

B   The effect of YYR motif mutations on KLF4 PARylation. Constructs of KLF4-Zinc 1-YKH/AAA mutation, KLF4-Zinc 2-YYR/AAA mutation, and KLF4-Zinc-2-Y451A mutation were co-transfected with Myc-PARP1 into 293T cells, respectively, and then pulled down using M2 agarose followed by measuring KLF4 PARylation with anti-PAR antibody.

C   KLF4-Zinc 2-YYR/AAA mutation has only effect on KLF4 PARylation but not on other types of posttranslational modifications such as protein methylation, ubiquitylation, acetylation, and sumoylation in 293T cells. The FLAG-tagged wild-type or mutant KLF4 were immunoprecipitated with FLAG and detected by anti-SYM10 (methylation), anti-SUMO (sumoylation), anti-ac-lysine (acetylation), and anti-PAR (PARylation).

D, E   Effect of PARP1 on the distribution of KLF4 subcellular localization among cytosol, soluble nucleus, and chromatin in MDA-MB-231 cells. While the activation of PARP1 in response to genotoxic stress enhances chromatin recruitment of KLF4 (D), inhibition of PARP1 by olaparib attenuates the KLF4 accumulation in chromatin fraction (E). ORC2 and H3 were the loading control for chromatid fraction. Ponceau S and PARP1 were the control of nuclear soluble fraction.

F   KLF4-Zinc 2-YYR/AAA mutation remarkably reduces DNA damage-induced chromatin recruitment of ectopic expressed KLF4 in U2OS. ORC2 is the loading control for chromatid fraction.

G   Confocal analysis of PAR and FLAG-tagged KLF4 showing that KLF4-Zinc 2-YYR/AAA mutation significantly reduces KLF4 PARylation in U2OS cells. Scale bars, 10 μm.

H   Two interaction sites are observed in PARP1-KLF4 interaction pose. One of them is at znf1 and the other is at znf2 (at the same site where RFA binds). H909 and T824 on PARP1 are the potential core amino acids mediating the interaction pose.

I   The effect of PARP1 mutations (H909A or T824A) on KLF4 PARylation, confirming the critical role of H909 in mediating KLF4 PARylation in 293T cells. Myc-PARP1 wild-type or mutations (H909A or T824A) and FLAG-KLF4 were co-transfected into 293T cells and then the FLAG-KLF4 was pull down by anti-FLAG antibody, the PARylation modification of KLF4 was blotted with anti-PAR antibody, and the binding PARP1 was blotted with anti-myc antibody.

J   Replacement of this histidine by alanine on PARP1 or replacement of the YYR motif of KLF4 by triple alanines leads to the attenuation of KLF4 PARylation. In PARP1$^{-/-}$ MEFs, wild-type or mutant PARP1 (H909A) were co-transfected with FLAG-tagged wild-type or KLF4-Zinc 2-YYR/AAA mutation followed by immunoprecipitation using M2-agarose and Western blotting by anti-PAR antibody.

K   KLF4 PARylation enhances its transcriptional function demonstrated by p21 promoter luciferase assay. 293T cells were co-transfected with p21 luciferase reporter plasmid with wild-type or mutant KLF4 (YYR/AAA) and wild-type or mutant PARP1 (H909A) and then submitted to luciferase assay. $n = 3$, $P = 4.74 \times 10^{-6}$ (PARP1$^{WT}$ vs. PARP1$^{H909A}$). Data are mean ± SEM; one-way ANOVA was used for the statistical analysis.

expression of a KLF4 PARylation-deficient mutant (KLF4$^{YYR/AAA}$) significantly decreased chromatin accumulation of KLF4 in U2OS cells (Fig 5F), suggesting the consequence of KLF4 PARylation is to promote its recruitment to chromatin. Results from immunostaining of PARylation and FLAG-tagged KLF4 examined by confocal microscope also showed YYR mutation of KLF4 remarkably reduced KLF4 co-localization with PARylation (Fig 5G).

It has been previously demonstrated that the DNA damage-induced regulation of KLF4 orchestrates transcription of p21 or Bax, which further determined the cell fate (Ghaleb *et al*, 2007; Zhou *et al*, 2009; El-Karim *et al*, 2013). To test whether KLF4 PARylation affects DNA damage-mediated KLF4 transcriptional function, we have measured the interaction between wild-type KLF4 or mutant KLF4 with promoter of p21 or Bax using a chromatin immunoprecipitation assay (ChIP) followed by measuring KLF4-mediated trans-activation using a luciferase assay (Komata *et al*, 2014). As shown in Appendix Fig S5F–J, the disruption of KLF4 PARylation did not affect KLF4 binding to DNA, which indicated the conserved function for zinc finger domain in binding to DNA. However, KLF4 PARylation resulted in decreased association between KLF4 and promoter of either p21 or Bax in ChIP assay that, in turn, attenuated gene transcription, suggesting a critical function for KLF4 PARylation in recruiting KLF4 protein to p21 or Bax promoter.

To further decipher the PARP1 residues involved in binding KLF4 and enabling its PARylation, we performed molecular docking simulations using the known structures for PARP1 (PDB id: 4OQB) (Patel *et al*, 2014) and KLF4 (PDB id: 2WBS) (Schuetz *et al*, 2011). Simulations revealed two putative binding poses for KLF4-PARP1 interactions (Appendix Fig S6A–F). PARP1 residues Y907 and H909 (site 1) were found to make interfacial contacts with KLF4 zinc finger motifs 1 or 2 in both poses; whereas PARP1 T824 was found to zinc finger 1 in one of the poses (Fig 5H and Appendix Fig S6A–F). To validate the role of H909 and/or T824 in mediating KLF4

PARylation, we have co-transfected wild-type or mutant PARP1 together with FLAG-KLF4 and then measured KLF4 PARylation by immunoprecipitation of KLF4 followed by immunoblotting using anti-PAR antibody. As shown in Fig 5I, while wild-type PARP1 or Myc-PARP1 (T824A) allows for KLF4 PARylation, the replacement of H909 by alanine dramatically reduces KLF4 PARylation, suggesting that H909 on PARP1 is critical to enabling KLF4 PARylation. The role of H909 on PARP1 is further confirmed by rescue experiment based on restoration of KLF4 PARylation by adding back wild-type PARP1 or Myc-PARP1 (H909A) to PARP1 knockout MEF cells (Fig 5J and K). Taken together, our molecular characterization has revealed that KLF4 PARylation by PARP1 governs the recruitment of KLF4 to chromatin that is important to ensure KLF4's transcriptional function.

## Mechanistic insight into PARylated KLF4 and PARylation-independent KLF4 in DNA damage response and DNA repair

The above characterization clearly demonstrated that the role of DNA damage-induced KLF4 PARylation in the recruitment of KLF4 to chromatin allows its transcriptional function. To analyze the impact of KLF4 PARylation in DNA damage response, we have further assessed the effect of KLF4 PARylation to determine cell fate after exposure to genotoxic stress by regulating the expression of p21 and Bax (Zhang *et al*, 2000; Zhou *et al*, 2009). As shown in Fig 6A, while γ-radiation at a physiological relevant dose (5 Gy) dramatically induces p21 accumulation that inhibits CDK1 and Histone 3 phosphorylation, knockdown KLF4 leads to downregulation of p21, resulting in increased CDK1 and Histone 3 phosphorylation in MDA-MB-231 cells.

Because p53 is also critical to controlling p21, we then asked whether PARylation of KLF4 regulated p21 through p53. We applied

p53 knockout HCT116 to dissect the role of p53 and discovered that upregulation of p21 by KLF4 depends on PARylation, but not on p53 (Appendix Fig S7A). Moreover, cellular staining with p-H3 further indicates that the deregulation of KLF4 PARylation causes significant cell cycle delay after cellular exposure to γ-radiation; the cell eventually develops severe DNA damage and genome instability

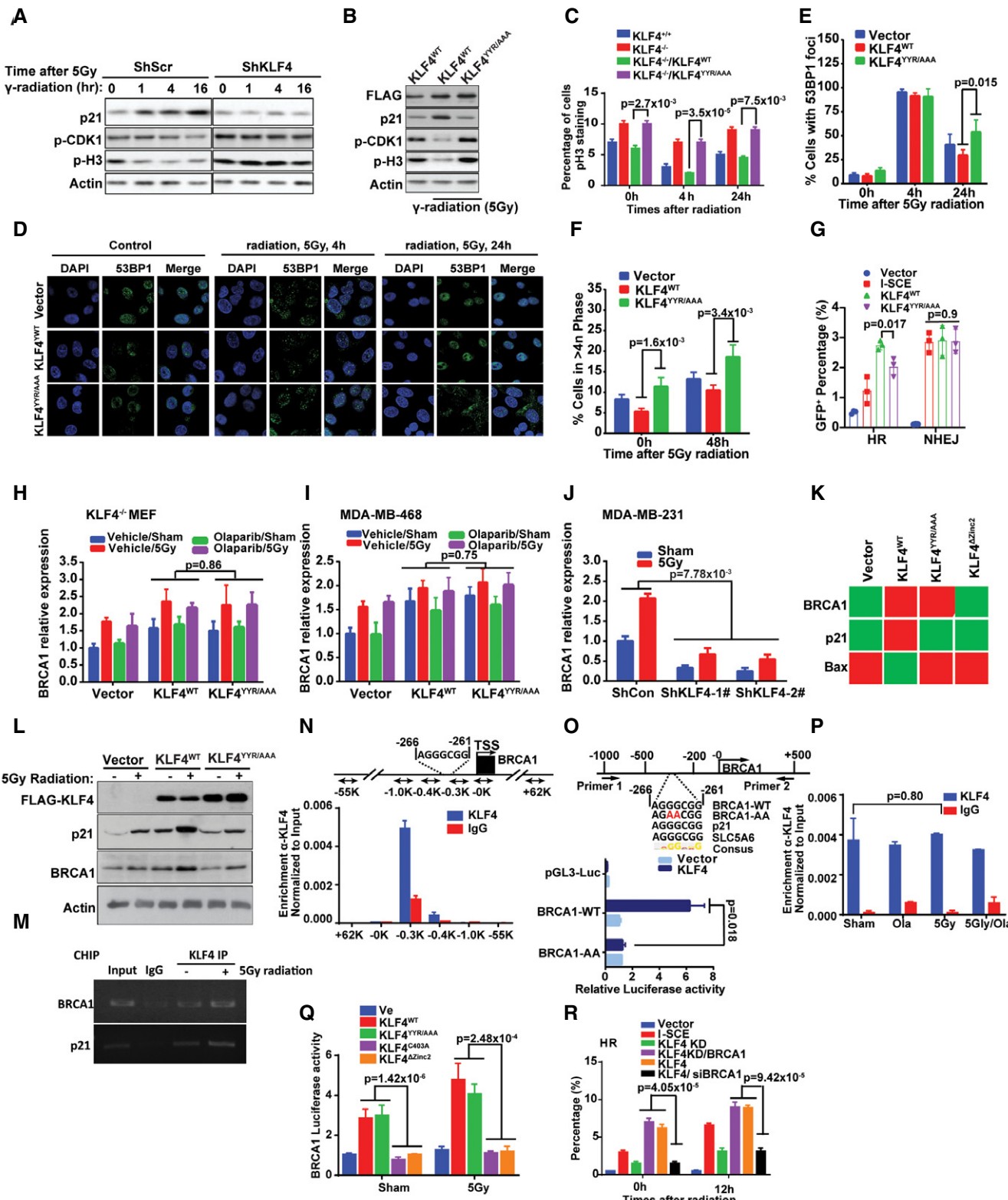

Figure 6.

◄

**Figure 6.　Mechanistic insights into PARylated KLF4 and PARylation-independent KLF4 in DNA damage response and DNA repair.**

A　Depletion of KLF4 directly diminishes the DNA damage-induced p21 expression, resulting in the failure of cell cycle arrest in MDA-MB-231 cells.

B　Abolishment of KLF4 PARylation disrupts KLF4-mediated p21 expression that in turn impairs DNA damage response in MDA-MB-231 cells.

C　p-H3 staining analysis of KLF4$^{+/+}$, KLF4$^{-/-}$ MEFs, KLF4$^{-/-}$ MEF with transfection of wild-type or KLF4-Zinc 2-YYR/AAA mutation. $n = 4$, $P$ values were labeled in figure, one-way ANOVA assay.

D, E　Abolishment of KLF4 PARylation leads to failure in removing damaged DNA as measured by 53BP1 foci. $n = 4$, $P = 4.74 \times 10^{-6}$ (KLF4$^{WT}$ vs. KLF4$^{YYR/AAA}$), one-way ANOVA assay. (E) Summary of (D).

F　Failure of KLF4 PARylation leads to increased aneuploidy population in U2OS cells. $n = 3$, $P$ values were labeled in figure, one-way ANOVA assay.

G　Effect of KLF4 PARylation on NHEJ and HR. Wild-type or KLF4-Zinc 2-YYR/AAA mutant KLF4 were co-transfected with I-SCE construction in U2Os-GFP-EJ5 cells (for NHEJ assay) or U2Os-DR-GFP (for HR assay). $n = 3$, $P$ values were labeled in figure, one-way ANOVA assay.

H–J　The effect of overexpression or knockdown KLF4 on mRNA levels of BRCA1 in KLF4$^{-/-}$ MEF (H), MDA-MB-468 (I) and MDA-MB-231 (J). $n = 3$, $P$ values were labeled in figures, one-way ANOVA assay.

K　Heatmap of BRCA1, P21, Bax expression on U2OS cells with expression of wild-type KLF4 or KLF4-Zinc 2-YYR/AAA mutant as well as KLF4-Zinc 2 (Zinc 2 domain deletion) mutant. No difference of BRCA1 expression was measured between expression of wild-type KLF4 or KLF4-Zinc 2-YYR/AAA mutant, while loss of Zinc domain causes drops of BRCA1 expression levels.

L　In early responsive window (1–2 h after exposure to γ-radiation), while elevation of wild-type KLF4 enhances the expression levels of p21, disruption of KLF4 PARylation diminishes KLF4-mediated p21 accumulation. Upon DNA damage, no matter PARylation or not, elevation of both wild-type or KLF4-PARylation-deficient mutant leads to BRCA1 accumulation suggesting the KLF4-mediated regulation of BRCA1 is independent of PARP1 in MDA-MB-231 cells.

M　KLF4 physically interacts with BRCA1 upstream promoter region measured by CHIP-PCR.

N　inset, top, schematic diagram of the BRCA1 promoter and relative positions of primer sets used in this study. ChIP analysis at the BRCA1 promoter using KLF4-specific or nonspecific control IgG (α-Gal4) in MDA-MB-231 cells. Shown is the enrichment at positions of the BRCA1 locus relative to the TSS, presented as percent recovery of input.

O　inset, top, schematic diagram of the BRCA1 promoter cloning primer and the alignment of potential KLF4 binding motif on BRCA1 promoter with KLF4 binding motif on p21 and SLC5A6 promoter. Shown is the wild-type (BRCA1-WT) or mutant (BRCA1-AA) BRCA1 promoter luciferase reporter activity when co-transfect with KLF4 plasmids. KLF4 co-transfection promotes BRCA1-WT but not BRCA1-AA promoter reporter transcription. $n = 3$, $P = 0.018$, one-way ANOVA assay.

P　ChIP analysis of KLF4 binding to the BRCA1 promoter in MDA-MB-231 at $-0.3K$ positions relative to the TSS in untreated and olaparib (10 μM for 8 h) or 5 Gy radiation treat cells. No significant difference of KLF4 binds to BRCA1 promoter between untreated and olaparib or radiation treat cells. $n = 3$, $P = 0.80$, one-way ANOVA assay.

Q　BRCA1 reporter assay in KLF4$^{-/-}$ MEFs, KLF4$^{-/-}$ MEF with transfection of wild-type (KLF4$^{WT}$) or mutant KLF4 (KLF4$^{YYR/AAA}$, KLF4$^{C403A}$ and KLF4$^{\Delta Zinc2}$). The depleted the zinc domain or mutated zinc finger (KLF4$^{C403A}$ and KLF4$^{\Delta Zinc2}$) on KLF4 impairs the KLF4-driven BRCA1 expression, while no effect was observed between wild-type and KLF4$^{YYR/AAA}$ mutant. $n = 3$, $P$ values were labeled in figure, one-way ANOVA assay.

R　HR analysis. U2OS-DR-GFP cells were transfected with I-SceI, BRCA1, and siBRCA1 in KLF4-wild-type and depletion condition, respectively. GFP-positive cells representing HR repair rate were measured by flow cytometry 48–72 h after then. Overexpression of BRCA1 restores the HR efficiency in KLF4 knockdown cells.

Data information: Data are mean ± SEM; $n = 3$, $P$ values were labeled in figure, one-way ANOVA assay.

as shown by measurements of 53 BP1 foci formation (Fig 6B–F and Appendix Fig S7B). In addition, the disruption of DNA damage signal-induced KLF4 PARylation leads to accumulated aneuploidy population (Fig 6F). Thus, DNA damage-induced KLF4 PARylation is important for governing DNA damage response through facilitating KLF4 chromatid recruitment. We went further by investigating whether the function of KLF4 PARylation in cell death depends on p53. We applied p53 knockout HCT116 to dissect the role of p53 and discovered that the cell-death prevention role of KLF4 PARylation is not correlated with the presence of p53 (Appendix Fig S7C).

To further examine whether KLF4 PARylation is involved in DNA repair, we have measured the effect of KLF4 PARylation on HR or NHEJ DNA damage repair (Hagos *et al*, 2009; El-Karim *et al*, 2013). We initially assessed the general role of KLF4 in DNA repair and then further determined if and how PARylation of KLF4 regulates DNA repair. Regarding the overall role of KLF4 in DNA repair, interestingly, our results from NHEJ and HR assays demonstrated that KLF4 only regulates HR; it does not regulate non-homologous end joining DNA repair (Fig 6G) (Wei *et al*, 2015). Results from further dissection of the impact of KLF4 PARylation showed that while KLF4 is involved in HR, PARylation of KLF4 is not necessary for either HR or NHEJ, suggesting a sophisticated regulation of KLF4 in DNA damage response and DNA repair, wherein recruiting of KLF4 to the promoter region for DNA damage responsive genes such as p21 and Bax needs modification of KLF4 by the polyADP-ribosylation chain, but KLF4 involvement in HR does not need the modification by polyADP-ribosylation.

Based on our observation that KLF4 participates in HR, we asked what the downstream target for KLF4 in the HR pathway is. Given the critical role of BRCA1 in HR, we decided to test the functional connection between KLF4 and BRCA1 in response to DNA damage signal (Powell & Kachnic, 2003). We subjected various types of cultured cells, including MEF, MDA-MB-231, and MDA-MB-468 cells, to γ-radiation and then measured the responsive alteration of BRCA1 expression. As shown in Fig 6H–J, to our surprise, our results indicate that BRCA1 expression is clearly regulated by KLF4 in KLF4-mediated HR DNA repair after cellular exposure to γ-radiation. Meanwhile, we also examined the effect of KLF4 PARylation on BRCA1 regulation in response to γ-radiation. As shown in Fig 6H–L and Appendix Fig S7D–I, while KLF4 plays a role in regulating BRCA1, no effect for KLF4 PARylation on BRCA1 was observed in KLF4$^{-/-}$ MEF, U2OS, and MDA-MB-231 as well as in p53-null background (HCT116 p53$^{+/+}$ vs. p53$^{-/-}$), suggesting PARylation of KLF4 is necessary only for DNA damage response but not for canonical ATM-CHK2-P53 pathway and KLF4-mediated regulation of BRCA1 and HR.

To further validate the role of KLF4 in regulating BRCA1 transcription, we have initially accessed whether KLF4 could directly bind to the BRCA1 promoter region both in the presence and the absence of genotoxic stress, using ChIP assay (Huang *et al*, 2014). As shown in Fig 6M, we observed that KLF4 is directly immunoprecipitated with the BRCA1 promoter region with or without application of γ-radiation. Further, we conducted ChIP analysis of the BRCA1 promoter using KLF4-specific or nonspecific control IgG (α-

Gal4) in MDA-MB-231 cells followed by PCR using six pairs of primers as indicated in Fig 6N. As compared to the nonspecific Gal4 antibody, a peak showing binding of BRCA1 in the proximity of the −0.3K proximal promoter region of the BRCA1 gene could be detected by α-KLF4 ChIP and quantitative real-time PCR. To further explore the underlying mechanism of how KLF4 binds to BRCA1, we have searched for possible KLF4 binding sites on the BRCA1 promoter region by blast analysis. Based on aligning the promoter of BRCA1 to well-known KLF4 targeting gene p21 and SLC5A6, we have identified a conserved AGGGCGG binding motif on the BRCA1 promoter region between −266 bp and −261 bp (Fig 6O) (Matys et al, 2003). We also observed that mutated AGGGCGG binding motifs on the BRCA1 promoter abrogate KLF4-driven BRCA1 transcription in luciferase promoter assay (Fig 6O). We then analyzed the direct binding between KLF4 to the BRAC1 promoter in the absence and presence of γ-radiation and observed consistent binding between KLF4 to the BRAC1 promoter without alteration, suggesting that KLF4-governed transcription regulation of BRCA1 depends on KLF4 abundance but not its efficiency in binding to the promoter (Fig 6P). Moreover, using KLF4$^{-/-}$ MEFs, and KLF4$^{-/-}$ MEFs with added back wild-type, YYR mutant (KLF4$^{YYR/AAA}$) and zinc finger loss of function mutant KLF4 (KLF4$^{C403A}$ is defect in zinc I domain and KLF4$^{\Delta Zinc2}$ is defect in zinc II domain), we further dissected the role of KLF4 PARylation in BRCA1 transcription and confirmed that KLF4-mediated BRCA1 transcription and expression does not need KLF4 PARylation, but it does need the zinc finger motif for its transcriptional function (Fig 6Q and Appendix Fig S7I). However, KLF4-mediated p21 transcription depends on both KLF4 PARylation and zinc finger motif (Fig 6B and Appendix Fig S7I and J).

To confirm the functional interaction between KLF4 and BRCA1 in HR DNA repair, we have tested the effect of BRCA1 in restoration of HR repair in KLF4 depletion cells. As shown in Fig 6R, overexpression of BRCA1 could rescue the impaired HR due to knockdown of KLF4, while BRCA1 depletion attenuates the promotion of HR efficiency induced by the elevation of KLF4. Recent studies

implicated loss of 53BP1 could alleviate the HR repair stress in BRCAness tumor (Kakarougkas et al, 2013). Therefore, we performed 53BP1 knockdown in KLF4$^{-/-}$ MEF cell and found that 53BP1 siRNA could partial recovery the HR activity in KLF4$^{-/-}$ MEFs (Appendix Fig S7K and L). Taken all together, our results demonstrated the mechanistic role of KLF4 PARylation in governing DNA damage response. We also confirmed the importance of KLF4 of regulating BRCA1 in HR, although this regulation is independent of PARylation.

## KLF4-PARP1 axis plays a critical role in regulating DNA damage response and tumorigenesis

So far, we have demonstrated that regulation of KLF4 by PARP1-mediated poly ADP-ribosylation is sophisticated. While DNA damage signal-induced PARylation of KLF4 by PARP1 is required for the transcription of KLF4-mediated DNA damage responsive genes, regulation of BRCA1 by KLF4 is independent of PARylation. This observation suggests KLF4 is a versatile transcriptional factor, for which the subset of transcriptome is potentially governed by various posttranslational modifications. Likely, the chromatin recruitment of PARylated KLF4 only facilitates the DNA damage responsive gene network, but it does not apply to the recruitment of KLF4 to the BRCA1 promoter. Knowing the critical role of PARylated KLF4 in governing the DNA damage response and genome stability through regulating p21 and Bax, we asked whether KLF4 PARylation impacts the DNA damage response-induced cell death and tumorigenesis due to genome instability. To date, we have measured the effect of KLF4 PARylation on DNA damage-induced cellular apoptosis. As shown in Fig 7A, in KLF4$^{-/-}$ MEF cells, the addback of KLF4 rescued the cell death in response to a DNA damage signal, while the PARylation-deficient KLF4 mutant failed to restore DNA damage-induced apoptosis. This result was replicated in U2OS cells by using various DNA damage drugs (Fig 7B). KLF4 has been reported to inhibit RasV12-induced senescence in a BTR

**Figure 7. Synergism of KLF4 and PARP1 in breast cancer treatment.**

A  KLF4$^{+/+}$, KLF4$^{-/-}$ MEFs, KLF4$^{-/-}$ MEF with wild-type KLF4 or KLF4-Zinc 2-YYR/AAA mutant were treated with 5–10 µM cisplatin for 48 h followed by measuring the expression of PARP1, p21, and Bax. Loss of PARylation on KLF4 impairs KLF4-mediated inhibition of apoptosis in the presence of genotoxic stress.

B  The apoptotic response in U2OS cells with expression of wild-type and mutant KLF4. U2Os cells with wild-type (pLenti-tet-on-KLF4$^{WT}$) or mutant (pLenti-tet-on-KLF4YYR/$^{AAA}$) KLF4 was incubated with 10 ng/ml doxycycline for 24 h and then treated with 10 µM doxorubicin (Dox) or cisplatin (CDDP) for additional 24 h. While expression of wild-type KLF4 decreases Bax expression and inhibits drug-induced PARP1 cleavage, expression of PARylation-deficient KLF4 leads to failure in inhibiting genotoxic-induced Bax expression and PARP1 cleavage.

C  Disruption of KLF4 PARylation leads to failure in inhibiting temperature-induced senescence in BTR model (RAS V12-induced senescence).

D–F  Disruption of KLF4 PARylation abolishes the KLF4-promoted cellular transformation in MDA-MB-231 (D), MCF10A (E) and U2OS (F) measured by soft agar analysis. $n = 3$, $P$ values were labeled in figure, one-way ANOVA assay.

G, H  KLF4 sensitizes cell to olaparib. The asterisk in the panels represents the significant difference ($P < 0.05$). (G) The representative of MEFs clonogenic assay. (H) is the summary of (G).

I  Disruption of KLF4 PARylation reduces KLF4 effect on olaparib efficacy.

J, K  While elevated KLF4 inhibits PARP1 inhibitor efficacy in killing MDA-MB-231 cells, disruption of KLF4 PARylation partially rescues the KLF4-mediated resistance to olaparib. (J) The representative of MEFs clonogenic assay. (K) is the summary of (J).

L  Elevated KLF4 expression increases resistance to olaparib in MDA-MB-468 (BRCA-proficient) cells, while disruption of KLF4 PARylation attenuates KLF4-driven resistance to olaparib.

M, N  Elevation of mutant KLF4 shows the same effect as WT KLF4 to olaparib in BRCA1 mutant cell lines MDA-MB-436 (M) and HCC1937 (N) cells. No significant difference between expression of wild-type KLF4 and KLF4 PARylation-deficient mutant was observed in both MDA-MB-436 and HCC1937 cells.

O, P  Modulating KLF4 by knockdown or overexpression affects cellular response of MDA-MB-231 (O) and MCF7 (P) to olaparib in clonogenic assay.

Q, R  Modulating KLF4 by knockdown or overexpression significantly affects synergism on olaparib/doxorubicin or olaparib/cisplatin in MDA-MB-231(Q) or MCF7 (R).

Data information: Data are mean ± SEM; one-way ANOVA was used for the statistical analysis. The asterisk in the panels represents the significant difference ($P < 0.05$). The exact $P$-values are supplied in Appendix Table S4.

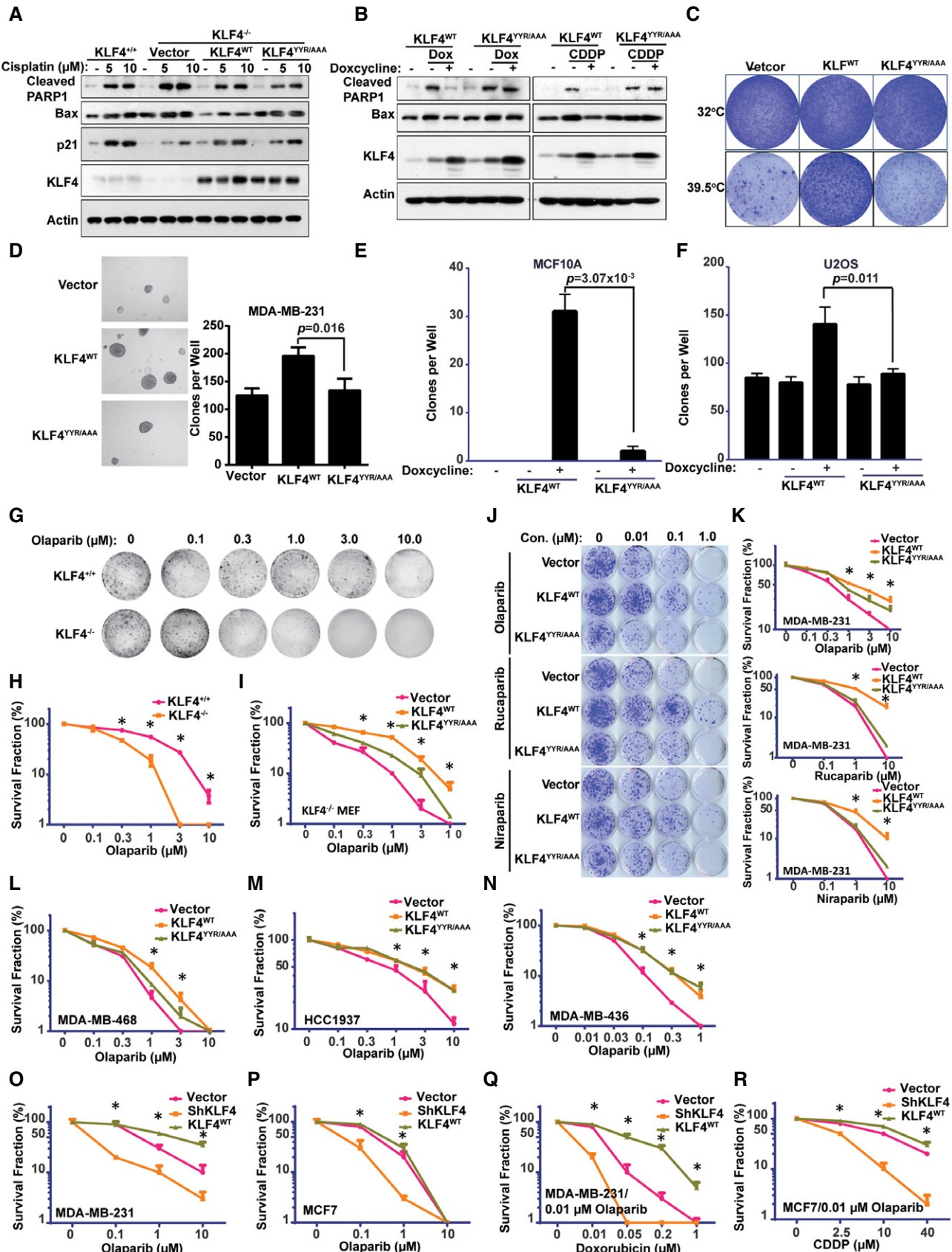

**Figure 7.**

MEFs model, which were conditionally immortalized with a temperature-sensitive (ts) mutant of SV40 large T antigen and co-expressing RasV12 (Rowland *et al*, 2005; Hu *et al*, 2015). The BTR cells are transformed at 32°C, but undergo RasV12-induced senescence at 39.5°C, when ts, large T antigen is inactive. We observed that the failure in KLF4 PARylation increases a barrier for induction of tumor senescence in this BTR model (Fig 7C; Peeper *et al*, 2002). We further observed that, while elevated KLF4 enhances colony formation in MDA-MB-231, MCF10A as well as U2OS cells, the KLF4 PARylation mutant loses its capability to promote tumor cell colony formation (Fig 7D–F), suggesting KLF4 PARylation plays a critical role in regulating DNA damage response-associated cell death, tumor initiation, and tumor progression.

**Suppression of KLF4 significantly increases the PARP inhibitor-induced killing efficacy for BRCA1-proficient cells**

Development of a PARP inhibitor has shed light on a synthetic lethal strategy to kill cancer cells in which BRCA1 dysfunction causes HR deficiency (McCann & Hurvitz, 2018; Sulai & Tan, 2018). Nevertheless, a PARP inhibitor has not shown an ideal killing effect in treating BRCA1-proficient TNBC cells. Given the observation that KLF4 is involved in regulating BRCA1 transcription (Fig 7), we asked whether inactivation of KLF4 could lead to inhibition of BRCA1 that, in turn, results in increasing the killing efficacy for BRCA1-proficient cells in response to a PARP inhibitor. We initially measured the overall effect of KLF4 on olaparib (PARP inhibitor)-induced cell death and observed that loss of KLF4 in KLF4$^{-/-}$ MEF cells significantly enhances olaparib-induced cell death (Fig 7G and H and Appendix Fig S7I). Moreover, we observed that expression of wild-type KLF4, but not the KLF4 PARylation-deficient mutant, improves cancer cell survival in the presence of olaparib (Fig 7I). In BRCA1-proficient TNBC cells, we observed the same result—that elevated wild-type KLF4 drastically promotes cancer cell survival in the presence of olaparib, whereas little effect was observed for expression of the PARylation-deficient KLF4 mutant (Fig 7J–L). Intriguingly, in BRCA1-deficient TNBC cells such as MDA-MB-436 and HCC1937, no significant difference in olaparib-induced cell death was observed between expression of wild-type KLF4 and the PARylation-deficient KLF4 mutant (Fig 7M

and N and Appendix Fig S7J). Results from further clonogenic assay showed that depletion of KLF4 in breast cancer cells significantly enhanced that efficacy of olaparib and other chemotherapy agents such as doxorubicin and cisplatin in killing cancer cells (Fig 7O–R). Collectively, our results demonstrate KLF4 could be a good clinical target in breast cancer treatment. Alteration of KLF4 levels could be a good strategy to promote the efficacy of olaparib in killing breast cancer cells, especially for the population of TNBC patients who have normal BRCA1 function.

**Suppression of KLF4 significantly synergizes PARP inhibitor in killing TNBC tumor**

KLF4 favors cancer cell survival and escape from treatment, depending on cancer type and patient population. Thus, blocking KLF4 could enhance cancer cell death and thereby benefit cancer therapy. Our discovery of the impact of KLF4 in orchestrating DNA damage response and HR provides a novel insight to synergize PARP inhibitors for synthetic lethality, specifically for BRCA1-proficient TNBC patients. This new finding could lead to an innovative strategy to overcome the current therapeutic challenge of PARP inhibitors that are only efficient at killing BRCA1-mutant TNBC cells due to HR deficiency but do not have the same effect on BRCA1-proficient TNBC patients (Johnson *et al*, 2011; Yazinski *et al*, 2017).

To validate the therapeutic relevance of KLF4 in breast cancer DNA repair/DNA damage targeting treatment, we have conducted an *in vivo* mouse xenograft study by using murine 4T1 model, which harbors wild-type BRCA1 expression (Castle *et al*, 2014). To this end, we have engineered 4T1 cells with stable knockdown of KLF4 for the xenograft study. To evaluate the impact of KLF4 in sensitizing the clinical efficacy for olaparib, we have administrated olaparib to mice with wild-type 4T1 or 4T1-KLF4 knockdown xenograft tumors. As shown in Fig 8A–C and Appendix Fig S8A and B, consistent with the observation in the cultured-cell model, KLF4 knockdown significantly decreases the breast tumor progression in comparison with the control group. Importantly, we have observed that depletion of KLF4 significantly boosts the efficacy of olaparib in suppressing 4T1 tumor progression (Fig 8A–C). The IHC staining of KLF4, PARP1, Ki67, p21, Bax, and active caspase-3 further indicates that KLF4 knockdown increases olaparib efficacy *in vivo* by

**Figure 8. Suppression of KLF4 sensitizes triple-negative breast tumor to olaparib.**

A   Depletion of KLF4 significantly enhances the efficacy of olaparib in killing TNBC cells (4T1). 4T1 cells with stable expression of shKLF4 were implanted into mammary fat pad of BALB/c nude mice. Drug treatment was started at 10th day. Placebo or olaparib at the dose of 100 mg/kg was administrated daily for 12 days. Tumor volumes were measured every other day. Tumor volumes were measured every other day. *n* = 9 per group, *P* values were labeled in figure, one-way ANOVA assay.

B, C   The image of 4T1 xenograft tumors (B) and tumors were weighed and summarized (C). *n* = 9 per group, *P* values were labeled in figure, one-way ANOVA assay.

D, E   The heatmap of synergy score distribution between olaparib with ABT-263 (D) and dasatinib (E). The synergy score was calculated based on Bliss model according to a model that permits synergistic interaction between olaparib with ABT-263 and dasatinib.

F–I   The combination effect when combining olaparib with ABT-263 and dasatinib in suppression of MDA-MB-231 xenograft tumor growth. 8-week SCID/Beige female mice were injected 8 × 10$^6$ MDA-MB-231 cells in matrigel (1:1 volume). When tumor reaches to 50 mm$^3$, 100 mg/kg olaparib (dissolve in 2% DMSO + 30% PEG400 + saline) was administrated by oral gavage daily, 30 mg/kg ABT-263 (dissolve in 5% DMSO) was administrated by intraperitoneal injection every other day, and 50 mg/kg olaparib (dissolve in 2% DMSO + 30% PEG400 + saline) or placebo (2% DMSO + 30% PEG400 + saline) was administrated by oral gavage every other day. Mice were treated with single or combination drugs for 4 weeks. Tumor volumes were measured every other day (F). The image of MDA-MB-231 xenograft tumors (G). The tumors were weighed and summarized (H). Placebo (*n* = 7), ABT263 (*n* = 9), dasatinib (*n* = 8), olaparib (*n* = 6), olaparib + ABT263 (*n* = 9), olaparib + dasatinib (*n* = 10); *P* values were labeled in figure, one-way ANOVA assay. (I) Staining of H & E, KLF4, PARP1, p21, and Bax in MDA-MB-231 xenograft tumors. Scale bars, 100 μm. Combination of ABT-263 or dasatinib with olaparib treatment decreases KLF4 and p21 expression, but increases Bax expression in xenograft tumors.

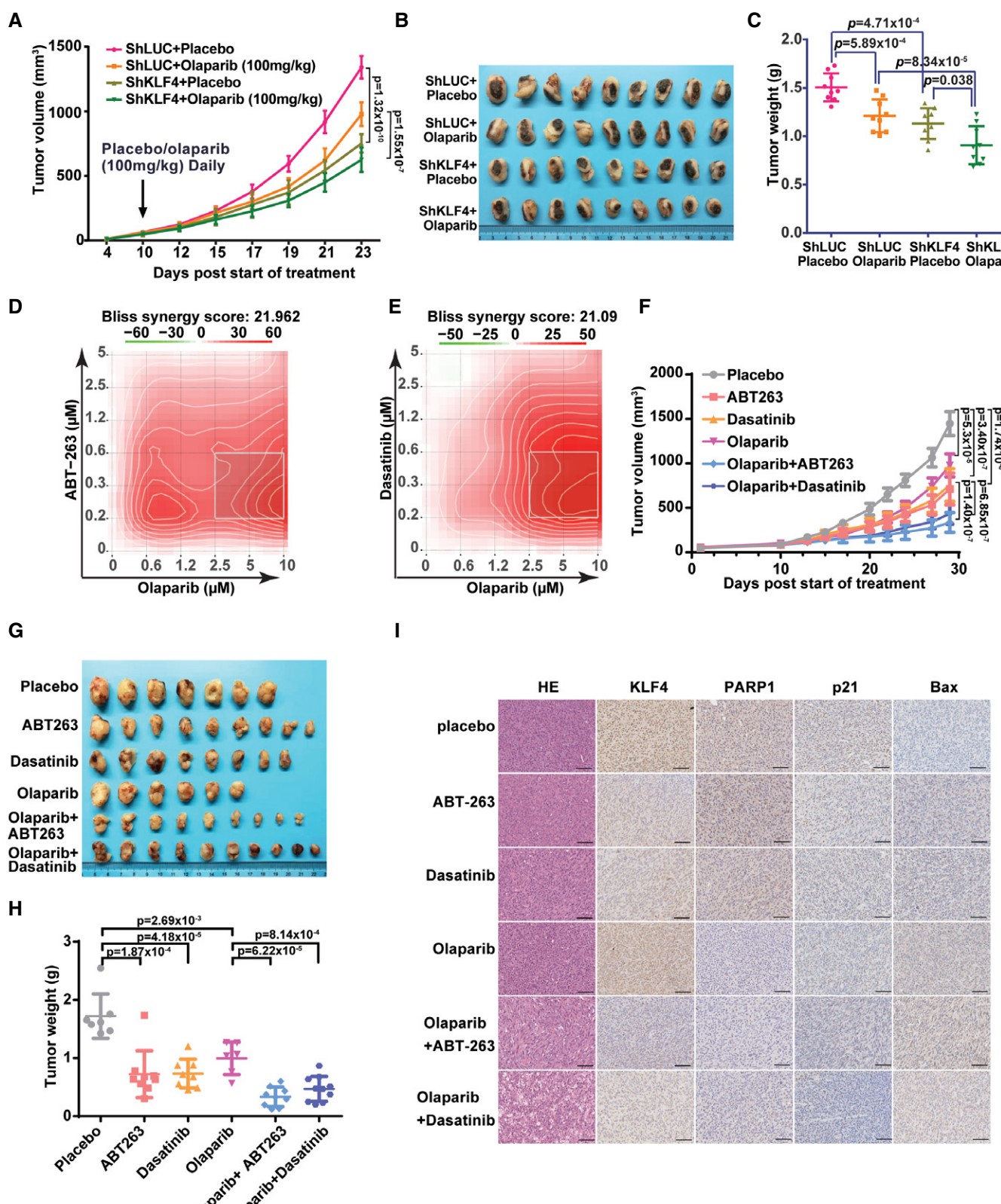

Figure 8.

decreasing cell proliferation and p21 expression and increasing Bax expression and cellular apoptosis (Appendix Fig S8C–E).

To search for drugs that could suppress KLF4 as a potential therapeutic strategy, we have conducted a nonbiased screening of an anti-cancer compound. We engineered stable expression of GFP-KLF4 MDA-MB-231 and evaluated the inhibitory effect for 422 anti-cancer compounds on KLF4 expression levels. As shown in Appendix Fig S8F–H, ABT-263 (BCL2 inhibitor) and dasatinib (Src inhibitor) were scored the highest for compounds in inhibiting KLF4 expression. This result is further confirmed by detecting the abundance of KLF4 protein using Western blotting (Appendix Fig S8H). To test the effect of ABT-263 and dasatinib on enhancing the olaparib-induced cell death, we have conducted cell killing experiments to estimate the combined effect of ABT-263 and dasatinib with olaparib in MDA-MB-231 cells, respectively. As shown in Fig 8D and E, and Appendix Fig S8I and J, both combinations, ABT-263 and olaparib and dasatinib and olaparib, synergistically kill TNBC cells as indicated by the calculation using Bliss-independence model (Sackton et al, 2014; Ianevski et al, 2017).

To further validate the result based on the 4T1 model (Fig 8A–C) and determine the clinical intervention of ABT-263 and olaparib as well as dasatinib and olaparib in anti-TNBC therapy, we have examined the synergistic effect of inactivation of KLF4 by ABT-263 and dasatinib with olaparib using an MBA-MD-231 human TNBC xenograft model. Mice were treated with single drugs or a combination drugs for 4 weeks. Tumor volumes were measured every other day. As shown in Fig 8F–H, we observed treatment with ABT-263, dasatinib, and olaparib individually leads to suppression of tumor progression, in comparison with the placebo group, while a combination of ABT-263 and olaparib or dasatinib and olaparib synergistically inhibited the human TNBC tumor growth (ABT-263 and olaparib, combination index is 1.42; dasatinib and olaparib, combination index is 1.38). The IHC staining of KLF4, PARP1, p21, and Bax further indicates that both ABT-263 and dasatinib increase olaparib efficacy *in vivo* by decreasing KLF4 and p21 expression and increasing Bax expression (Fig 8I). In summary, the results from our preclinical model confirmed the potential clinical value for synergizing KLF4 and PARP1 in TNBC breast cancer therapy. Inhibition of KLF4 could efficiently block both DNA damage response and HR in BRCA1-proficient, TNBC tumors that, in turn, generates synthetic lethality for BRCA1-proficient TNBC tumor cells in combination with blocking PARP function.

# Discussion

### A novel insight into PARylation of KLF4 by PARP1 in regulating genome stability and carcinogenesis

Results from recent TCGA and pathophysiological studies have unveiled the critical role of KLF4 in breast carcinogenesis and drug resistance, although the underlying mechanism remains unclear (Fletcher et al, 2011; Li et al, 2013). We have demonstrated that KLF4, as a cellular fate decision factor in response to genotoxic stress and oncogenic signaling, is tightly regulated by sophisticated posttranslational modifications such as ubiquitylation and protein methylation (Foster et al, 1999; Foster et al, 2000; Pandya et al, 2004; Foster et al, 2005; Rowland et al, 2005; Dong et al, 2014). The

interplay between VHL-mediated ubiquitylation and PRMT5-mediated methylation determines KLF4 protein stability (Hu et al, 2015).

Here, we report our finding that KLF4 is PARylated by PARP1 in response to a DNA damage signal. Our molecular characterization reveals the novel mechanism that PARylation of KLF4 orchestrates KLF4-mediated genome stability and carcinogenesis. We have investigated and described in-depth the mechanism that determines the recruitment of KLF4 from the nucleus to chromatin, ensuring the transcription of KLF4-governed downstream genes, such as p21 and Bax: KLF4 PARylation by PARP1 on the YYR motif near the carboxyterminal region of KLF4 is what determines the recruitment. Previous studies have linked the impact of protein modification by PARylation to various cellular processes including RNA-processing. Recruitment of KLF4 from soluble nucleus to chromatin and regulation of subcellular compartmentation of RNA-processing proteins FUS share similarity with regard to protein modification by PARylation. While it is clear that localization of FUS is through formation of paraspeckles via liquid–liquid phase separation with accumulated PARylated proteins, damaged DNA, PARP-1, and mRNA (Singatulina et al, 2019), whether PAR:KLF4 also undergoes self-assembly by phase separation like RNA-processing proteins FUS remains unclear. Given the importance of p21 and Bax in determining cellular fate, cell cycle arrest for tumor cell survival or onset of apoptosis for tumor death, demonstration of biochemical consequences on KLF4 PARylation by PARP1 in dictating KLF4-mediated transcriptional regulation of p21 and Bax uncovers a new aspect of the molecular basis of DNA damage response (Fig 9). Our biochemical analyses further demonstrate a previously undocumented pivotal role for KLF4 in modulating HR through regulating BRCA1 transcription. With comprehensive analyses using mouse genetic knockout, tissue arrays based on breast cancer patient specimens, and a xenograft breast tumor model, our work demonstrates the synergism between KLF4 and PARP1 in tumorigenesis with respect to cancer therapy for the first time, thus providing a new therapeutic strategy to kill BRCA1-proficient TNBC tumors by synthetic lethality through suppressing KLF4 in combination with blocking PARP1 function (Fig 9).

### KLF4 function is determined by multiple posttranslational modifications

KLF4 is a versatile transcription factor regulating a variety of biological processes such as cell division, apoptosis, DNA damage, and stem cell renewal (Tetreault et al, 2013). KLF4 function is tightly regulated by multiple types of posttranslational modifications. It has been previously reported that KLF4 expression and function are subjected to acetylation and sumoylation (27, 28). Addition of Sumo1 to KLF4 and acetylation of KLF4 by p300/CBP is thought to be necessary to facilitate KLF4-mediated transactivation (27, 28). In addition to acetylation and sumoylation, our study of KLF4 regulation in response to DNA damage signals, using purification of protein complex coupled with mass spectrometry, led to identification of ubiquitylation and arginine protein methylation of KLF4 as additional regulatory mechanisms. While ubiquitylation of KLF4 by VHL/VBC governs KLF4 protein degradation, addition of a methylgroup to KLF4 protein catalyzed by PRMT5 antagonizes KLF4 ubiquitylation through disassociation of binding between VHL and KLF4, thereby stabilizing KLF4 (Hu et al, 2015).

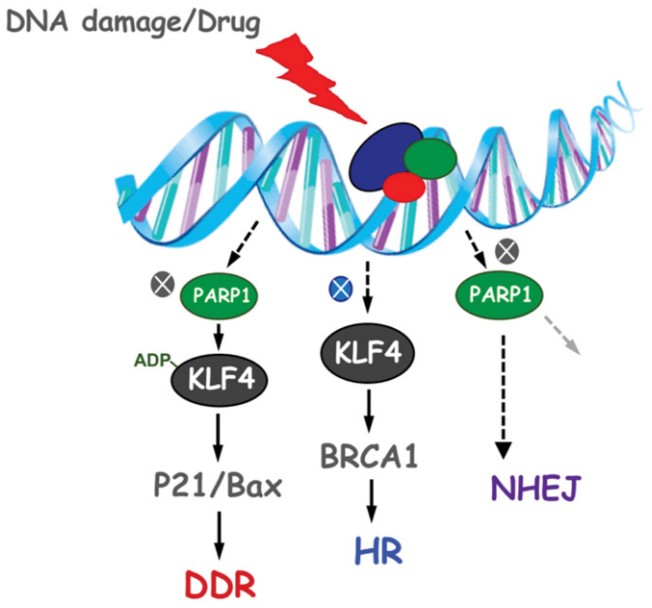

**Figure 9.  Diagram of proposed working model**.

Our current work in studying KLF4 cytosolic trafficking, cytosol/nucleus translocation, and regulation of KLF4 from the soluble nucleus to chromatin further explores a previously unpublished posttranslational modification: PARylation in governing KLF4 function. Here, we have demonstrated that addition of a poly ADP-ribosylation chain to KLF4 generates the recruitment signal for KLF4, allowing KLF4 to move from the nucleus to chromatin that in turn ensures KLF4's transcriptional function. This work not only unveils a new layer of regulatory mechanism for KLF4 by posttranslational modification but also, for the first time, uncovers the mystery of how KLF4 is recruited to chromatin in response to DNA damage signal, which enhances our understanding of the molecular control for DNA damage response.

#### PARylation of KLF4 is required for KLF4-mediated regulation of p21 or Bax in DNA damage response but not necessary for KLF4-mediated regulation of BRCA1-HR

As a transcriptional factor, the role of KLF4 has been linked to a variety of target genes (Rowland & Peeper, 2006). It has been previously reported that multiple physiological roles for KLF4 could be facilitated through targeting different sets of transcriptional networks (Rowland *et al*, 2005; Rowland & Peeper, 2006; Tetreault *et al*, 2013). One hypothesis for multiple types of posttranslational modifications in regulating KLF4 is that different types of posttranslational modifications are required for various signaling that determines the individual physiological destination. Under certain circumstances, specific signals modulate KLF4 according to certain posttranslational modification by different protein modifiers such as E3 ligase, protein methyltransferase, acetyltransferase, and poly (ADP-ribose) polymerase 1. Different types of modifications ensure KLF4 binding to different promoter regions that confer specificity for KLF4 in governing various signal-mediated transcription.

In our study, we observed regulation of KLF4 function in genome stability is sophisticated. We showed that DNA damage-induced PARylation of KLF4 promotes KLF4 chromatin recruitment that, in turn, enhances KLF4 binding to p21 and Bax promoter, resulting in cell cycle arrest as part of a DNA damage response. We also demonstrated that KLF4 directly binds to BRCA1 promoter and executes BRCA1 transcription. Our observations suggest that regulation of KLF4 binding to specific promoters is complex. Posttranslational modifications of KLF4 play a critical role in directing KLF4 with respect to its specificity to bind to various promoter DNA motifs. This notion is supported by recent reports that protein methylation plays an important role in rendering KLF4 to specific DNA motifs (Hu *et al*, 2013; Liu *et al*, 2014). Given the documented transcriptional role of KLF4 being involved in a variety of biological processes, we speculate there could be numerous KLF4-associated transcriptional DNA binding motifs that need to be identified in the near future (Shields & Yang, 1998; Ramsahoye *et al*, 2000). Posttranslational modifications of KLF4 could be the critical determining mechanism in recruiting KLF4 to specific gene promoter DNA motifs. In this work, we have demonstrated, while non-PARylated KLF4 binds to BRCA1 promoter region, DNA damage signal-induced KLF4 PARylation confers its specific binding to the upstream DNA binding motifs of various KLF4-mediated DNA damage responsive genes such as p21 and Bax.

#### Clinical implication for synergism between KLF4 and PARP1 in breast cancer therapy

Although approximately 75% of breast cancer patients with expression of estrogen receptor (ER+) can be treated by endocrine therapy, there are about 15–20% of breast cancer patients with TNBC cells that do not express the above markers, whose treatment options are limited. Hence, the identification of new molecular targeting strategies could benefit these patients. Development of PARP inhibitors (olaparib, rucaparib, niraparib, and talazoparib) provides a promising new strategy to target homologous repair-deficient TNBC breast cancer due to its dysfunctional BRCA1 status (McCann & Hurvitz, 2018; Sulai & Tan, 2018). The diagnosed population for BRCA1 mutation rate in TNBC breast cancer, including both germline and somatic mutation, is about 15–20%, which means approximately 80% of BRCA1-proficient TNBC breast cancer patients could not benefit from PARP inhibitors treatment (Hartman *et al*, 2012; Greenup *et al*, 2013; Johnson *et al*, 2016). While extensive efforts have been made on the translational study of PARP inhibitors in BRCAness patient, the non-BRCA1/2 mutants such as RAD51, BRIP1, PALB2, and FANCA are slow relative low (Johnson *et al*, 2011; Min *et al*, 2015). Thus, searching for a new strategy to generate synthetic lethality for the 80% of BRCA1-proficient TNBC breast cancer patients is urgent.

In this work, we have demonstrated that KLF4 is a pivotal causal factor for breast tumor initiation and invasion in both cultured cells and animal models; its abnormal accumulation tightly correlates with poor breast cancer prognosis. In addition, we observed the elevation of KLF4 levels antagonizes killing efficacy for endocrine therapeutic agents as well as various chemotherapy agents in breast cancer treatment (Zhou *et al*, 2020a). Our new findings highly suggest that KLF4 is a good target

for breast cancer treatment. Our mechanistic studies further revealed that the KLF4 transcriptional function is regulated by several types of posttranslational modifications, including PARP1 in response to DNA damage signal, which determines tumor cell survival in the presence of genotoxic stress. We have demonstrated that KLF4 not only governs DNA damage response through modification by PARP1 but also regulates BRCA1-mediated HR in a PARylation-independent manner. In TNBC cultured-cell model, depletion of KLF4 or interference of KLF4 PARylation by expression of a KLF4-PARylation-resistant mutant largely sensitizes TNBC cells to PARP inhibitors. In the 4T1 breast cancer mouse model, suppression of KLF4 expression significantly sensitizes breast cancer tumors to olaparib, clearly demonstrating the clinical relevance of KLF4 in synergizing PARP1 in anti-TNBC breast cancer treatment. In general, our new findings regarding PARP1-dependent and PARP1-independent KLF4 functions in regulating DNA damage response and DNA repair provide an in-depth understanding of the mechanism by which KLF4 governs genome stability. Results of our preclinical study on synergism between KLF4 and PARP1 have shed light on a novel therapeutic strategy for the BRCA1-proficient population of TNBC breast cancer patients.

# Materials and Methods

### Cell lines and cell culture

HEK293T, MCF10A, MCF12A, MDA-MB-231, MDA-MB-468, MDA-MB-361, HCC1937, MCF-7, T47D, SKBR3, and BT474 cells were obtained from the American Type Culture Collection (Manassas, VA). The KLF4$^{+/+}$ MEF and KLF4$^{-/-}$ MEF were gifts from Dr. Engda Hagos (Colgate University). The BTR cells were provided by Dr Daniel S. Peeper (Netherlands Cancer Institute). The p53$^{+/+}$ and p53$^{-/-}$ HCT116 cells were provided by Lin Zhang (University of Pittsburgh). The retroviral packaging line Phoenix-A cells were the gift from Edward V. Prochownik (University of Pittsburgh). U2OS-EJ5-GFP cells and U2Os-DR-GFP cells (for HR) were provided by Lan Li (University of Pittsburgh). All cell lines were Mycoplasma tested every 3 months using MycoProbe Mycoplasma Detection Kit (R&D Systems). The length of time between cell line thawing and use in experiment did not exceed 1 month (two or more passages). KLF4$^{+/+}$, KLF4$^{-/-}$, HEK293T, Phoenix-A, and breast cancer cell lines were maintained in DMEM supplemented with 5% or 10% FBS, 1 × antibiotic/antimycotic solution (100 units/ml streptomycin and 100 units/ml penicillin) (all from Invitrogen). MCF10A and MCF12A were cultured in Dulbecco's modified Eagle's medium and Ham's F12 medium with 5% horse serum, 20 ng/ml human epidermal growth factor, 100 ng/ml cholera toxin, 0.01 mg/ml bovine insulin, and 500 ng/ml hydrocortisone. All cells were cultured at 37°C in a humidified atmosphere containing 5% CO$_2$.

### Plasmids and transfection

The full-length or partial coding PARP1 plasmids were engineered by the Michael O. Hottiger laboratory. KLF4 and PARP1 constructs were generated by PCR amplification of the full-length or partial coding sequence of human KLF4 and PARP1 and subsequent subcloning into mammalian expression vectors with FLAG-HA or

HA tag. PARP1 and KLF4 point mutations were generated by site-directed mutagenesis. pRetroSuper-KLF4 shRNA was a gift from Dr. Daniel S. Peeper (Netherlands Cancer Institute). The lentivirus plasmid pLCN DSB Repair Reporter (DRR), pimEJ5GFP, pLenti CMV Puro DEST (w118-1), and pLenti CMV/tight Puro DEST (w768-1) were purchased from Addgene. Mouse KLF4 shRNA plasmids (TRCN0000238250 and TRCN0000095370) were purchased from Sigma. shRNA oligos encoding target mouse sequences against mKlf4 (5′-GACATCGCCGGTTTATATTGA-3′) were constructed by pLKo.1 (Addgene). I-SceI plasmid was provided by Lan Li (University of Pittsburgh). PARP1 shRNA plasmids (pLV.PARP1#1, pLV.PARP1#2) were purchased from Addgene. The p21 promoter plasmid (p21-Luc/WWP-Luc) was purchased from Addgene. Human BRCA1 promoter −1,500~+100 (BRCA1-Luc) was amplified by using primer 5′-ATCGGTACCGCATTCTGAACCACAGACTCT-3′ and 5′-ACTAGATCTACCTCATGACCAGCCGACGTT-3′ and then subcloned into pGL3-Luc plasmids (Promega). Point mutation of BRCA1 promoter (BRCA1-AA-Luc) was generated by primer 5′-CTGGA-GACCTCCATTAGAACGGAAAGAGTGGGGGATG-3′ and 5′-ATCCCC-CACTCTTTCCGTTCTAATGGAGGTCTCCAG-3′. For transfection, cells were plated to form a 50–70% confluent culture and transfected using Lipofectamine 2000 (Invitrogen). The siRNA sequence target BRCA1 is BRCA1-S 5′-GGAACCUGUCUCCACAAAG-3′ and BRCA1-AS 5′-CUUUGUGGAGACAGGUUCC-3′; the control siRNA sequence is Luc-S 5′-CGUACGCGGAAUACUUCGA-3′ and Luc-A S 5′-UCGAAGUAUUCCGCGUACG-3′. siRNAs were transfected using Lipofectamine 2000 (Invitrogen).

### Antibodies and chemicals

Specific antibodies against KLF4 (D1F2, 1:1,000), PARP (46D11, 1:1,000), Cleaved PARP (Asp214) (D64E10, 1:1,000), FLAG (D6W5B, 1:1,000), Phospho-Histone H2A.X (Ser139, 1:1,000), 53BP1 (P550, 1:1,000), and BRCA1 (A8X9F, 1:1,000) were purchased from cell signaling (Beverly, MA). KLF4 (H-180, 1:1,000), KLF4 (F-8, 1:1,000), PARP1 (H-300, 1:1,000), HA (F-7, 1:1,000), and Myc tag (9E10, 1:1,000) were from Santa Cruz Biotechnology, Inc. (Santa Cruz, CA). Antibodies against β-actin (AC-15, 1:5,000) and FLAG (M2, 1:2,000) were from Sigma-Aldrich. Antibodies against PAR were purchased from Trevigen (4336-BPC-100, 1:1,000). The anti-FLAG M2 affinity gel was from Sigma-Aldrich. The PARP1 inhibitor niraparib, olaparib, and rucaparib were purchased from Selleckchem (Houston, TA). Puromycin and blasticidin were from Invitrogen. Cycloheximide was from Sigma. The anti-cancer compound library was purchased from Selleckchem. The compounds ABT-263 and dasatinib used in animal model were purchased from Selleckchem.

### Western blotting and immunoprecipitation assay

Cells were harvested and lysed in radioimmune precipitation assay lysis buffer (25 mM Tris pH 7.6, 150 mM NaCl, 1% Triton X-100, 0.5% sodium deoxycholate, and 0.1% SDS) containing protease inhibitor mixture (Sigma) or 1× SDS loading buffer (50 mM Tris–HCl pH 6.8, 2% SDS, 10% glycerol, 12.5 mM EDTA, and 0.02% bromophenol blue). The protein concentration was determined using Bio-Rad protein assay reagent. Western blotting was performed using antibodies against PARP1, KLF4, BRCA1,

and HRP-conjugated goat anti-mouse or antirabbit secondary antibody (Promega). Signals were detected with ECL reagents (Bio-Rad). Semi-quantification of data was performed using NIH Image. For immunoprecipitation assay, cell lysate was incubated with anti-FLAG M2 gel (Sigma), anti-PARP1 (Santa Cruz), or anti-KLF4 (Santa Cruz) antibody overnight at 4°C on a rotator, followed by the addition of protein A/G plus agarose (Pierce) to the reaction for 2 h at 4°C. After five washes with radioimmune precipitation assay lysis buffer supplemented with protease inhibitor mixture, complexes were released from the anti-FLAG M2 gel and protein A/G plus agarose by boiling for 5 min in 2× SDS–PAGE loading buffer.

### Purification of KLF4 complex and mass spectrometry

U2OS cells stably expression FLAG/HA-tagged KLF4 were exposed to 5Gy γ-radiation. 4 h after radiation, U2OS cells were washed twice with PBS and lysed with NP-40 buffer (1% NP40, 10% glycerol, 25 mM Tris–HCl [pH 7.9], and protease inhibitor cocktails). KLF4-interacting proteins were purified by immunopurification followed by washing four times with TBST buffer (137 mM NaCl, 20 mM Tris–HCl [pH 7.6], 0.1% Tween-20). The complex was eluted with 3 × FLAG peptide in TBS buffer. The elute was then separated on SDS–PAGE followed by Coomassie blue staining. The interest bands were cut for mass spectrum analysis (Zhang *et al*, 2010; Gamper *et al*, 2012; Zhou *et al*, 2017b).

### Lentiviral and retroviral infection

The lentivirus plasmids including pLenti-KLF4$^{WT}$ and pLenti-KLF4$^{YYR-AAA}$, pLenti-KLF4$^{C403A}$ and pLenti-KLF4$^{\Delta Zinc2}$ were co-transfected with pVSV-G, pRRE, and pRSV-REV into HEK293T. The retrovirus was packaged in Phoenix-A cells. Lipofectamine 2000 was used for transfection. The packaged lentiviral or retroviral particles were collected, mixed with polybrene, and then added into target cells. The stable cell lines were established by culturing cells in the medium containing antibiotic blasticidin (10 μg/ml) or puromycin (2 μg/ml).

### AAV serotype screening

For AAV serotype screening, $2 \times 10^{11}$ seven types of AAV-GFP particles (AAV1, 2, 5, 6, 7, 8, and 9) were administrated into the C57BL/6J (6–8 weeks) via tail vein or intraperitoneal injection. 3 or 6 weeks later, the expressions of AAV-GFP in the tissue including duodenum, small and large intestine, esophageal, stomach, heart, liver, lung, and kidney were detected by GFP bioluminescent Imaging and Western blot evaluation.

### Generation of AAV7-Cre/KLF4$^{loxp/loxp}$ mouse model

The KLF4$^{loxp/loxp}$ mice (C57BL/6 mice background) were purchased from the Mouse Mutant Regional Resources Centers (MMRRC), and this mice string was previously described (Katz *et al*, 2002). Mice were bred and housed in an AAALAC-accredited barrier facility for specific pathogen-free (SPF) mice. The KLF4$^{loxp/loxp}$ mice at the age 6–8 weeks were intraperitoneally administrated $2 \times 10^{11}$ particles of AAV7-Cre-mCherry or AAV7-mCherry. Five

weeks after AAV administration, the intestine of mice was removed and tissue sectioned for validating the depletion of KLF4 expression.

### γ-irradiation procedure

Mice were exposed to total-body γ-irradiation with a $^{137}$Cs source, with a dose rate of 0.56 Gy/min, for a total of 8 Gy. Another group of mice (sham) were placed in the room without being exposed to irradiation. Animals were either observed for survival postirradiation or were killed by $CO_2$ asphyxiation followed by cervical dislocation at set times after irradiation, and the small intestine was removed for further analysis. For the survival experiment, we used the moribund state as the experimental endpoint, defined as an animal that lost more than 15% of its body weight and was unresponsive and immobile. For this purpose, the animals were monitored daily for body posture, eye appearance, and activity level. Animals reaching the moribund state were killed as mentioned above.

### Immunofluorescence and immunohistochemistry

For immunostaining of paraffin-fixed tissue, sections were deparaffinized in xylene, rehydrated in ethanol gradient, and then recovered by 10 mM sodium citrate, pH 6.0 (Sigma), at 120°C for 10 min in a pressure cooker. For immunofluorescence staining, the histological sections were incubated with blocking buffer contains 3% BSA for 1 h at room temperature. Sections were then stained using goat anti-KLF4 (1:200; Santa Cruz), rabbit anti-cleaved caspase-3 (1:500; Cell signaling), rabbit anti-53BP1 (1:100; Cell signaling), rabbit monoclonal anti p21, rabbit anti-Ki67 (1:500; BioCare Medical), and rabbit anti-phosphorylated histone H2AX (γH2AX) (1:100; Cell Signaling) at 4°C overnight. Washes were done using TTBS, and detection of primary antibodies for immunofluorescence was carried out using Texas-Red (Molecular Probes) at 1:150 dilutions in 3% BSA in TTBS for 30 min at 37°C, counterstained with Hoechst 33258 (2 μg/ml), mounted with Prolong gold (Molecular Probes), and cover-slipped. To analyze the *in situ* interaction between endogenous PARP1 and KLF4, the *in situ* proximity ligation assay (PLA) has been down as previous report (Song *et al*, 2020; Zhou *et al*, 2020b). The PARP (46D11, 1:100) rabbit antibody and KLF4 mouse antibody (F-8, 1:50) were used to staining following the manufacture's protocol. Tissue microarrays (10 mm tissue cores for each tissue) were constructed. For immunostaining of culture cells, MDA-MB-231 cells were planted to coverslip and culture for 24 h. Then, cells were treated with 5Gy γ-irradiation or sham for 4 h and fixed in 4% paraformaldehyde. Cells were blocked with 3% BSA in PBS and stained with goat anti-KLF4 (1:100; Santa Cruz), PARP1 rabbit anti-PARP1 (1:100; Santa Cruz), rabbit anti-PAR (1;100, Trevigen), and mouse anti-FLAG (1;100, Sigma) at 4°C overnight. Washes were done using TTBS, and detection of primary antibodies for immunofluorescence was carried out using Alexa488, Texas-red (Molecular Probes) at 1:150 dilutions in 3% BSA in TTBS for 30 min at 37°C, counterstained with Hoechst 33258 (2 μg/ml), and cover-slipped. IHC staining was carried out following standard streptavidin–biotin–peroxidase complex method. Briefly, section was deparaffinized, and nonspecific bindings were blocked with 10% normal goat serum for 30 min. Section was then incubated

with antibody overnight at 4°C. For negative controls, the primary antibody was replaced by non-immune serum. After immunostaining, sections were scanned, and imaged by a single investigator who was not informed of the clinical characteristics. The value of the integral intensity was measured by Aperio's ImageScope software (Vista, CA).

## Soft agar colony formation assays

The tumorigenicity of KLF4 was measured by soft agar colony formation assays in duplicate in three independent experiments. Briefly, 1-ml underlayers of 0.6% agar medium were prepared in 35-mm dishes by combining equal volumes of 1.2% noble agar and 2× DMEM with 40% fetal bovine serum (Difco). The cells were trypsinized, centrifuged, and resuspended, and $2 \times 10^3$ MDA-MB-231, $2 \times 10^3$ U2Os, $1 \times 104$ MCF10A cells were plated in 0.3% agar medium. 1-ml top layers of 0.6% agar medium were prepared and add. The surface was kept wet by addition of a small amount of growth medium. After 3 weeks, dishes were stained with 0.005% crystal violet and colonies were photographed and counted.

## Clonogenic assay

Cells were trypsinized and plated for 24 h, and then culture medium was replaced with either complete medium (for non-treated controls) or complete medium containing olaparib, niraparib, or rucaparib for 3 days, or doxorubicin or cisplatin for 1 h. Cells were then washed once in PBS and replaced with fresh medium. After an additional 7–10 days of culture, cells were fixed with an acetic acid/methanol (1:3) solution and stained with a dilute crystal violet (0.33%, $w/v$) solution, and surviving colonies consisting of 50 or more cells were counted.

## Cell viability assay

Cells were trypsinized and plated for 24 h, and then treated with ABT-263, dasatinib, or olaparib for 3 days. The viable cells were detected by using a Cell Counting Kit-8 (CCK-8) (Dojindo). The combination effect of ABT-263, dasatinib, or olaparib was calculated based on online SynergyFinder package software (https://synergyfinder.fimm.fi) (Ianevski et al, 2017). The Bliss, Loewe, highest single agent (HSA), and zero interaction potency (ZIP) scores were calculated without baseline correction and using default parameters with the exception that Emin was specified as 0 and Emax as 100.

## Modeling PARP1 and KLF4 structure

For the modeling of KLF4 PARylation, the crystal structure of KLF4 (2WBS) was utilized. For the similarity for the PBZ motif, the crystal structure of APLF (2KQE) was used. The crystal structure of PARP1 (4OQB) was used to stimulate PARP1-KLF4 interaction. The software ClusPro3uses uses a fully automated algorithm to model protein–protein interactions by performing PARP-KLF4 docking simulations. The algorithm evaluates millions of putative complexes and first selects those with favorable surface complementarities. The resulting complexes are filtered based on good

electrostatic interactions and desolvation energies for further clustering.

## Tissue specimens

Informed consent was obtained from all subjects, and the experiments conformed to the principles set out in the WMA Declaration of Helsinki and the Department of Health and Human Services Belmont Report. This study was approved by the ethical committee of the Cancer Institute & Hospital, Chinese Academy of Medical Sciences, and informed consent was obtained from each patient. For this study, two batch tissue arrays have been used for immunohistochemistry staining. The first batch of tissue array, 183 breast invasive ductal carcinoma, and 10 pairs of primary cancer and adjacent normal tissue specimens were analyzed. The 10 pairs of tissue were obtained from patients who were treated with surgical resection alone in 2014 at the Cancer Hospital Chinese Academy of the Medical Science. None of the patients had received radiotherapy or chemotherapy before surgery. The specimens were immediately fixed in 4% polyformaldehyde and completely embedded in paraffin. Clinical characteristics of patients are summarized in Appendix Table S2. The second batch tissue array XT16-054 which contains 117 invasive ductal breast cancer and clinical follow-up to assess prognosis outcome was purchased from Shanghai Outdo Biotech Company. Clinical characteristics of patients are summarized in Appendix Table S3.

## Mice experiment

All animal procedures were reviewed and approved by the Institutional Animal Care and Use Committee of the Chinese Academy of Medical Sciences Cancer Hospital. For 4T1 mouse model, $1.5 \times 10^5$ cells 4T1 cells harbor ShLUC or ShKLF4 in PBS were injected into the 6-week female BALB/c nude mice (Charles River) mammary fat pad. When tumors are palpable, tumor growth was measured for 3 weeks and calculated as $0.5 \times L \times W^2$ (L = Length, W = Width). The observer was blinded to which animal was being measured. 10 days after 4T1 injection, mice were randomized between treatment groups, and 100 mg/kg olaparib or placebo (2% DMSO + 30% PEG400 + saline) was orally administrated daily for 2 weeks. The mice were sacrificed at the end of olaparib or placebo (2% DMSO + 30% PEG400 + saline) treatment, and the xenograft tumors were collected and formalin-fixed, paraffin-embedded, and sectioned. For MDA-MB-231 mouse model, 8-week SCID/Beige female mice (Charles River) were injected $8 \times 10^6$ MDA-MB-231 cells in matrigel (1:1 volume) at mammary fat pad. When tumors reach to 50 mm³, mice were randomized between treatment groups, 100 mg/kg olaparib (dissolve in 2% DMSO + 30% PEG400 + saline) was administrated by oral gavage daily, 30 mg/kg ABT-263 (dissolve in 5% DMSO) was administrated by intraperitoneal injection every other day, and 50mg/kg olaparib (dissolve in 2% DMSO + 30% PEG400 + saline) or placebo (2% DMSO + 30% PEG400 + saline) was administrated by oral gavage every other day. Mice were treated with single or combination drugs for 4 weeks. The mice were sacrificed at the end of ABT-263, dasatinib, olaparib, or treatment, and the xenograft tumors were collected and formalin-fixed, paraffin-embedded, and sectioned. The nature of combined effect of ABT263 or dasatinib with olaparib was determined by using the

published methods based on the principles described by Chou and Talalay (Chou & Talalay, 1984; Zhou *et al*, 2004). The expected value of combination effect between treatment 1 and treatment 2 was calculated as [(observed treatment 1 value)/(control value)] × [(observed treatment 2 value)/(control value)] × (control value); and the combination index was calculated as the ratio of (expected value)/(observed value). A ratio of > 1 indicated a synergistic effect, and a ratio of < 1 indicated a less than additive or antagonistic effect (Zhou *et al*, 2004; Mai *et al*, 2007).

## CHIP

To identify BRCA1 binding sites, MDA-MB-231 cells were not treated or treated with 10 μM olaparib 8 h or 5 gy gamma radiation before cross-linking with 1% formaldehyde and subjecting to ChIP. ChIP was performed was by ChromaFlash High-Sensitivity ChIP Kit according to the manufacturer's protocol (EpiGentek Inc., Brooklyn, NY). KLF4 CHIP was performed by using anti-KLF4 polyclonal antibodies (1:100, Santa Cruz Biotechnology, Santa Cruz, CA) as described previously (Loh *et al*, 2006; Chan *et al*, 2009). Primer sequences for BRCA1 promoters are as follows: BRCA1 55Kb forward, 5′-AAAGAGATGGGACTGTAACTGAGAAGGACC-3′ and reverse, 5′-TGTTTATAGGGAGACTGATGAATGGGC-3′; BRCA1 −1 Kb forward, 5′-CGTCGACGCAATCGCCACCA and reverse 5′-CAGCTTCCCGCCCCCTGGGGA-3′; BRCA1 −0.3 Kb, 5′- CGCAACGCATGCTGGAAATA-3′ and reverse 5′- ACGAAGGTCAGAATCGCTACC-3′; BRCA1 −0.4 Kb forward, 5′-TTCCCTCCACCCCCCCAACAATC-3′ and reverse 5′-CCCAATCCCCCACTCTTTCCGCC-3′; BRCA1 −0 Kb forward, 5′-CGACTGCTTTGGACAATAGGTAGCG-3′ and reverse 5′-AGTCTGCCCCCGGATGACGTAA-3′; and BRCA1 −62 Kb forward 5′-GCGGGAGGAAAATGGGTAGTTAGC-3′ and reverse 5′-CCATTTTCCCAGCATCACCAGC-3′. Primers sequences for p21 are as follows: p21 (−150/−4), forward, 5′-GCTGGGCAGCCAGGAGCCTG-3′ and reverse 5′-CTGCTCACACCTCAGCTGGC-3′; p21 (−190/+3) forward 5′-GCTGGCCTGCTGGAACTC-3′ and reverse 5′-GGCAGCTGCTCACACCTC-3′; p21 (−200) forward 5′-CTGGGCTATTCTCTTGTCAC-3′ and reverse 5′-AGGGCTTCACTTCCAGCAAG-3′. Primer sequences for Bax promoters are as follows: Bax (−870) forward 5′-TGGCTCAAGCCTGTAATCTCAGCA-3′; Bax (−870) reverse 5′-ACTGTCCAATGAGCATCTCCCGAT-3′; Bax (−683) forward 5′-ATTCCAGACTGCAGTGAGCCATGA-3′; Bax (−683) reverse 5′-TTTCCCATATCCGGCATATGA-3′.

## RNA isolation and RT–qPCR

RNAs were extracted with TRIzol™ Reagent (Invitrogen), and cDNA was synthesized using a High Capacity RNA-to-cDNA Kit (Applied Biosystems). The real-time PCR was performed with SYBR® Green PCR Master Mix (Applied Biosystems) and normalized to β-actin. Primer sequences are as follows: ACTB, 5′-AAGATCATTGCTCCTCCTGAGC-3′ and 5′-CATACTCCTGCTTGCTGATCCA-3′; BRCA1, 5′-TGAAATCAGTTTGGATTCTGC-3′ and 5′-CATGCAAGTTTGAAACAGAAC-3′; p21, 5′-CCTGTCACTGTCTTGTACCCT-3′ and 5′-GCGTTTGGAGTGGTAGAAATCT-3′.

## Luciferase assay

The Luciferase assay was performed using a Promega kit by following the manufacturer's instructions. Briefly, cells from 60-mm-

diameter dishes were lysed in 250 μl of luciferase lysis buffer containing 1% Triton X-100 as the detergent and incubated at room temperature for 15 min and lysates were transferred to microcentrifuge tubes. After the addition of 100 μl of luciferase assay reagent, luciferase activity on 25 μl of each sample was measured by using a luminometer.

## Electrophoretic mobility shift assay

The electrophoretic mobility shift assay (EMSA) was performed using an invitrogen kit with SYBR Green (E33075; Invitrogen) by following the manufacturer's instructions. Briefly, the promoter DNA from p21-Luc or pLeftyl-Luc was enzymatic cut and purified and then incubated with purified GST fused wild-type or mutated KLF4 for 20 min. Reactions were run on 6% nondenaturing polyacrylamide, stained by using SYBR Green and then scanned by using a Bio-Rad gel imaging system.

## BTR senescence assay

The BTR cells (immortalized MEFs, co-expressing a temperature-sensitive simian virus 40 (SV40) large T mutant and RASV12) were maintained proliferation indefinitely at 32°C but became senescent at 39.5°C owing to both the rapid disappearance of large T antigen and the presence of RASV12. BTR cells with overexpressed wild-type or mutant KLF4 were planted at 32°C and then shifted to the restrictive temperature (39.5°C). Colonies started to appear after 7–10 days. The cells were left and refed on tissue culture plates for 3 weeks, then fixed, and stained with crystal violet.

## Cytogenetic analysis

The MEFs were initially plated in DMEM containing 10% FBS until they reached 60–70% confluency. Cells then were incubated in the presence of 0.1 μg/ml colcemid (Invitrogen, CA) for 4 h to induce metaphase arrest, centrifuged, and resuspended in 75 mM hypotonic potassium chloride for 10 min. Cells were then fixed with freshly prepared methanol: acetic acid (3:1, *v/v*) solution drop wise, whereas the tubes were vortexed at low speed. The cells were collected by low-speed centrifugation for 5 min. The cell suspension was then spread onto glass slides and then air-dried. Slides were aged at 60°C overnight before the addition of 2 μg/ml Hochest 33452. Metaphase spread images were acquired using a BioTek Lionheart FX Automated Microscope at 60×. The numbers of chromosomes in metaphase (*n* = 100 cells) from each genotype were counted and analyzed.

## NHEJ and HR assays

NHEJ and HR assays were previously described. Briefly, to express I-SceI, pCMV-NLS-I-SceI was introduced by transfection, using Lipofectamine 2000 reagent (Invitrogen), into U2OS-EJ5-GFP cells (for NHEJ) or U2Os-DR-GFP cells (for HR) pre-transfected with KLF4 or ShKLF4 for 48 h using Lipofectamine 2000 (Invitrogen). EGFP-positive cells were counted with Cellquest software. For FACS analysis, cells were harvested by trypsinization, washed with PBS, stained, and applied on the FACS caliber apparatus (Becton Dickinson).

## The paper explained

### Problem

Results from recent "The Cancer Genome Atlas" and pathophysiological studies have unveiled the critical role of KLF4 in genomic integrity, breast carcinogenesis, and drug resistance. In response to oncogenic signaling as well as genotoxic stress, KLF4 is posttranslationally modified, such as in ubiquitylation and methylation. However, the posttranslational modifications orchestrate KLF4 in DNA damage response and DNA repair and its underlying mechanism remains unclear.

### Results

We discovered that the function of KLF4 is tightly regulated by poly (ADP-ribosyl)ation (PARylation) for cellular compartment. The PBZ domain YYR motif (Y430, Y451 and R452) on KLF4 that enables PARP1-mediated PARylation, which recruits KLF4 from the nucleus to the chromatin and facilitates KLF4-mediated transcriptional function on p21 and Bax, promotes DNA damage response. KLF4 also modulates homologous recombination through regulating BRCA1 transcription, which is independent of PARylation. Combination of KLF4 inactivation and PARP1 inhibition leads to synthetic lethality that provides a novel strategy to kill BRCA1-proficient triple-negative breast cancer tumors.

### Impact

Our study reveals the PARylation-dependent KLF4 function in DNA damage response and PARylation-independent function in homologous recombination. Finding a functional interaction between KLF4 and PARP1 in DNA damage response and KLF4-BRCA1 in homologous recombination unveils a novel approach to induce synthetic lethality that further leverages PARP inhibitors specifically to benefit BRCA-proficient TNBC patients.

### Ethics statement

This study protocol had been reviewed and approved by the ethical committees of Chinese Academy of Medical Sciences Cancer Hospital. All participants gave written informed consent.

### Statistical analysis

Statistical analysis was performed using two-tailed, two independent, or paired sample t-tests, one-way ANOVA tests depending on the number of groups with SPSS17.0 (SPSS, Chicago, IL, USA).

# Data availability

This study includes no data deposited in external repositories.

Expanded View for this article is available online.

### Acknowledgements

We are grateful to Drs. Wade Harper and Jianping Jin for kindly providing the TAP purification vector. We appreciate Dr. Engda Hagos for providing KLF4$^{-/-}$ MEF cells. We thank Dr. Lan Li for sharing with us the PARP$^{-/-}$ MEF cells as well as HR and NHEJ assay systems. We appreciate the Proteomics Core at the University of Pittsburgh for mass spectrometry analyses. We thank all members of Wan, Liu, and Bahar laboratories for their helpful comments and discussion. This work was supported by the Northwestern University Zell scholar fund, and effort for YW is covered by NIH R01CA250110, R01CA154695, and R01CA202963. Support from NIH grants U54 HG008540 and P41 GM103712 is gratefully acknowledged by IB. Support from National Nature Science Foundation of China (81130043) is acknowledged by ZL.

### Author contributions

YW is the PI who managed the whole project. YW and ZZ designed the experiments and wrote the manuscript. ZZ comprehensively conducted analyses including biochemical, cell biological, mutagenesis, DNA damage, DNA repair, tumorigenicity, and drug treatment assays. FH and RZ conducted mouse genetic and xenograft studies as well as characterization under the supervision from ZL. AL performed human breast cancer tissue array and immunohistochemistry. IS and IB performed the structural modeling and simulations. MH contributed PARP1 analyses. MC supervised IHC and drug efficacy evaluation.

### Conflict of interest

The authors declare that they have no conflict of interest.

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
