## [Review Process File · EMBO Molecular Medicine]

New insight into the significance of KLF4 PARylation in genome stability, carcinogenesis and therapy

Zhuan Zhou, Furong Huang, Indira Shrivastava, Rui Zhu, Aiping Luo, Michael Hottiger, Met Bahar, Zhihua Liu, Maximo Cristofanilli, and Yong Wan

DOI: [10.15252/emmm.202012391](https://doi.org/10.15252/emmm.202012391)

Corresponding authors: Yong Wan (yong.wan@northwestern.edu)

Review Timeline:

Submission Date:	24th Mar 20
Editorial Decision:	6th May 20
Revision Received:	31st Jul 20
Editorial Decision:	23rd Sep 20
Revision Received:	1st Oct 20
Accepted:	19th Oct 20

Editor: Jingyi Hou

Transaction Report:

6th May 2020

Dear Prof. Wan,

Thank you for the submission of your manuscript to EMBO Molecular Medicine. We have now received feedback from the three referees whom we asked to evaluate your manuscript. As you will see from the reports below, the referees acknowledge the potential interest of the study. However, they also raise substantial concerns about your work, which should be convincingly addressed in a major revision of the present manuscript.

In particular, during our cross-commenting process (in which the referees are given the chance to make additional comments, including on each other's reports), all referees agreed that the issue of the use of different (or undefined) cell lines in different experiments **MUST** be rectified. Referee #3 added "I concur about the confusion being able to follow the manuscript and it seems to largely come from the combination of two different aspects of KLF4 - it feels like two papers that have been pushed together. I also agree that there are many places where either the cell line used or there are data from one cell line that is used as a basis for interpreting phenomena in other cell lines. The authors should both make it clear what cell line is being used in each experiment and perform biological replicates (other cell lines)." Referee #2 added "The issue with reviewer #1 drugs combinations needs to be determined if they are synergistic, additive and neither by conducting drug index studies. After reading reviewer #3 's comments on figure 7 and 8, the suggests need to be addressed. Please remove my comments that Fig. 7 and 8 are "ok" and replace with the comments below. I concur with Reviewer #3 comments that KLF4 PARylation is require for p21 but not BRCA1 expression and thus raises a good point, can BRCA1 rescue HR in the absence of KLF4? Additionally, I agree that the authors should investigate the levels of apoptosis and status of p21 in response to the drug treatment by either doing immunoblotting directly from tumor extracts and or by Immunohistochemistry analysis." This referee also thinks that the manuscript needs to be shortened and additional editing would be necessary to improve clarity.

I think that the referees' recommendations are rather clear and there is no need to reiterate their comments. Importantly, the clarity in data/study presentation needs to be improved, more details and information (especially with regards to in which cell type these experiments were done) must be provided, and the referee #3's concerns with regards to cell cycle need to be addressed.

We would welcome the submission of a revised version within three months for further consideration. Please note that EMBO Molecular Medicine strongly supports a single round of revision and that, as acceptance or rejection of the manuscript will depend on another round of review, your responses should be as complete as possible.

We are aware that many laboratories cannot function at full efficiency during the current COVID-19/SARS-CoV-2 pandemic and have therefore extended our "scooping protection policy" to cover

the period required for a full revision to address the experimental issues. Please let me know should you need additional time, and also if you see a paper with related content published elsewhere.

I look forward to receiving your revised manuscript.

Yours sincerely,

Jingyi Hou

Jingyi Hou
Editor
EMBO Molecular Medicine

*** Instructions to submit your revised manuscript ***

** PLEASE NOTE ** As part of the EMBO Publications transparent editorial process initiative (see our Editorial at <https://www.embopress.org/doi/pdf/10.1002/emmm.201000094>), EMBO Molecular Medicine will publish online a Review Process File to accompany accepted manuscripts.

To submit your manuscript, please follow this link:

Link Not Available

- 1) a .doc formatted version of the manuscript text (including Figure legends and tables). Please make sure that the changes are highlighted to be clearly visible to referees and editors alike.
- 2) separate figure files*
- 3) supplemental information as Expanded View and/or Appendix. Please carefully check the authors guidelines for formatting Expanded view and Appendix figures and tables at

<https://www.embopress.org/page/journal/17574684/authorguide#expandedview>

4) a letter INCLUDING the reviewers' reports and your detailed responses to their comments (as Word file)

Also, and to save some time should your paper be accepted, please read below for additional information regarding some features of our research articles:

5) The paper explained: EMBO Molecular Medicine articles are accompanied by a summary of the articles to emphasize the major findings in the paper and their medical implications for the non-specialist reader. Please provide a draft summary of your article highlighting

6) For more information: There is space at the end of each article to list relevant web links for further consultation by our readers. Could you identify some relevant ones and provide such information as well? Some examples are patient associations, relevant databases, OMIM/proteins/genes links, author's websites, etc...

7) Author contributions: the contribution of every author must be detailed in a separate section (before the acknowledgments).

8) EMBO Molecular Medicine now requires a complete author checklist (<https://www.embopress.org/page/journal/17574684/authorguide>) to be submitted with all revised manuscripts. Please use the checklist as a guideline for the sort of information we need WITHIN the manuscript as well as in the checklist. This is particularly important for animal reporting, antibody dilutions (missing) and exact p-values and n that should be indicated instead of a range.

9) Every published paper now includes a 'Synopsis' to further enhance discoverability. Synopses are displayed on the journal webpage and are freely accessible to all readers. They include a short stand first (maximum of 300 characters, including space) as well as 2-5 one sentence bullet points that summarise the paper. Please write the bullet points to summarise the key NEW findings. They should be designed to be complementary to the abstract - i.e. not repeat the same text. We encourage inclusion of key acronyms and quantitative information (maximum of 30 words / bullet point). Please use the passive voice. Please attach these in a separate file or send them by email, we will incorporate them accordingly.

You are also welcome to suggest a striking image or visual abstract to illustrate your article. If you do please provide a jpeg file 550 px-wide x 400-px high.

10) A Conflict of Interest statement should be provided in the main text

11) Please note that we now mandate that all corresponding authors list an ORCID digital identifier. This takes <90 seconds to complete. We encourage all authors to supply an ORCID identifier, which will be linked to their name for unambiguous name identification.

Currently, our records indicate that there is no ORCID associated with your account.

Please click the link below to provide an ORCID:

Link Not Available

12) The system will prompt you to fill in your funding and payment information. This will allow Wiley to send you a quote for the article processing charge (APC) in case of acceptance. This quote takes into account any reduction or fee waivers that you may be eligible for. Authors do not need to pay any fees before their manuscript is accepted and transferred to our publisher.

Photos 400-800 DPI

*Additional important information regarding figures and illustrations can be found at <http://bit.ly/EMBOPressFigurePreparationGuideline>

***** Reviewer's comments *****

Referee #1 (Comments on Novelty/Model System for Author):

In several cases, the cells in which experiments were performed cannot be found anywhere in manuscript. This is of key importance, given that KLF4 has opposing roles in oncogenesis depending on cell type. In Fig 5, and 6A,B, one cannot find anywhere (text or legends) in which cells these experiments were done, very sloppy!

Referee #1 (Remarks for Author):

The authors make the potentially interesting observation that KLF4 PARylation has a role in the DNA damage response in cancer.

KLF4 is known to have opposing roles in oncogenesis depending on the cell type, hence is it crucial to stick to one cell type for experiments to make a point. The authors go back and forth between cell types, often not even indicating which cells were used (Fig. 5, 6A, B). This makes it impossible to evaluate the manuscript at this stage. Please revise indicating exactly in which cell line which experiment was done, in text and in figure legends.

Please help your reader by explaining in results section what was done. For instance, What is inducible with dox in Fig. 7B, what is the BRT model used in Fig. 7C? Without this information, the manuscript is impossible to read.

Technical comments:

Fig 2A, please show Western blot for KLF4 and PARP1 after 5Gy irradiation. This the small increase in binding a consequence of change in abundance?

The different effects of PARylation on p21 expression versus BRCA1 expression is remarkable. Is this also the consequence of differences in chromatin binding between p21 and BRCA1 promoters? In Figure 8, the effects of combining olaparib and KLF4 knockdown are additive at best, please provide synergy calculation.

Dasatinib and ABT-263 appear to inhibit exogenous GFP-KLF4 fusion protein. Was this also tested on endogenous protein? Fig 8F does not specify what the protein was that was blotted.

The concept of BRCA-ness is not mentioned. Is it possible that KLF4 low breast tumors display BRCAness? This would greatly increase translational relevance.

It is remarkable that the two drugs that act on KLF4, dasatinib and ABT-263 are both powerful senolytic drugs, coincidence? May be test quercetin also as this is also a senolytic, together with dasatinib.

Referee #2 (Comments on Novelty/Model System for Author):

The manuscript provides convincing data that at times is very well controlled and at other it needs correction. The novelty of PARylation mediated PARP1 on KLF4 and the mapping of the physical contact were well done. I am concern of the length of the manuscript it appears that two papers were put together so the story is not fluid. It was difficult to read because of was huge amounts of overlapping datasets. I was less convince that KLF4 is predictive of TNBC but rather more general for all breast cancers. Nonetheless a therapy may arise form this for TNBC.

Referee #2 (Remarks for Author):

In this manuscript, Zhou et al provide evidence that Kruppel-like factor 4 (KLF4) is regulated by poly (ADP-ribosyl)ation (PARylation) in response to genotoxic stress. The functional role of KLF4 as a tumor suppressor an or as an oncogene is controversial that is context and tissue dependent. The PIs generate a conditional KLF4^{Flx/Flx} mice and utilized adeno-associated virus (AAV) gene delivery system to intestinal epithelial cells, which carries Cre-recombinase, they conditionally knockout KLF4 in the intestinal epithelia's cells. Upon exposure to 8Gy of radiation the overall survival of mice decrease KLF4 null mice. The increase expression of DNA damage markers was noted in vivo this may suggesting that KLF4 is functioning in survival, upon γ -radiation exposure. In support of the above results, the PI demonstrated that in mouse embryonic cells (MEFs) that have been KO for KLF4 have increased the percentage of cells with chromosome instability and upon radiation exposure, the KLF4 null cells, are associate with markers for homologous recombination (HR) but not with nonhomologous end joining (NHEJ). Suggesting KLF4 plays a role in HR DNA repair but not NHEJ. To gain mechanistic insights, mass spectrometry analysis with tagged-KLF4 pull downs revealed physical interactions between KLF4 and PARP1. The PIs posit that this interaction is increased upon exposure to γ -radiation. They further map the YYR (i.e. Y430, Y451 and R452) motifs on KLF4, that undergo PARylation that is mediated by PARP1 causing KLF4 lead to enter the nucleus and bind to the chromatin. The authors argue that this lack of DNA damage respond that modulates HR is through BRCA1 activity, as KLF4 is shown to regulate BRCA1 expression upon γ -radiation. The authors provide evidence that PARP1 mediate the recruitment chromatin to ensure its transcriptional activity. Furthermore, evidence is provided that PARylated

KLF4 facilitates apoptosis in response to DNA damage and this is reduced in the YYR mutant in triple negative cell lines. The presence of wt KLF4 sensitizes PARP inhibitors in combination with DNA damage drugs in several TNBC cell lines that are either wt for BRCA1 and or mutated for BRCA1 in vitro, this sensitivity is lost in part when a version of the mutated KLF4 YYR is used. The author also able to demonstrates that in vivo the wt KLF4 is able to sensitize PARP inhibitors Olirapab in combination with Dasatinib (a Src inhibitor) and or ABT263 which targets BCL2 members as negative regulators of apoptosis. The manuscript provides extensive amounts of data to argue for the relevance of PARlytaed KLF4 mediate by PARP1 to collaborate in chromosome instability and promoting tumors. As it stands the manuscript is viable for publication, but several major and minor problem need to be address.

General concerns of the text and results description were found to be sparse and often lacking in accurate details that match the text to access the results shown in the figures. Requiring one to go to the legends and at times vital information was missing. This is particularly true for the description of which cell lines where been used for several of the figures. One problem is that so many different models are been interplay intestinalis in vivo, MEFS , osteosarcomas, breast cancer cell lines ER+ vs. TNBC, using one cell line and make conclusion on another cell line model is often used which raise concerns. Additionally, with the vast amount of the data one needed to read several times to ensure accuracy for the conclusion which where been made and at times this was not always easy to follow and or it was a reasonable conclusion that one could agree with. The manuscript needs to be streamlined and shorten for general readers who are not experts in the field and to help with the reading

Major/Minor Concerns.

1. Figure 1 using the AAV model system for intestinal epithelial cells has nothing to do with breast cancer and the cause of death in the mice is not shown, did the mice die from tumors of the intestines? This was not shown or did the mice die form another cause as KLF4 is ubiquitously expressed and AAV model system can transduced most any cell and or tissue organ. It argues that KLF4 protect from radiation death.
2. Published work with the AAV system to generate tumors has been shown at 5Gy this model used 8Gy which is quite toxic to survival of the mice, what happened at 5Gy?
3. The supplemental figure 2 is well control in mefs, was it to argue for the intestinal model? this was not clear other than to state that KLF4 plays an important role in MEFS for chromosome instability and for HR and not NHEJ.
4. Figure 2 is an example were neither the text , figure and or the legends tells us which cell line is been used. U20S, 293T , MEFS or breast cancer?
5. Fig2 L use 10Gy to show for the PAR antibody was up in response to radiation, yet all the preceding figure used 5Gy why? Not explained in the text no rationale for this or which cell line was used.
6. Figure 3 the survival curves for the importance for KLF4 is for breast cancer in general and not TNBC. Some of the figures in the supplemental figure 3 are the same as in the general figure?
7. No p-values are shown for the TMA results Figure S3C-B. No apparent differences between HER2+ and or TNBC and as this is the main argument for your model later in the MS this may be more important as a main figure rather than as a supplement and may argue against published work
8. Published work has shown the in TNBC KLF4 low high expression correlates with better survival while low expression has a poorer survival outcome contradicting the conclusion of the authors.
9. Figure S3A is out of place to show and IP and in what cell line Figure S3I has no loading control

and is out of place.

10. Figure 3 I-J the IF is of very poor quality.

11. Figure 4 no problems very well done and nicely control mapping the YYR motif on KLF4.

12. Figure 5 is another example where the text and the figure legend does not tell us which cell lines that assay are been done I assume 293T?

13. Figure 5B the flag IPs with the mutations YYR is under load compare to wt and or YKH KLF4 that can explain why so see less KLF4Par, similar for mutation of Y451A. This is a critical dataset to argue for the model need corrections.

14. Fig5E IP flag should have a nuclear marker for loading control H3?

15. Figure 5F the literature does not support the loss of ORC2 when cells are exposed to 5Gy on the chromatid, yet this is clear and when KLF4YYR is use it is more pronounce and again no loading control for the chromatid unless ORC2 was supposed to be that control. Similarly Figure S5 D and E no loading controls.

16. Figure 5I a critical figure for mechanism to differentiate PARP1 motif necessary to interact with KLF4 is underloaded i.e. H909A to explain the loss of PAR antibody. Need to be address. Figure 5I control second lane is underloaded in response to 5Gy exposures

17. KLF4 PARYlation is require for p21 but not BRCA1 expression and thus raises a good point, can BRCA1 rescue HR in the absence of KLF4? Additionally, the authors should investigate the levels of apoptosis and status of p21 in response to the drug treatment by either doing immunoblotting directly from tumor extracts and or by Immunohistochemistry analysis.

Referee #3 (Comments on Novelty/Model System for Author):

Overall the manuscript relates novel findings on KLF4 in response to damage, apparently outlining two responses that appear independent of one another. This gets complicated to follow.

Technically there are some controls that are lacking and a lack of consideration of cell cycle and potentially confounding factors such as p53 state. These will need to be addressed before it can be considered for publication.

Referee #3 (Remarks for Author):

Zhou et al present an interesting manuscript describing a role for KLF4 parylation in genome instability and cancer. The work outlines how PARYlation of KLF4 regulates its location. KLF4 is a transcription factor. The authors mapped interacting motifs between KLF4 and PARP1 and showed how disrupting their interaction impacts DNA damage response, particularly BRCA1 promoted homologous recombination events via regulation of BRCA1 transcription, though there seems to be separable functions (PARYlation of KLF4 and transcriptional regulation of BRCA1). Though the overall findings are interesting there are some major concerns and additional aspects of this finding that should be explored/dealt with.

Initial work was to establish a role for KLF4 in DNA damage response in vivo, using conditional KLF4 knockout and exogenous exposure to an adenovirus Cre that expresses in the intestine. Impact of conditional deletion of KLF4 on morphology and sensitivity to 8Gy irradiation was measured. Zhou et al observed increased DNA damage and apoptosis based on morphological changes and IHC of DNA damage markers (H2Ax and 53BP1) and apoptosis. They went on to verify these results in cell culture models measuring chromosomal aberrations, accumulation of damage by 53BP1 and H2Ax foci and then directly measuring HR and NHEJ using cell based assays DR-GFP and EJ5. A defect

was found in HR, but not NHEJ.

To complement these studies, Zhou et al should also evaluate classic DDR signaling responses (ATM-CHEK2-p53).

To better understand KLF4 role in damage repair, Zhou et al looked for protein interactors. PARP1 and KLF4 interact and this interaction increases in response to damage, both in the cytoplasm and in the nucleus.

Other than PARP1 peptides, what other protein interactions were found. This may provide insight into the type of dynamics associated with PARylation of KLF4. Were any of these known PARP1 targets? This can be compared to various proteomic screens such as in PMID: 24055347

Where are the controls for purity of obtained fractions?

Fig2E. Is KLF4 also interacting with cleaved PARP1? How much commitment is there to apoptosis is with these cells at 5 Gy at 4 hrs?

Need controls for the total amount of proteins (KLF4 and PARP1) and how it changes upon 5Gy at 4 hrs. The localization of KLF4 appears to move to more defined focal locations in response to damage, that look similar to PARP1 locations. Are these foci sites of damage or something like parseckles with accumulated PARylated RNA processing proteins? Can the proportion of co-localization be quantified? Perhaps a PLA assay would clarify this IF based assessment.

What are the cell cycle impacts of their various modulations as this is poorly described? Fig6C suggests altered cell cycle response, but this only measures G2/M proportion. Especially important is evaluating G1/S checkpoint and S-phase progression.

The authors went on to evaluate KLF4 and PARP1 levels in TNBC and observed increase levels compared to normal mammary epithelial cells and co-accumulation of these 2 proteins. Higher KLF4 expression was associated with shorter survival.

The authors then performed deletion mapping to identify fragments necessary for KLF4:PARP1 interaction. They went on to identify a YYR motif as a potential PARylation site on KLF4. Using site specific mutant the authors then show the requirement of these sites, without impacting other known KLF4 posttranslational modifications. In response to damage KLF4 accumulates on chromatin in a PARP1 dependent manner, and this accumulation was attenuated by mutation of YYR. This also resulted in reduced KLF4 ChIP at p21 and BAX promoters.

Fig5D, it appears in this figure that in response to Dox, PARP1 and KLF4 protein levels increase at least in chromatin, while KLF4 soluble nuclear fraction increased with PARP1i with 1 μ M Dox, so what are the protein level dynamics of these proteins (endogenous) in response to damage and/or DOX?

Fig 6F, are the cells accumulating in G2/M? Need to see progression with BrdU progression assay. Is this in U2OS cells?

Fig 6G - how is the HR result significant? Perhaps plot the individual datapoints over the bargraphs? And if the cells are arresting in cell cycle, is this the basis of the HR defect considering that the DR-GFP assay relies on amplification of the number of cells through 2+ cell cycles.

Considering the cell cycle changes and that BRCA1 expression is cell cycle regulated, how much of the effect is associated with this and could the changes observed be due to this? Are the gene expression changes reflected at the protein level? BRCA1 protein level also display post translational changes over cell cycle (R. Baer paper) - is this observed or altered. Fig 6L for BRCA1 protein bands appears to be compressed. This should be presented in a gradient gel.

Are the effects, particularly on p21 and Bax expression, p53 dependent? Again - considering this question, what is the cell cycle impact? Is there a change in proportion of cells in G1 in response to damage? And how is KLF4 dependent response altered by p53 status?

Fig 7R. Can this decrease be rescued by co-depleting 53BP1? Please also note typographical error "KLF4/siRBCA1"

Fig7 H-R - typographical error "Vetcor".

With these viability assays there are some confusing results. KLF4wt expression seems to improve in general. Is this BRCA1 dependent (siBRCA1) or can shKLF4 be rescued with si53BP1 co-depletion? HCC1937 is known to be BRCA1 mutant but PARP1i resistant (see for example PMID 25984718) - is the differences in viability altered with wt BRCA1 expression and is it dependent upon p53 status? And most of the tested cell lines have various p53 mutations, do these alter sensitivity and impact of KLF4 status?

Does expression of YYR mutant impact HR?

KLF4 binds to upstream region of BRCA1. This does not require KLF PARylation, while p21 expression depends on PARylation. Expression of BRCA1 rescued HR in absence of KLF4.

Authors then go on to look at clinical relevance and pre-clinical use with 4T1 and MDA231 mouse orthotopic tumor models.

Fig 8 - tumor size appears to be decreased upon combined KLF4 depletion and PARP1i treatment, but still progresses. Can the authors measure levels of apoptosis and p21 in response to treatments? To complement this work 4T1 cell should also be examined in vitro as they seem less responsive in vivo than MDA231 cells.

Overall the variations in different drugs used is confusing and not explained throughout the manuscript. It would also help to update the model figure to capture better the relationships of PARP1 related activities and transactivation activities and how this relates to damage response.

Response Letter**Dear Editor and Reviewers:**

We greatly appreciate your comprehensive evaluation of our manuscript—the compliments as well as the instructive critiques, technical concerns, and mechanistic questions. Our team has made a tremendous effort during this pandemic period to respond to your comments and improve the quality of our manuscript, which now includes 9 main figures (91 panels), 8 supplementary figures (69 panels), and 3 supplementary tables. We performed a series of new experiments to directly address your concerns; our findings have strengthened the conclusions of our manuscript regarding (1) Identification of a novel mechanism by which the chromatin recruitment of Krüppel-like Factor 4 (KLF4), a critical transcription factor governing genome stability and tumorigenesis, is regulated by poly ADP-ribosylation (PARylation); (2) How KLF4 orchestrates genome stability through both PARylation-dependent and -independent mechanisms—PARylated KLF4 facilitates p21- and Bax-mediated DNA damage response, while non-PARylated KLF4 regulates BRCA1-mediated homologous recombination; and (3) The critical role KLF4 plays in synergizing PARP1, the inhibition of which otherwise induces synthetic lethality specifically for BRCA1-proficient triple negative breast cancer tumors. We have also expanded our research to improve rigor and reproducibility, as described in our point-by-point explanations below (in blue text).

Reviewer #1: Pages 3-8

Reviewer #2: Pages 9-22

Reviewer #3: Pages 23-39

To avoid any confusion with the figure numbers, which have changed in the revised version, all new figure numbers in the descriptions below are shown in red text.

Our point-by-point response to the editor and each reviewer's comments are below. To avoid any confusion in the figure numbers, which we have changed in the revised version, all new figure numbers in the description below are colored red.

Response to the editor's comments

Thank you for the submission of your research manuscript to EMBO reports. We have now received the full set of referee reports that are copied below.

Thank you for the submission of your manuscript to EMBO Molecular Medicine. We have now received feedback from the three referees whom we asked to evaluate your manuscript. As you will see from the reports below, the referees acknowledge the potential interest of the study. However, they also raise substantial concerns about your work, which should be convincingly addressed in a major revision of the present manuscript.

In particular, during our cross-commenting process (in which the referees are given the chance to make additional comments, including on each other's reports), all referees agreed that the issue of the use of different (or undefined) cell lines in different experiments MUST be rectified. Referee #3 added "I concur about the confusion being able to follow the manuscript and it seems to largely come from the combination of two different aspects of KLF4 - it feels like two papers that have been pushed together. I also agree that there are many places where either the cell line used or there are data from one cell line that is used as a basis for interpreting phenomena in other cell lines. The authors should both make it clear what cell line is being used in each experiment and perform biological replicates (other cell lines)." Referee #2 added "The issue with reviewer #1 drugs combinations needs to be determined if they are synergistic, additive and neither by conducting drug index studies. After reading reviewer #3 's comments on figure 7 and 8, the suggests need to be addressed. Please remove my comments that Fig. 7 and 8 are "ok" and replace with the comments below. I concur with Reviewer #3 comments that KLF4 PARylation is require for p21 but not BRCA1 expression and thus raises a good point, can BRCA1 rescue HR in the absence of KLF4? Additionally, I agree that the authors should investigate the levels of apoptosis and status of p21 in response to the drug treatment by either doing immunoblotting directly from tumor extracts and or by Immunohistochemistry analysis." This referee also thinks that the manuscript needs to be shortened and additional editing would be necessary to improve clarity.

I think that the referees' recommendations are rather clear and there is no need to reiterate their comments. Importantly, the clarity in data/study presentation needs to be improved, more details and information (especially with regards to in which cell type these experiments were done) must be provided, and the referee #3's concerns with regards to cell cycle need to be addressed.

We would welcome the submission of a revised version within three months for further consideration. Please note that EMBO Molecular Medicine strongly supports a single round of revision and that, as acceptance or rejection of the manuscript will depend on another round of review, your responses should be as complete as possible.

We appreciate the editor's comments and summary! The concerns summarized above are addressed in the point-by-point responses to individual reviewers. In addition, we have carried out a series of new experiments to strengthen our conclusion.

Referee #1 (Comments on Novelty/Model System for Author):

In several cases, the cells in which experiments were performed cannot be found anywhere in manuscript. This is of key importance, given that KLF4 has opposing roles in oncogenesis depending on cell type. In Fig 5, and 6A,B, one cannot find anywhere (text or legends) in which cells these experiments were done, very sloppy!

Thanks to the reviewer for the comments! We have added the detailed description of each cell line being used in all experiments in figure legends and/or main manuscript text, especially in the legends for Figures 5 and 6 A,B and the correlated main text.

Referee #1 (Remarks for Author):

The authors make the potentially interesting observation that KLF4 PARylation has a role in the DNA damage response in cancer. KLF4 is known to have opposing roles in oncogenesis depending on the cell type, hence is it crucial to stick to one cell type for experiments to make a point. The authors go back and forth between cell types, often not even indicating which cells were used (Fig. 5, 6A, B). This makes it impossible to evaluate the manuscript at this stage. Please revise indicating exactly in which cell line which experiment was done, in text and in figure legends.

Thanks to the reviewer for the comments! We made corrections and added clear descriptions about cell lines in the legends for Figures 5 and 6 A,B and the correlated text in the manuscript. In addition, we checked all possible related issue in all figures and text and made any needed additions or corrections.

Please help your reader by explaining in results section what was done. For instance, What is inducible with dox in Fig. 7B, what is the BTR model used in Fig. 7C? Without this information, the manuscript is impossible to read.

We agree with reviewer's comment and rewrote this section. In Figure 7B, doxycycline was used to induce the transient expression of wildtype or mutant (YYR/AAA) KLF4. We incorporated this detailed information in the figure legend: "*U2Os cells with wildtype (pLenti-tet-on-KLF4^{WT}) or mutant (pLenti-tet-on-KLF4^{YYR/AAA}) KLF4 were incubated with 10 ng/ml doxycycline for 24h and then treated with 10uM doxorubicin (Dox) or cisplatin (CDDP) for an additional 24h.*"

Regarding the BTR model used in Fig. 7C, we improved our description and added more information in the original figure legend of Fig. 7C: "*temperature-induced senescence in BTR model (RasV12-induced senescence).*" and we added this information in the main text: "*Moreover, we observed that the failure in KLF4 PARylation increases a barrier for induction of tumor senescence in BTR model (RasV12 -induced senescence).*" Meanwhile, we have clarified the statement regarding the BTR model in the materials and methods: "*BTR senescence assay. The BTR cells (immortalized MEFs, co-expressing a temperature-sensitive simian virus 40 (SV40) large T mutant and RasV12) maintained proliferation indefinitely at 32°C but became senescent at 39.5°C owing to both the rapid disappearance of large T antigens and the presence of RasV12. BTR cells with overexpressed wild-type or mutant KLF4 were planted at 32°C and then shifted to the restrictive temperature (39.5°C). Colonies started to appear after 7–10 d. The cells were left and refed on tissue culture plates for 3 weeks, then fixed and stained with crystal violet.*" This information clearly indicates what the BTR model is (immortalized MEFs, co-expressing a temperature-sensitive simian virus 40 (SV40) large T mutant and RasV12) and what it is used for (temperature-induced senescence).

The BTR model (RasV12-induced senescence) was a gift from Daniel S. Peeper (Netherlands Cancer Institute) (*Nat Cell Biol.* 2005 Nov;7(11):1074-82.), and we have used this model to evaluate the effect of KLF4 on RasV12-induced senescence (*Nat Commun.* 2015 Sep 30;6:8419.). This cell line (referred to as BTR) is mouse embryonic fibroblasts (MEFs), which were conditionally immortalized with a temperature-sensitive (ts) mutant of SV40 large T antigen and co-expressing RasV12. This cell line is transformed at 32°C, but undergoes RasV12-induced

senescence at 39.5°C, when ts, large T antigen is inactive. If genes could overcome RasV12-induced senescence at 39.5°C, there will be clone growth 39.5°C. To strength the clarity of the BTR model explanation, we rewrote the sentence in the main text as: “*KLF4 has been reported to inhibit RasV12-induced senescence in a BTR model, mouse embryonic fibroblasts (MEFs), which were conditionally immortalized with a temperature-sensitive (ts) mutant of SV40 large T antigen and co-expressing RasV12 (Hu et al., 2015; Rowland et al., 2005). The BTR cells are transformed at 32°C, but undergo RasV12-induced senescence at 39.5°C, when ts, large T antigen is inactive. We observed that the failure in KLF4 PARylation will no longer to be a barrier for induction of tumor senescence in this BTR model (Fig 7C) (Peeper et al, 2002).*”

Technical comments:

Fig 2A, please show Western blot for KLF4 and PARP1 after 5Gy irradiation. This the small increase in binding a consequence of change in abundance?

We did show the Western blot for FLAG-KLF4 and PARP1 after 5Gy irradiation in the original Fig. 2C. In Fig. 2C, we carried out pulldown of ectopic KLF4 (tagged with Flag/HA) complex by using anti-Flag beads followed by measuring co-immunoprecipitated PARP1 in the absence and presence of gamma radiation. As indicted in Fig. 2C, while no striking change of protein expression levels for either FLAG-KLF4 and PARP1 were observed, we did observe a significant increase of abundance of PARP1 being co-immunoprecipitated with KLF4 after the cellular exposure to gamma radiation (see both Fig 2A & C).

The different effects of PARylation on p21 expression versus BRCA1 expression is remarkable. Is this also the consequence of differences in chromatin binding between p21 and BRCA1 promoters?

Yes. We observed that PARylation of KLF4 has no obvious effect on BRCA1 expression, but it significantly enhances p21 transcription. As shown in Fig. 6O & P, we found KLF4 could accumulatively bind to the BRCA1 promoter, localized around -266~-261. However, neither the PARylation inhibitor olaparib nor 5Gy radiation affects the binding between KLF4 with the BRCA1 promoter, indicating chromatin binding of KLF4 and BRCA1 promoter is independent of PARylation. For the p21 promoter, we observed that the mutation of the PARylation site on KLF4 dramatically decreases the binding of KLF4 protein to the promoter (Fig. S5H). We thus concluded the different effects of PARylation on p21 expression versus BRCA1 expression is the consequence of differences in chromatin binding between p21 and BRCA1 promoters.

Figure 6. Mechanistic insights into PARylated KLF4 and PARylation-independent KLF4 in DNA damage response and DNA repair. (O) *inset, top*, schematic diagram of the BRCA1 promoter cloning primer and the alignment of potential KLF4 binding motif on BRCA1 promoter with KLF4 binding motif on p21 and SLC5A6 promoter. Shown is the wild-type (BRCA1-WT) or mutant (BRCA1-AA) BRCA1 promoter luciferase reporter activity when co-transfect with KLF4 plasmids. KLF4 co-transfection promotes BRCA1-WT but not BRCA1-AA promoter reporter transcription. (P) ChIP analysis of KLF4 binding to the BRCA1 promoter in MDA-MB-231 at -3K positions relative to the TSS in untreated and olaparib (10 μ M for 8h) or 5Gy radiation treat cells. No significant difference of KLF4 binds to BRCA1 promoter between untreated and olaparib or radiation treat cells.

Figure S5. Mechanistic role of DNA damage-induced KLF4 PARylation in orchestrating the recruitment of KLF4 to the chromatin. (H) Validation of the impact of KLF4 PARylation on KLF4-mediated transcription using p21 and Bax promoter luciferase assay in 293T cells. Mutation of KLF4 PARylation decreases KLF4 binding to p21 and Bax promoter.

In Figure 8, the effects of combining olaparib and KLF4 knockdown are additive at best, please provide synergy calculation.

The combination of olaparib and KLF4 knockdown are, indeed, additive. We calculated the combination effect between olaparib and KLF4 knockdown as $[(\text{observed treatment 1 value})/(\text{control value})] \times [(\text{observed treatment 2 value})/(\text{control value})] \times (\text{control value})$; the result is 1.00. A ratio >1 indicates a synergistic effect, and a ratio <1 indicates a less-than-additive or antagonistic effect (Mai et al, 2007; Zhou et al., 2004). Therefore, the effects of combining olaparib and KLF4 knockdown are additive.

Treatment	shLuci/ placebo	shLuci/Placebo	shklf4 placebo	shklf4/Olaparib
Tumor weight (g)	1.506 \pm 0.144	1.211 \pm 0.169	1.131 \pm 0.158	0.907 \pm 0.196
Combination index	1.002494218			

Dasatinib and ABT-263 appear to inhibit exogenous GFP-KLF4 fusion protein. Was this also tested on endogenous protein? Fig 8F does not specify what the protein was that was blotted. Fig 8F is endogenous. Needs clarify in the legends The concept of BRCA-ness is not mentioned. Is it possible that KLF4 low breast tumors display BRCAness? This would greatly increase translational relevance.

We tested endogenous KLF4 in Fig. S8H. The legend for Fig. S8H ("The Western blot of KLF4 protein after treating with indicated compounds at 10 μ M for 24h") indicated its expression of KLF4 protein but not GFP-KLF4

protein. To make it clear, we added the word “endogenous” to specify KLF4 and rewrote the figure legend as “The Western blot of endogenous KLF4 protein after treating with indicated compounds at 10 μ M for 24h in MDA-MB-231 cells” for Fig. S8H.

The canonical definition of BRCAness is a defect in homologous recombination repair, mimicking BRCA1 or BRCA2 loss (*Trends Cell Biol.* 2019;29(9):740-751.). We added BRCAness conception to the introduction: “Development of PARP inhibitors, including olaparib, rucaparib, niraparib and talazoparib, has provided a method to treat triple negative breast cancer (TNBC) patients with BRCAness. The BRCAness mimic BRCA1 or BRCA2 loss and are deficient with regards to homologous recombination (HR), based on their synthetic lethal effect (McCann & Hurvitz, 2018; Papadimitriou et al, 2018).” We also added it to the discussion part of the main text: “While extensive efforts have been made on the translational study of PARP inhibitors in BRCAness patient, the non-BRCA1/2 mutants such as RAD51, BRIP1, PALB2, and FANCA are slow relative low (Johnson et al., 2011; Min et al, 2015).”.

For the question regarding if it is possible that KLF4 low breast tumors display BRCAness, in our homologous recombination assay of KLF4 knockdown and overexpression in U2OS-DR-GFP, a reporter cell line for homologous recombination, we found decreased KLF4 inhibits homologous recombination that reflects BRCAness (Fig. 6G & R). Therefore, we hypothesize KLF4 low expression breast tumors would display BRCAness and will sensitize PARP-inhibitors, which we will examine further in our ongoing research.

Figure 6. Mechanistic insights into PARylated KLF4 and PARylation-independent KLF4 in DNA damage response and DNA repair. (G) Effect of KLF4 PARylation on NHEJ and HR. Wild-type or KLF4-Zinc 2-YYR/AAA mutant KLF4 were co-transfected with I-SCE construction in U2Os-GFP-EJ5 cells (for NHEJ assay) or U2Os-GFP-DR (for HR assay). (R) HR analysis. U2OS-DR-GFP cells were transfected with I-SceI, BRCA1 and siBRCA1 in KLF4-wild-type and depletion condition, respectively. GFP positive cells representing HR repair rate were measured by flow cytometry 48–72 hour after then. Overexpression of BRCA1 restores the HR efficiency in KLF4 knockdown cells. Data are mean \pm SEM; one-way ANOVA was used for the statistical analysis.

Need to analysis whether KLF4 low breast tumors display BRCAness. It is remarkable that the two drugs that act on KLF4, dasatinib and ABT-263 are both powerful senolytic drugs, coincidence? May be test quercetin also as this is also a senolytic, together with dasatinib.

In our homologous recombination assay of KLF4 knockdown and overexpression in U2OS-DR-GFP, a reporter

cell line for homologous recombination, we found decrease KLF4 inhibits homologous recombination, which reflects BRCAness (Fig. 6G & R). Therefore, we hypothesize KLF4 low expression breast tumors would display BRCAness and sensitize PARP-inhibitors.

Figure 6. Mechanistic insights into PARylated KLF4 and PARylation-independent KLF4 in DNA damage response and DNA repair. (G) Effect of KLF4 PARylation on NHEJ and HR. Wild-type or KLF4-Zinc 2-YYR/AAA mutant KLF4 were co-transfected with I-SCE construction in U2Os-GFP-EJ5 cells (for NHEJ assay) or U2Os-GFP-DR (for HR assay). **(R)** HR analysis. U2OS-DR-GFP cells were transfected with I-SceI, BRCA1 and siBRCA1 in KLF4-wild-type and depletion condition, respectively. GFP positive cells representing HR repair rate were measured by flow cytometry 48–72 hour after then. Overexpression of BRCA1 restores the HR efficiency in KLF4 knockdown cells. Data are mean \pm SEM; one-way ANOVA was used for the statistical analysis.

In our recent publication (*Cell Signal 70: 109574*), we have studied the mechanism of how dasatinib decreases the KLF4 expression and revealed that dasatinib promotes KLF4 ubiquitination for that dasatinib inhibits phosphorylation of VHL by Src kinase, and subsequently enhances proteolysis of VHL, that in turn leads to upregulation of KLF4 and increases endocrine resistance (Adapted Fig.2G & H and Fig. 8 in *Cell Signal 70: 109574*). Dasatinib enhances KLF4 ubiquitination at low doses (0.1 μ M), where no obvious onset of apoptosis is visible. Therefore, we hypothesize that though dasatinib is a powerful senolytic drug, its effect on suppressing KLF4 is independent from onset of apoptosis, but via Src-VHL axis-mediated KLF4 ubiquitination pathway.

Regarding the suggestion for testing another senolytic drug, quercetin, quercetin is already included in the 422 anti-cancer compounds, and the screening results indicated that quercetin has no effect on GFP-KLF4 expression (Fig. S8F, GFP-KLF4/DAPI index results: quercetin (1.201), average compound index (1.191 \pm 0.088) vs dasatinib (1.025) and ABT-263 (0.987)) (Fig. S8F).

Adapted from Fig. 2G & H in *Cell Signal 70: 109574*

Figure removed

Adapted from Fig. 8 in *Cell Signal* 70: 109574

Figure removed

Figure S8. Suppression of KLF4 sensitizes triple negative breast tumor to olaparib. (F) MDA-MB-231 with pLenti-GFP-KLF4 stable expression cells were treated with an anti-cancer compound (10 μ M) for 24 hours and then the GFP and DAPI fluorescence strength was assayed. The rankings of the GFP fluorescence strength of 422 anti-cancer compounds library are shown by the plot map; each dot represents 1 anti-cancer compound. The top 2 anti-cancer compounds are listed.

Referee #2 (Comments on Novelty/Model System for Author):

The manuscript provides conceiving data that at times is very well controlled and at other it needs correction. The novelty of PARylation mediated PARP1 on KLF4 and the mapping of the physical contact were well done. I am concern of the length of the manuscript it appears that two papers were put together so the story is not fluid. It was difficult to read because of was huge amounts of overlapping datasets. I was less convince that KLF4 is predictive of TNBC but rather more general for all breast cancers. Nonetheless a therapy may arise form this for TNBC.

We greatly appreciate the reviewer's recognition of novelty and the significance of our work. I understand the critique regarding the length of the manuscript and trimmed substantial portions of data by reorganizing them as the supplemental information shown in the revised manuscript. To make the story fluid after data reorganization, we laid a foundation to bridge the PARP1/KLF4/P21 DNA damage response pathway part and PARylation-independent KLF4/BRCA1/HR repair pathway portion.

We agree with the notion that KLF4 protein is a general predictor for all breast cancers, as shown in Fig 3E & F, due to accumulation of KLF4 protein in all three type of breast cancer as compared to the normal. In this manuscript, we are focusing on the role of KLF4 in DNA damage in TNBC cohort. The rationale to concentrate on KLF4 in TNBC is that the critical role of KLF4 in regulating PARP1/KLF4/P21 DNA damage response pathway as well as the PARylation-independent KLF4/BRCA1/HR repair pathway. Given the powerful treatment of endocrine therapy and targeted therapy on luminal A/B and Her2 types of breast cancer (there is no need to consider new therapeutic strategies for these types), the impact of KLF4 in governing tumor cell genome stability provides a good avenue to target TNBC tumors.

Figure 3. Interaction between KLF4 and PARP1 in breast cancer cells and expression of KLF4 and PARP1 in human breast cancer specimens. (E) Statistical analysis of PARP1 staining among normal, ER/PR positive (ER/PR), HER2 positive, and triple negative breast cancer (TNBC). **(F)** Statistical analysis of KLF4 protein staining among normal, ER/PR positive, HER2 positive and TNBC.

Referee #2 (Remarks for Author):

In this manuscript, Zhou et al provide evidence that Kruppel-like factor 4 (KLF4) is regulated by poly (ADP-ribose)ylation (PARylation) in response to genotoxic stress. The functional role of KLF4 as a tumor suppressor or as an oncogene is controversial that is context and tissue dependent. The PIs generate a conditional KLF4^{Flx/Flx} mice and utilized adeno-associated virus (AAV) gene delivery system to intestinal epithelial cells, which carries Cre-recombinase, they conditionally knockout KLF4 in the intestinal epithelia's cells. Upon exposure to 8Gy of radiation the overall survival of mice decrease KLF4 null mice. The increase expression of DNA damage markers was noted in vivo this may suggesting that KLF4 is functioning in survival, upon γ -radiation exposure. In support of the above results, the PI demonstrated that in mouse embryonic cells (MEFs) that have been KO for KLF4 have increased the percentage of cells with chromosome instability and upon radiation exposure, the KLF4 null cells, are associate with markers for homologous recombination (HR) but not with nonhomologous end joining (NHEJ). Suggesting KLF4 plays a role in HR DNA repair but not NHEJ. To gain mechanistic insights, mass spectrometry analysis with tagged-KLF4 pull downs revealed physical interactions between KLF4 and PARP1. The PIs posit that this interaction is increased upon exposure to γ -radiation. They

further map the YYR (i.e. Y430, Y451 and R452) motifs on KLF4, that undergo PARylation that is mediated by PARP1 causing KLF4 lead to enter the nucleus and bind to the chromatin. The authors argue that this lack of DNA damage respond that modulates HR is through BRCA1 activity, as KLF4 is shown to regulate BRCA1 expression upon γ -radiation. The authors provide evidence that PARP1 mediate the recruitment chromatin to ensure its transcriptional activity. Furthermore, evidence is provided that PARylated KLF4 facilitates apoptosis in response to DNA damage and this is reduced in the YYR mutant in triple negative cell lines. The presence of wt KLF4 sensitizes PARP inhibitors in combination with DNA damage drugs in several TNBC cell lines that are either wt for BRCA1 and or mutated for BRCA1 in vitro, this sensitivity is lost in part when a version of the mutated KLF4 YYR is used. The author also able to demonstrates that in vivo the wt KLF4 is able to sensitize PARP inhibitors Olirapab in combination with Dasatinib (a Src inhibitor) and or ABT263 which targets BCL2 members as negative regulators of apoptosis. The manuscript provides extensive amounts of data to argue for the relevance of PARylated KLF4 mediate by PARP1 to collaborate in chromosome instability and promoting tumors. As it stands the manuscript is viable for publication, but several major and minor problem need to be address.

Thanks to the reviewer for summarizing the whole story and indicating the quality of our manuscript is viable for publishing. We are able to address all major and minor concerns listed below.

General concerns of the text and results description were found to be sparse and often lacking in accurate details that match the text to access the results shown in the figures. Requiring one to go to the legends and at times vital information was missing. This is particularly true for the description of which cell lines where been used for several of the figures. One problem is that so many different models are been interplay intestinalis in vivo, MEFS , osteosarcomas, breast cancer cell lines ER+ vs. TNBC, using one cell line and make conclusion on another cell line model is often used which raise concerns. Additionally, with the vast amount of the data one needed to read several times to ensure accuracy for the conclusion which where been made and at times this was not always easy to follow and or it was a reasonable conclusion that one could agree with. The manuscript needs to be streamlined and shorten for general readers who are not experts in the field and to help with the reading Cell line, model, need to be clarify. Manuscript needs to be streamlined and shorten

Thanks to the reviewer for pointing out the excessive length and the need for streamlining certain parts of the original manuscript . To improve, we rewrote the results to streamline and shorten the text. We also have added detail on cell types and models in all figure legends and/or main manuscript text.

Major/Minor Concerns.

1. Figure 1 using the AAV model system for intestinal epithelial cells has nothing to do with breast cancer and the cause of death in the mice is not shown, did the mice die from tumors of the intestines? This was not shown or did the mice die form another cause as KLF4 is ubiquitously expressed and AAV model system can transduced most any cell and or tissue organ. It argues that KLF4 protect from radiation death.

The aim of Figure 1 using the AAV model system for intestinal epithelial cells is to dissect the impact of KLF4 in governing genome stability, but not specifically in breast cancer. In this model, we used intraperitoneal injection of AAV7-Cre-mCherry into KLF4^{loxp/loxp} in mice, which induced significant local KLF4 knockout in intestinal tissue, and then subjected the mice to 8 Gy total-body γ -radiation. We then detected the γ -radiation-induced DNA damage in the intestine in the 24h-96h period and monitored mice survival; the onset of death was 1 week after radiation. During this whole process, no mammary tumors were observed on the mice. The mice died due to the γ -radiation-induced DNA damage in the intestine, not because of tumors of the intestines. The point is that the result of this experiment drew our attention to the ability of KLF4 to be a cellular radiosensitizer.

KLF4 is ubiquitously expressed in various tissues and could act as either tumor suppressor or oncogene, depending on the tissue type and physiological context. One of the caveats for using germline intestine conditional knockout mice is it potentially generated many pathological changes in the intestine, such as distorted

architecture with increased glandular formation and reduced numbers of goblet cells at 3 weeks of age (*Dev Biol.* 2011 Jan 15;349(2):310-20.). It also increased genetic instability as well as accelerated progression of colitis-associated colorectal cancer (*Mol Cancer Res.* 2019 Jan;17(1):165-176.). Thus, the inducible AAV model system being utilized in our study could overcome this issue.

The early onset of pathological changes for the germline knockout will affect the studies of KLF4's role in response to DNA damage. In our work, we applied a new approach and employed KLF4^{loxp/loxp} coupled with the AAV7-Cre-inducible mouse model, which is an adult, induced conditional KLF4 knockout. The advantage of this conditional knockout is that it largely minimizes pathological changes due to KLF4 knockout before treatment with radiation.

In addition, this model applied intraperitoneal injection AAV-GFP, which could have limited adult induced KLF4 knockout due to tissue tropism of AAV. Intraperitoneal injection of AAV2/10/11 has been reported as possibly targeting the liver and spleen (please see adapted figure from *Virology.* 2004 Dec 20;330(2):375-83). We also found the tissue tropism of AAV-GFP serotype 7 3 weeks after intraperitoneal injection targets the stomach, duodenum, jejunum and ileum. The gastrointestinal (GI) injury is a major cause of acute death after total-body exposure to large doses of ionizing radiation (*Radiat Res.* 2010 May; 173(5): 579–589.), and the tissue tropism of AAV-GFP serotype 7 is limited to gastrointestinal injury. We concluded that our AAV-induced, adult conditional knockout mice model is a reasonable model to investigate and confirm the protective effect of KLF4 from radiation death.

Figure removed

2. Published work with the AAV system to generate tumors has been shown at 5Gy this model used 8Gy which is quite toxic to survival of the mice, what happened at 5Gy?

We performed intraperitoneal injections of AAV-GFP into KLF4^{loxp/loxp} mice to establish a gastrointestinal system, adult, conditional KLF4 knockout mouse model to investigate the role of KLF4 in DNA damage *in vivo*. We did not use this conditional model to study tumorigenesis, because this model is prone to form gastrointestinal system tumors but not mammary gland tumors. We use 8Gy instead of 5Gy to study the radio-protective effects of KLF4 because 8Gy for mice is a reasonable dose for radio-protection research for that at this dose, the wildtype mice have incidence of irradiation induced death in 1 to 2 weeks at. We think using a radiation dose of 5Gy would not cause survival issues after radiation (please see adapted Fig. 2a from *Spectrochim Acta A Mol Biomol Spectrosc.* 2019 Dec 5;223:117282.).

Adapt from Fig 2a of *Spectrochim Acta A Mol Biomol Spectrosc.* 2019 Dec 5;223:117282.

Figure removed

3. The supplemental figure 2 is well control in mefs, was it to argue for the intestinal model? this was not clear other than to state that KLF4 plays an important role in MEFS for chromosome instability and for HR and not NHEJ.

In Figure 1, we used intraperitoneal injections of AAV-GFP to establish a gastrointestinal system, adult conditional KLF4 knockout mouse model to reveal the radio-protective effects of KLF4 *in vivo*. In the tissue section of KLF4 knockout mice, we found an increase in DNA damage and cell death by staining with 53BP1, γ -H2AX and active-caspase-3. Then we wanted to determine the role of KLF4 in DNA damage response and DNA repair. But in the *in vivo* gastrointestinal system model, it is hard to dissect the various pathway of DNA damage response and DNA repair. We further used the *in vitro* MEFs as a model to discern the mechanism of KLF4 in DNA damage. In addition, the MEF cells from KLF4 knockout mice are a good model for studying DNA damage response and DNA repair (*Mol Carcinog.* 2015 Sep;54(9):889-99.; *Mol Cancer.* 2013 Aug 6;12:89.; *Oncogene.* 2009 Mar 5;28(9):1197-205.). Thus, we examined the MEFs and revealed that KLF4 plays an important role in MEFS for chromosome instability and for HR, but not for NHEJ. From the *in vivo*, radiation-induced gastrointestinal system DNA damage to the *in vitro* MEFs model, we focused on the role of KLF4 in DNA damage response.

4. Figure 2 is an example were neither the text, figure and or the legends tells us which cell line is been used. U20S, 293T, MEFS or breast cancer?

Thank you for the comments! We have added detail regarding cell lines in the legend of Fig. 2G-I. We improved the description of cell lines in other panels of Fig. 2 in the original manuscript. We double-checked all figures and added necessary information for all cell lines.

5. Fig2 L use 10Gy to show for the PAR antibody was up in response to radiation, yet all the preceding figure used 5Gy why? Not explained in the text no rationale for this or which cell line was used.

We are sorry for the confusion! The original legend for Fig. 2L “(L) PARP1 inhibitors decrease KLF4 PARylation. U2OS cells were pretreated with 10 μ M various PARP1 inhibitor niraparib, olaparib and rucaparib for 1hr followed by exposure to 10 Gy radiation for 4 hours. KLF4 was pulled down and the PARylation was detected” used the cell line U2OS.

At the beginning of this project, we initially tested both 5Gy and 10Gy γ -radiation and found both doses could induce KLF4 PARylation, while 10Gy looks stronger in U2OS cells (please see figure below). Doses from 5Gy to 10Gy have been used in published papers, such as *Proc Natl Acad Sci U S A. 2018 Feb 20;115(8):E1759-E1768*. We applied 5Gy for most experiments. In Figure 2L, we compared various clinical PARP1 inhibitors, including niraparib, rucaparib and olaparib. To see the difference, so we used a stronger dose to induce the KLF4 PARylation.

Fig. U2OS cells were exposure to 5 or 10 Gy radiation for 4 hours. KLF4 was pulled down and the PARylation was detected.

6. Figure 3 the survival curves for the importance for KLF4 is for breast cancer in general and not TNBC. Some of the figures in the supplemental figure 3 are the same as in the general figure?

Yes, Fig. 3 is the survival curves, which indicate the importance of KLF4 for breast cancer in general. There is some limitation of the batch of tissue array XT16-054, which contains 117 cases of invasive ductal breast cancer tissue samples with clinical follow-up, but only 12 out of the 117 cases are TNBC. So, we cannot obtain a robust result for TNBC in survival curves.

We double-checked Fig. 3 and Fig. S3 and did not see the same issue. Maybe the reviewer was referring to Fig. S3E and Fig. 3G; Fig. 3E is a representative staining of PARP1 and KLF4 in the first batch of tissue arrays containing 183 breast invasive ductal carcinomas and 10 pairs of primary cancer and adjacent normal tissue specimens. Fig. S3G is representative staining of PARP1 and KLF4 in the first batch of tissue arrays containing 117 invasive ductal breast tissue samples and clinical follow-up. Please see detailed information in the materials part: “For immunohistochemistry staining, two batches of tissue arrays have been used. The first batch of tissue arrays, 183 cases of breast invasive ductal carcinomas tissue specimens, and 10 pairs of primary cancer and adjacent normal tissue specimens were analyzed. The 10 pairs of tissue were obtained from patients who were treated with surgical resection alone in 2014 at the Cancer Hospital Chinese Academy of the Medical Science. None of the patients had received radiotherapy or chemotherapy before surgery. The specimens were immediately fixed in 4% polyformaldehyde and completely embedded in paraffin. Clinical characteristics of patients are summarized in Table S2. The second batch of tissue arrays, XT16-054, which contains 117 cases of invasive ductal breast cancer tissue samples with clinical follow-up, used to assess prognosis outcome was purchased from Shanghai Outdo Biotech Company. Clinical characteristics of patients are summarized in Table S3.” To avoid confusion, we added the tissue array information in the legend of Fig. 3E: “(G & H) Elevated expression of PARP1 and KLF4 are significantly correlated in 183 breast invasive ductal carcinoma human breast cancer tissue specimens. Representatives of paired IHC staining of PARP1 and KLF4 (case 1-3) are shown (G). Scale bars, 100 μ m.

Figure S3. Interaction between KLF4 and PARP1 in breast cancer cells and expression of KLF4 and PARP1 in human breast cancer specimens. (B) Survival analysis of KLF4 protein expression in 78 ER positive breast cancer patient. Compared to patients with low KLF4 protein expression, patients with high protein expression had an inferior cumulative survival rate (LogRank P=0.001). (C) Survival analysis of PARP1 protein expression in 117 breast cancer patients. No significant difference in cumulative survival rate between patient group with high PARP1 protein expression and patient group with low PARP1 protein expression (LogRank P=0.519). (D) Survival analysis of PARP1 protein expression in 78 ER positive breast cancer patient. Compared to patients with low PARP1 protein expression, patients with high protein expression had an inferior cumulative survival rate (LogRank P=0.026). (E & F) Elevated expression of PARP1 and KLF4 are significantly correlated in 117 cases of human breast cancer tissue specimens. Represents of paired IHC staining of PARP1 and KLF4 (case 1-3) are shown. (F) Statistic analysis of IHC staining in 117 cases of breast cancer patient which contains prognosis data indicates that PARP1 expression is positively correlated with KLF4 expression in breast cancer ($r=0.292$, $p=0.001$).

Figure 3. Abnormal KLF4-PARP1 axis correlates with breast cancer poor prognosis. (C & D) Statistic results of PARP1 and KLF4 IHC staining. (E) Statistical analysis of PARP1 staining among normal, ER/PR positive (ER/PR), HER2 positive, and triple negative breast cancer (TNBC). (F) Statistical analysis of KLF4 protein staining among normal, ER/PR positive, HER2 positive and TNBC. (G & H) Elevated expression of PARP1 and KLF4 are significantly correlated in 183 breast invasive ductal carcinoma human breast cancer tissue specimens. Representatives of paired IHC staining of PARP1 and KLF4 (case 1-3) are shown (G). Scale bars, 100 μ m. (H) Statistic analysis of IHC staining indicates that PARP1 expression is positively correlated with KLF4 expression in breast cancer. (I) Survival analysis of KLF4 protein expression in 117 breast cancer patients. Compared to patients with low KLF4 protein expression, patients with high KLF4 levels had an inferior cumulative survival rate (LogRank P=0.012); Low, staining weak and moderate; High, staining strong. (J) Survival analysis of both KLF4 and PARP1 protein expression in 66 breast cancer patients. Compared to patients with both low KLF4 and PARP1 protein expression levels, patients with both high KLF4 and PARP1 levels had lower cumulative survival rate (LogRank P=0.022).

7. No p-values are shown for the TMA results Figure S3C-B. No apparent differences between HER2+ and/or TNBC and as this is the main argument for your model later in the MS this may be more important as a main figure rather than as a supplement and may argue against published work

The p-values are now labeled (Fig 3E &F). We observed that abnormal KLF4-PARP1 axis correlates with poor prognosis for all breast cancer. Our work focuses more on the KLF4 function in DNA damage response and DNA repair. The reason to concentrate on TNBC is that targeting DNA damage response and DNA repair is more beneficial in TNBC. We do not think this will be a main argument for our model later in the manuscript to focus KLF4/PARP1 in TNBC. We did not claim KLF4 prognosis value is for TNBC alone; it is for all breast cancer patients. As the reviewer suggested, we moved this figure to the main figures as Fig 3E &F.

Figure 3. Interaction between KLF4 and PARP1 in breast cancer cells and expression of KLF4 and PARP1 in human breast cancer specimens. (E) Statistical analysis of PARP1 staining among normal, ER/PR positive (ER/PR), HER2 positive, and triple negative breast cancer (TNBC). **(F)** Statistical analysis of KLF4 protein staining among normal, ER/PR positive, HER2 positive and TNBC.

8. Published work has shown the in TNBC KLF4 low high expression correlates with better survival while low expression has a poorer survival outcome contradicting the conclusion of the authors.

Published work that has shown that based on mRNA expression, KLF4 low expression correlates better with survival, while low expression has a poorer survival in TNBC (*Breast Cancer*. 2017 Mar;24(2):326-335.). We agree with the observation that KLF4 mRNA is downregulated in breast cancer tissue as shown in various database and published papers (*J Biol Chem*. 2012 Apr 20;287(17):13584-97). However, when we look at two databases–The Cancer Genome Atlas (TCGA) for KLF4 mRNA expression and Clinical Proteomic Tumor Analysis Consortium (CPTAC) database for RNA protein expression–(please see attached results), we observed obviously uncoupled phenomena between the KLF4 transcript and its protein expression levels. Therefore, the KLF4 transcript cannot truly reflect the role of KLF4 protein expression in breast cancer.

Why are KLF4 transcript and protein levels dramatically uncoupled? Previous studies revealed a striking feature of KLF4; it is tightly regulated by posttranslational modifications. KLF4 protein stability is orchestrated by the interplay between arginine protein methylation and ubiquitination, providing a possible mechanism to explain the molecular underlying mechanisms for the uncoupling between KLF4 transcript and its protein expression levels (*Genes Dev*. 2019 Aug 1;33(15-16):1069-1082.; *Nat Commun*. 2015 Sep 30;6:8419.; *J Biol Chem*. 2011 Mar 4;286(9):6890-901.; *Mol Cell*. 2012 Jan 27;45(2):233-43.). In addition, KLF4 undergoes proteolysis in response to various oncogenic signals such as an estrogen signal (*J Biol Chem*. 2012 Apr 20;287(17):13584-97), RAS/RAF/MEK/ERK signal (*Oncogene*. 2017 Jun 8;36(23):3322-3333.) and EGFR/HER2 signal (*Cell Death Dis*. 2015 Mar 19;6:e1699; *Mol Carcinog*. 2019 Nov;58(11):2118-2126.). These oncogenic signals potentially enhance KLF4 protein stabilization, that in turn causes KLF4 protein accumulation in breast cancer tissue. Indeed, the results of immunohistochemistry analysis from us and others based on various cohorts of breast cancer patient specimens clearly demonstrated the accumulation of KLF4 protein in breast cancer tissue (*J Biol Chem*. 2012 Apr 20;287(17):13584-97; *Onco Targets Ther*. 2014 Oct 24;7:1963-9.; *Cancer Lett*. 2019 Dec 28;467:19-28; *Clin Cancer Res*. 2004 Apr 15;10(8):2709-19.). Thus, information based on the KLF4 mRNA levels cannot reflect the accumulation of KLF4 protein. In addition, Dong MJ, *et al* reported that KLF4 proteins function as an independent predictive marker for pathologic complete remission in breast cancer neoadjuvant

chemotherapy in a case-control study (*Onco Targets Ther.* 2014 Oct 24;7:1963-9.), which further confirmed the observation from us and others (please see adapted **Fig.1 of *Onco Targets Ther.* 2014 Oct 24;7:1963-9.**).

Similar to the uncoupled KLF4 transcript and protein levels in breast cancer, *Dhaliwal NK, et al* also reported KLF4 transcript and protein levels are uncoupled in embryonic stem cells (*Genes Dev.* 2019 Aug 1;33(15-16):1069-1082.), which indicated the mRNA expression of KLF4 cannot reflect KLF4 protein levels in various pathological and physiological pathways.

Figure removed

Adapted from Fig. 1 of *Onco Targets Ther.* 2014 Oct 24;7:1963-9.

Figure removed

9. *Figure S3A is out of place to show and IP and in what cell line Figure S3I has no loading control and is out of place.*

Thanks for the comments! We added detailed cell line information in the legend of Fig. S3A. This figure corresponds to main Figure 2C. We incorporated this figure into Fig. S3, which shows the interaction between

KLF4 and PARP1 in breast cancer cells and the expression of KLF4 and PARP1 in human breast cancer specimens.

Actually, we carried out and had a whole panel of loading control in Fig. S3I (Fig S3G in revised manuscript). The reason for not including it is we thought the Western blot of pull-down KLF4 may be more accurate than pure loading control. However, we have now added the loading control that we acquired previously, please see the new Fig. S3G with original loading control.

10. Figure 3 I-J the IF is of very poor quality.

Fig. 3I & J is staining of KLF4, PARP1 in tissue sections to show their co-location. To improve the quality issue, we have moved Figs. 3I & J to the supplementary data as Fig. S3H & I. We performed a new experiment and added the proximity staining of KLF4 and PARP1, which is more accurate, to see the co-location of PARP and KLF4 (Fig. 2I).

Figure 2. Identification of KLF4 PARylation by PARP1 in KLF4-mediated DNA damage response. . (I). Validation of the interaction between endogenous KLF4 and PARP1 by in situ proximity ligation assay (PLA). No positive staining in KLF4/Rabbit IgG antibody PLA assay. Scale bar, 100 μ m. The right panel show the blow-up.

11. Figure 4 no problems very well done and nicely control mapping the YYR motif on KLF4.

We appreciate the reviewer's nice comments.

12. Figure 5 is another example where the text and the figure legend does not tell us which cell lines that assay are been done I assume 293T?

Thanks for comments! We have added the detailed cell line information for all main figures and supplementary figures, including Fig.5.

13. Figure 5B the flag IPs with the mutations YYR is under load compare to wt and or YKH KLF4 that can explain why so see less KLF4Par, similar for mutation of Y451A. This is a critical dataset to argue for the model need corrections.

The Y451A has almost equal, or slightly more KLF4:PAR as compared to the mutant YYR, suggesting the tyrosine 451 could be the site where the PAR chain modified.

When we use 2'-O-a-D-ribofuranosyl adenosine (RFA) from APLF pdb structure (2KQD), which mimics the PAR chain, to dock onto KLF4, using SMINA (an interface for Autodock), the RFA clearly dock on the tyrosine 451 site (Fig. 5A & B and Fig. S4A-C). Principally, the mutant of tyrosine 451 will abrogate the attached PAR chain, which should be same effect as the triple mutant YYR/AAA. However, we still see a faint band in Y451A, which is why we used the mutation YYR for most experiments. The Flag IPs with the YYR mutation are similar to the mutations of Y451A in less KLF4:PAR modification further suggested that the YYR motif, especially the tyrosine 451 site, is where the PAR chain modified. To make it clear, we rewrote the results part of Y451A: “We expected the tyrosine 451 site to be the RFA docking site, based on computation modelling (Fig. 5A and Fig. S4A-C). Same as triple the KLF4-YYR mutant Y430A/Y451A/R452A, the mutation of tyrosine 451 to alanine almost abrogate the PARylation modification, which further implicated tyrosine 451 as the site of PAR chain attachment (Fig. 5B).”

Figure 5. Mechanistic role of DNA damage-induced KLF4 PARylation in orchestrating the recruitment of KLF4 to chromatin. (A) Simulation analysis of the potential KLF4 PARylation site (PBZ domain) with or without DNA binding. The triple amino acids Y430, Y451, R452 on KLF4 zinc finger 2 (Znf2) is identified as the potential YYR motif involved in mediating PARP1 modification. **(B)** The effect of YYR motif mutations on KLF4 PARylation. Constructs of KLF4-Zinc 1-YKH/AAA mutation, KLF4-Zinc 2-YYR/AAA mutation and KLF4-Zinc-2-Y451A mutation were co-transfected with Myc-PARP1 into 293T cells, respectively, and then pulled-down using M2 agarose followed by measuring KLF4 PARylation with anti-PAR antibody.

Figure S4. Results on KLF4-PARP1 interactions (A) Protein Binding Zinc (PBZ) domains in APLF (2KQE). The RFA (2'-O- α -D-ribofuranosyl adenosine) binds the zinc finger motif containing the YYR motif formed by Y386, Y381 and R387. **(B)** The PBZ domain Y430/Y451/R452 motif (YYR) in KLF4 (PDB id: 2WBS) is structurally comparable to that in APFL. **(C)** Docking of RFA onto the YYR motif of KLF4. *Left panel*, without DNA; *Right panel*, with DNA.

14. Fig5E IP flag should have a nuclear marker for loading control H3?

We actually measured ORC2 as a loading control for chromatid protein control in our original data. We did not include the chromatid protein loading control ORC2 in Fig. 5E, because we thought when we made subcellular fractions of cell extracts, the fractions collected from each step (cytosol/nuclear/chromatid fraction) are from the same batch of cells. So, we only showed one control of actin. To improve the control quality according to the reviewer's suggestion, we added the chromatin control ORC2 acquired in our original data. We also ran the same batch of lysates for Histone 3 as suggested by the reviewer (Fig. 5E).

15. Figure 5F the literature does not support the loss of ORC2 when cells are exposed to 5Gy on the chromatid, yet this is clear and when KLF4YYR is use it is more pronounce and again no loading control for the chromatid unless ORC2 was supposed to be that control. Similarly Figure S5 D and E no loading controls.

In Figure 5F, we used ORC2 as a control for chromatid proteins. ORC2 was used as a control for chromatid protein loading control in a previous publication (please see adapted Fig. 3A from *EMBO J. 2010 May*

19;29(10):1726-37.). Many other labs also measured and found ORC2 was only localized to chromatin fraction (please see adapted Fig. 2B from *EMBO J. 2004 Jul 7;23(13):2651-63.*). We consistently use ORC2 as a control for loading of chromatin fraction, which is the same as histone 3 (H3) protein. In Fig. 5D and Fig. S5B & C, we extracted whole cell lysates, nuclear soluble, nuclear matrix and chromatin fraction, and then use ORC2 as a control as well.

Adapted from Fig. 2B in *EMBO J. 2004 Jul 7;23(13):2651-63.*

Adapted from Fig. 3A in *EMBO J. 2010 May 19;29(10):1726-37.*

Figure removed

Figure removed

16. Figure 5I a critical figure for mechanism to differentiate PARP1 motif necessary to interact with KLF4 is underloaded i.e. H909A to explain the loss of PAR antibody. Need to be address. Figure 5I control second lane is underloaded in response to 5Gy exposures

Reviewer may have misunderstood the left panel of Fig. 5I. The left panel shows the Western blot detecting PAR, Myc-PARP1 and FLAG-KLF4 after the immunoprecipitation with anti-FLAG antibody. The right panel shows the input of PAR, Myc-PARP1 and FLAG-KLF4. In the left panel. The second lane does not show underloaded but indicates the significantly decreased binding between Myc-PARP1 and FLAG-KLF4, which is consistent with the first lane decreasing PARylation of KLF4.

There are no significant differences in pulldown FLAG-KLF4 and the input of PAR, Myc-PARP1 as well as FLAG-KLF4. Therefore, these results indicated that H909 of PARP1 plays an essential role in binding to KLF4 and mediating KLF4 PARylation. To make this clear, we rewrote the figure legend as: “(I) The effect of PARP1 mutations (H909A or T824A) on KLF4 PARylation confirms the critical role of H909 in mediating KLF4 PARylation in 293T cells. Myc-PARP1 wildtype or mutations (H909A or T824A) and FLAG-KLF4 were co-transfected into 293T cells, and then the FLAG-KLF4 was pulled down by anti-FLAG antibody. The PARylation modification of KLF4 was blotted with anti-PAR antibody, and the binding PARP1 was blotted with anti-myc antibody.”

Figure 5. Mechanistic role of DNA damage-induced KLF4 PARylation in orchestrating the recruitment of KLF4 to chromatin. (I) The effect of PARP1 mutations (H909A or T824A) on KLF4 PARylation, confirming the critical role of H909 in mediating KLF4 PARylation in 293T cells. Myc-PARP1 wildtype or mutations (H909A or T824A) and FLAG-KLF4 were co-transfected into 293T cells and then the FLAG-KLF4 was pulldown by anti-FLAG antibody, the PARylation modification of KLF4 was blotted with anti-PAR antibody, and the binding PARP1 was blotted with anti-myc antibody.

17. KLF4 PARylation is require for p21 but not BRCA1 expression and thus raises a good point, can BRCA1 rescue HR in the absence of KLF4? Additionally, the authors should investigate the levels of apoptosis and status of p21 in response to the drug treatment by either doing immunoblotting directly from tumor extracts and or by Immunohistochemistry analysis.

In the original Fig. 6R, we have completed experiments adding back BRCA1 in KLF4 KD U2OS-DR-GFP reporter cells and discovered that overexpression of BRCA1 could increase GFP positive cell percentage, indicating BRCA1 could rescue HR in the absence of KLF4.

Second, as the reviewer suggested, we added experiments to investigate the levels of apoptosis by Bax and the status of p21 in response to the drug treatment by immunohistochemistry analysis. We stained P21 and Bax in 4T1 xenograft tumors with KLF4 knockdown and olaparib treatment and found the combination of KLF4 knockdown and olaparib significantly suppressed P21 expression but increased Bax expression (Fig. S8C). We also stained H & E, KLF4, PARP1, p21 and Bax in MDA-MB-231 xenograft tumors. The combination of ABT-263 or dasatinib with olaparib treatment decreased KLF4 and p21 expression but increased Bax expression in xenograft tumors (Fig. 8I).

Figure 6. Mechanistic insights into PARylated KLF4 and PARylation-independent KLF4 in DNA damage response and DNA repair. (R) HR analysis. U2OS-DR-GFP cells were transfected with I-SceI, BRCA1 and siBRCA1 in KLF4-wild-type and depletion condition, respectively. GFP positive cells representing HR repair rate were measured by flow cytometry 48–72 hour after then. Overexpression of BRCA1 restores the HR efficiency in KLF4 knockdown cells. Data are mean \pm SEM; one-way ANOVA was used for the statistical analysis.

Figure S8. Expression of KLF4 in 4T1 cells and xenograft tumors. (C) Staining of H & E, Ki67, KLF4, PARP1, p21, Bax and activated caspase 3 in 4T1 xenograft tumors. Scale bars, 100 μ m. Combination of KLF4 knockdown and Olaparib treatment decrease Ki67 positive population cells, decrease p21, but increase Bax and activated caspase-3 in xenograft tumors.

Figure 8. Suppression of KLF4 sensitizes triple negative breast tumor to olaparib. (I) Staining of H & E, KLF4, PARP1, p21 and Bax in MDA-MB-231 xenograft tumors. Scale bars, 100 μ m. Combination of ABT-263 or dasatinib with Olaparib treatment decrease KLF4 and p21 expression, but increase Bax expression in xenograft tumors.

Referee #3 (Comments on Novelty/Model System for Author):

Overall the manuscript relates novel findings on KLF4 in response to damage, apparently outlining two responses that appear independent of one another. This gets complicated to follow. Technically there are some controls that are lacking and a lack of consideration of cell cycle and potentially confounding factors such as p53 state. These will need to be addressed before it can be considered for publication.

We appreciate reviewer's compliment regarding the novelty of our findings. We agree with reviewer's critique and suggestion regarding the length of manuscript. We have consolidated and shortened our results, which streamlines the manuscript. We also added the details for cell types and models in all figure legends and/or main manuscript text. Regarding some issues of control and cell cycle measurement as well as confounding factors such as the p53 state, we carried out several new experiments and included the new results in the revised manuscript.

Referee #3 (Remarks for Author):

Zhou et al present an interesting manuscript describing a role for KLF4 parylation in genome instability and cancer. The work outlines how PARYlation of KLF4 regulates its location. KLF4 is a transcription factor. The authors mapped interacting motifs between KLF4 and PARP1 and showed how disrupting their interaction impacts DNA damage response, particularly BRCA1 promoted homologous recombination events via regulation of BRCA1 transcription, though there seems to be separable functions (PARYlation of KLF4 and transcriptional regulation of BRCA1). Though the overall findings are interesting there are some major concerns and additional aspects of this finding that should be explored/dealt with.

Initial work was to establish a role for KLF4 in DNA damage response in vivo, using conditional KLF4 knockout and exogenous exposure to an adenovirus Cre that expresses in the intestine. Impact of conditional deletion of KLF4 on morphology and sensitivity to 8Gy irradiation was measured. Zhou et al observed increased DNA damage and apoptosis based on morphological changes and IHC of DNA damage markers (H2Ax and 53BP1) and apoptosis. They went on to verify these results in cell culture models measuring chromosomal aberrations, accumulation of damage by 53BP1 and H2Ax foci and then directly measuring HR and NHEJ using cell based assays DR-GFP and EJ5. A defect was found in HR, but not NHEJ.

To complement these studies, Zhou et al should also evaluate classic DDR signaling responses (ATM-CHK2-p53).

Thanks to the reviewer for describing our findings as interesting! To complement these studies, as suggested by the reviewer, we have added results based on a series of new experiments that evaluated classic DNA damage response signaling responses (ATM-CHK2-p53). First, we detected the p-CHK2/Thr68 and p53 expression in the intestinal epithelium of KLF4^{loxp/loxp} mice, injected with AAV7-mCherry or AAV7-Cre-mCherry followed by radiation treatment. This revealed no p-CHK2 difference at different time points between the AAV7-mCherry and AAV7-Cre-mCherry group. P53 levels fluctuated—lower after 6h but higher at 96h after radiation in the AAV7-Cre-mCherry group. These results indicate that adult conditional knockout KLF4 has a weak effect on the ATM-CHK2-P53 pathway after exposure to radiation (Fig. S1D & E).

We then asked whether PARYlation of KLF4 regulated p21 through p53. To date, we applied p53 knockout HCT116 (a standard cultured-cell model to study p53 in DNA damage response) to dissect the role of p53. As shown in Fig. S7A, we observed that upregulation of p21 by KLF4 depends on its PARYlation but not on p53. We then asked if the function of KLF4 PARYlation in cell death depends on p53. We applied the same model to answer this question and found that the cell death prevention role mediated by KLF4 PARYlation depends on p53 (Fig S7C). Moreover, we observed in MDA-MB-231 cells that the knockdown KLF4 dramatically decreased p21 and BRCA1 expression, but only a slight increase in p53 expression was measured (Fig S7G).

Finally, we employed the U2OS cells (a model cell line for DNA damage response) based on ectopic expression of KLF4^{WT} or KLF4^{YYR/AAA} followed by exposure to γ -radiation for 48 hr. Cell lysates were collected 4 hr after radiation, and then were immune blotted with BRCA1, KLF4, p-CHK2/Thr68, CHK2, P53, PARP1 and actin. As shown in Fig. S7H, no obvious difference in BRCA1, CHK2, p-CHK2/Thr or p53 was observed between the wildtype and mutant KLF4 (KLF4^{YYR/AAA}).

In summary, results from the above experiments suggest that no correlation with the canonical ATM-CHK2-P53 pathway is observed, either for the PARylation-dependent KLF4-P21-DNA damage response or the PARylation-independent KLF4-BRCA1-HR pathway.

Figure S1. Establishment of KLF4^{loxp/loxp}/AAV-Cre-mCherry conditional adult gut-intestine knockout mice model. (D & E) Immunofluorescent staining of p53 and p-CHK2/Thr68 in the intestinal epithelium of KLF4^{loxp/loxp} mice with injection of AAV7-mCherry or AAV7-Cre-mCherry followed by treatment of irradiation. Tissues were collected from sham mice and mice at different time after exposure to irradiation and then staining with indicated antibodies. **(E)** Quantification of p53 and p-CHK2/Thr68 based on the Immunofluorescent staining results presented in **D**. Data are mean \pm SEM; one-way ANOVA was used for the statistical analysis. Scale bars, 60 μ m. No difference of p-CHK2 at different time point between AAV7-mCherry and AAV7-Cre-mCherry group. P53 seems lower in 6h but higher at 96h after radiation in AAV7-Cre-mCherry group, which is not consistent.

C

Figure S7. KLF4 regulates BRCA1 expression. (C) p53. p53^{+/+} or p53^{-/-} HCT116 cells were transfected with p53, wild-type (KLF4^{WT}) or mutant KLF4 (KLF4^{YYR/AAA}) were treated with 5Gy or 20Gy radiation, and 48h later the cell viability was measured by CCK-8. KLF4^{WT} suppress cell death independent with p53.

G

H

Figure S7. KLF4 regulates BRCA1 expression.

(G) MDA-MB-231-ShScr and ShKLF4 cells were treated with 5Gy radiation and the cell lysates were collected at indicated time. BRCA1, p21, p53, PARP1, KLF4 and actin were blotted. Knockdown KLF4 decrease BRCA1 and KLF4 expression, but slightly increase p53 expression.

(H) U2OS cells were transfected with KLF4^{WT} or KLF4^{YYR/AAA} plasmids and then subjected for γ -radiation treatment after 48h. Cell lysates were collected at 4h after radiation, and then blotting with BRCA1, KLF4, p-CHK2/Thr68, CHK2, P53, PARP1 and Actin. No difference of BRCA1, CHK2, p-CHK2/Thr, p53 between wildtype and mutant KLF4 (KLF4^{YYR/AAA}). **(H)** The expression of BRCA1 and p21 in KLF4^{-/-} MEFs with transfection of wildtype (KLF4^{WT}) or mutant KLF4 (KLF4^{YYR/AAA}, KLF4^{C403A} and KLF4 ^{Δ Zinc2}). While the wildtype (KLF4^{WT}) or mutant KLF4 (KLF4^{YYR/AAA}) could promote BRCA1 expression, only the wildtype (KLF4^{WT}) could enhance p21 expression.

To better understand KLF4 role in damage repair, Zhou et al looked for protein interactors. PARP1 and KLF4 interact and this interaction increases in response to damage, both in the cytoplasm and in the nucleus. Other than PARP1 peptides, what other protein interactions were found. This may provide insight into the type of dynamics associated with PARylation of KLF4. Were any of these known PARP1 targets? This can be compared to various proteomic screens such as in PMID: 24055347

In this manuscript, we only excised the obvious upregulated bands (after exposure to gamma-radiation) for mass spectrometry assay, leading to the identification of PARP1 as a binding partner for KLF4. The biochemical interaction between KLF4 and PARP1 was validated at various levels, including co-IP, proximity-ligation assay and co-localization by immuno-staining. In our previous purification (*Mol Cell.* 2012 Jan 27;45(2):233-43.),

PARP1 was also found in the KLF4 complex with other binding partners, such as VHL, elongin B, Cullin 2 as well as PRMT5. These binding partners are not known PARP1 targets in PMID: 24055347.

Where are the controls for purity of obtained fractions?

We thank the reviewer for the comment. We repeated the experiment and added the Histone 3 (H3) as a loading control for nuclear fraction (NE) and actin for cytosol fraction loading control (Fig. 2E).

Fig2E. Is KLF4 also interacting with cleaved PARP1? How much commitment is there to apoptosis is with these cells at 5 Gy at 4 hrs?

KLF4 binds to PARP1 C-terminal 829Aa-1014Aa, especially to the amino acid H909 (Fig. 4F & G and 5H). We hypothesize KLF4 could bind to the 89KD PARP1 fragments. Indeed, we were able to observe a faint band in the original Fig. 2E, although the band did not appear all the time because of the limitation of the subset activated apoptotic population and cleavage of PARP1.

No obvious cell apoptosis happened at 5Gy after 4 hrs of exposure. At 4 hrs after irradiation, cells were still exhibiting DNA damage response followed by the recruitment of DNA repair machinery to the damaged lesion site for repair. Even at 48 hrs after radiation at 5 Gy, approximately 5% of apoptotic cells were being measured (Fig S7C).

Figure 4. Identification of molecular motifs on KLF4 and PARP1 that mediate KLF4 PARylation. (F) Schematic diagram of human PARP1 functional domains and strategy for engineering a series of PARP1 deletion mutants. (G) Identification of amino acid stretch 829-1014 on the C-terminal segment of PARP1 to be involved in mediating its interaction with KLF4 in 293T cells.

Figure 5. Mechanistic role of DNA damage-induced KLF4 PARylation in orchestrating the recruitment of KLF4 to chromatin.) Two interaction sites are observed in PARP1-KLF4 interaction pose. One of them is at znf1 and the other is at znf2 (at the same site where RFA binds). H909 and T824 on PARP1 are the potential core amino acids mediating the interaction pose.

Original Fig. 2. (E) Validation of interaction between endogenous KLF4 and PARP1 in cytosolic lysate and nuclear lysate using immunoprecipitation and Western blotting in MDA-MB-231 cells.

Need controls for the total amount of proteins (KLF4 and PARP1) and how it changes upon 5Gy at 4 hrs. The localization of KLF4 appears to move to more defined focal locations in response to damage, that look similar to PARP1 locations. Are these foci sites of damage or something like parspeckles with accumulated PARylated RNA processing proteins? Can the proportion of co-localization be quantified? Perhaps a PLA assay would clarify this IF based assessment.

We confirmed that PARP1 modified KLF4 and then assist KLF4 recruitment to chromatin where KLF4 function as transcriptional function, such as p21 and Bax promoter, then KLF4 will determined genes such as Bax, p21. This process is one example of PARylation regulates from chromatin remodeling, recruitment, transcriptional regulation, mRNA synthesis, RNA export, mRNA stabilization to protein expression and function. We highly agree that these foci sites of damage or transcriptional hotspot are something like parspeckles with accumulated PARylated RNA processing proteins. To confirm these hypotheses would be our next scope.

We have conducted the in situ proximity ligation assay (PLA) by using KLF4 mouse monoclonal antibody and PARP1 rabbit antibody. The results have been incorporated into Fig 2I. The PLA assay clear indicated the direct interact with KLF4 and PARP1 *in vivo*.

Figure 2. Identification of KLF4 PARylation by PARP1 in KLF4-mediated DNA damage response. (I). Validation of the interaction between endogenous KLF4 and PARP1 by in situ proximity ligation assay (PLA). No positive staining in KLF4/Rabbit IgG antibody PLA assay. Scale bar, 100 μ m. The right panel show the blow-up.

Discussion PARylated RNA processing proteins.

We add the discussion of PARylated RNA processing proteins in to discussion part : "*Recruitment of KLF4 from soluble nucleus to chromatin and regulation of subcellular compartmentation of RNA-processing proteins FUS share similarity with regards to protein modification by PARylation. While it is clear that localization of FUS is through formation of parspeckles via liquid-liquid phase separation with accumulated PARylated proteins, damaged DNA, PARP-1, and mRNA (Singatulina et al, 2019), if PAR:KLF4 also undergoes self-assembly by phase separation like RNA-processing proteins FUS remains unclear.*"

What are the cell cycle impacts of their various modulations as this is poorly described? Fig6C suggests altered cell cycle response, but this only measures G2/M proportion. Especially important is evaluating G1/S checkpoint and S-phase progression.

In response to radiation-induced DNA damage, there are two major checkpoint responses, including G1/S and G2/M. The G2/M checkpoint is a fast but insensitive process. Low-dose irradiation fails to completely prevent entry into mitosis, resulting in many cells entering mitosis with double-strand breaks (DSBs). Higher doses (≥ 0.5 –1 Gy) fully initiate the G2/M checkpoint with only a few cells escaping arrest. The activation of the G1/S checkpoint is a slow process, but at the first 4–6 h after ionizing radiation (IR), only a slowing of S-phase entry can be observed, allowing many cells to enter S phase even after high doses. Thus, during this time, many cells enter S phase with high DSB levels. Under this circumstance, the G2/M checkpoint is more responsible for irradiation DNA damage (*Crit Rev Biochem Mol Biol. 2011 Aug; 46(4): 271–283.*). Therefore, we decided to focus more on the G2/M transition.

In the original manuscript, we found KLF4 depletion decreased p21 expression and abrogated radiation-induced G2/M arrest by measuring p-CDK1 and p-H3 phosphorylation after radiation and we observation of high p3 positive population (Fig. 6A-C).

Mutation of KLF4 by YYR/AAA cannot restore the depletion of G2/M arrest, indicating that PARylation of KLF4 is essential for KLF4-mediated G2/M arrest (Fig. 6A-C). The inhibiting effect of KLF4 on aneuploidy is consistent with the observed effect on G2/M transition (Fig. 6F).

Regarding the effect of p21 in G1/S transition, we measured the G1/S transition based on BrdU-chase staining. As shown in Fig. S7B, the radiation causes a weak effect on G1/S phase arrest. However, KLF4^{+/+} and KLF4^{-/-} show differently in S phase, suggesting KLF4^{-/-} MEF has more S phase accumulation than KLF4^{+/+} MEF.

Figure 6. Mechanistic insights into PARylated KLF4 and PARylation-independent KLF4 in DNA damage response and DNA repair. (A) Depletion of KLF4 directly diminishes the DNA damage-induced p21 expression, resulting in the failure of cell cycle arrest in MDA-MB-231 cells. **(B)** Abolishment of KLF4 PARylation disrupts KLF4-mediated p21 expression that in turn impairs DNA damage response in MDA-MB-231 cells. **(C)** p-H3 staining analysis of KLF4^{+/+}, KLF4^{-/-} MEFs, KLF4^{-/-} MEF with transfection of wild-type or KLF4-Zinc 2-YYR/AAA mutation.

Figure 6. Mechanistic insights into PARylated KLF4 and PARylation-independent KLF4 in DNA damage response and DNA repair. (F) Failure of KLF4 PARylation leads to increased aneuploidy population in U2OS cells.

Figure S7. KLF4 regulates BRCA1 expression. (B) BrdU chase assay. KLF4^{+/+}, KLF4^{-/-}, KLF4^{-/-}/KLF4^{WT}, KLF4^{-/-}/KLF4^{YYR/AAA} with 5Gy irradiation 4h treatment and then chase with BrdU for 2 hrs, then staining with Alexa488-anti-BrdU and 7-AAD. KLF4^{-/-} and KLF4^{-/-}/KLF4^{YYR/AAA} show more S phase in compare with KLF4^{+/+}, KLF4^{-/-}/KLF4^{WT}.

The authors went on to evaluate KLF4 and PARP1 levels in TNBC and observed increase levels compared to normal mammary epithelial cells and co-accumulation of these 2 proteins. Higher KLF4 expression was associated with shorter survival.

The authors then performed deletion mapping to identify fragments necessary for KLF4:PARP1 interaction. They went on to identify a YYR motif as a potential PARylation site on KLF4. Using site specific mutant the authors then show the requirement of these sites, without impacting other known KLF4 posttranslational modifications. In response to damage KLF4 accumulates on chromatin in a PARP1 dependent manner, and this accumulation was attenuated by mutation of YYR. This also resulted in reduced KLF4 ChIP at p21 and BAX promoters.

Fig5D, it appears in this figure that in response to Dox, PARP1 and KLF4 protein levels increase at least in chromatin, while KLF4 soluble nuclear fraction increased with PARP1i with 1 μ M Dox, so what are the protein level dynamics of these proteins (endogenous) in response to damage and/or DOX?

Actually, we measured the dynamics of total protein level of these proteins (endogenous) in response to damage and/or DOX in the original Fig. S5C (Fig. S5B & C). As shown in Fig. S5C, the endogenous PARP1 protein amount stayed even because there was no apoptosis onset in the short time after DNA damage/response and no alteration of PARP1 cleavage was monitored. The KLF4 protein levels also stayed even in the short term. We also carried out additional experiments to show PARP1 and KLF4 expression in response to radiation in revised Fig S7G. No obvious change of PARP1 and KLF4 expression from 0 hr to 6 hr after radiation was observed.

Fig 6F, are the cells accumulating in G2/M? Need to see progression with BrdU progression assay. Is this in U2OS cells?

Yes, it is U2OS cells. In Fig 6F, as compared to KLF4^{WT} and KLF4^{YYR/AAA} with or without radiation, more G2/M accumulation was observed in U2OS with KLF4^{YYR/AAA} at 48h after 5Gy radiation. We now have included an additional BrdU progression assay for KLF4^{+/+} MEF, KLF4^{-/-} MEF, KLF4^{-/-}/KLF4^{WT}, KLF4^{-/-}/KLF4^{YYR/AAA} in Fig. S7B. The KLF4^{-/-}/KLF4^{YYR/AAA} has a relative higher G2/M than KLF4^{-/-}/KLF4^{WT} at 4h after 5Gy radiation.

F

Figure 6. Mechanistic insights into PARylated KLF4 and PARylation-independent KLF4 in DNA damage response and DNA repair. (F) Failure of KLF4 PARylation leads to increased aneuploidy population in U2OS cells.

B

Figure S7. KLF4 regulates BRCA1 expression. (B) BrdU chase assay. KLF4^{+/+}, KLF4^{-/-}, KLF4^{-/-}/KLF4^{WT}, KLF4^{-/-}/KLF4^{YYR/AAA} with 5Gy irradiation 4h treatment and then chase with BrdU for 2 hrs, then staining with Alexa488-anti-BrdU and 7-AAD. KLF4^{-/-} and KLF4^{-/-}/KLF4^{YYR/AAA} show more S phase in compare with KLF4^{+/+}, KLF4^{-/-}/KLF4^{WT}.

Fig 6G - how is the HR result significant? Perhaps plot the individual datapoints over the bargraphs? And if the cells are arresting in cell cycle, is this the basis of the HR defect considering that the DR-GFP assay relies on amplification of the number of cells through 2+ cell cycles. Change bargraph to datapoint.

We calculated the p-value and labeled it in the figure in original manuscript. Though mutant KLF4 promotes G1/S transition and G2/M arrest, we do not think this effect on cell cycle significantly affects the cell proliferation and HR, based on our measurement. We changed the bar graph to data points (Fig 6G).

Figure 6. Mechanistic insights into PARylated KLF4 and PARylation-independent KLF4 in DNA damage response and DNA repair. (G) Effect of KLF4 PARylation on NHEJ and HR. Wild-type or KLF4-Zinc 2-YYR/AAA mutant KLF4 were co-transfected with I-SCE construction in U2Os-GFP-EJ5 cells (for NHEJ assay) or U2Os-DR-GFP (for HR assay).

Considering the cell cycle changes and that BRCA1 expression is cell cycle regulated, how much of the effect is associated with this and could the changes observed be due to this? Are the gene expression changes reflected at the protein level? BRCA1 protein level also display post translational changes over cell cycle (R. Baer paper) - is this observed or altered. Fig 6L for BRCA1 protein bands appears to be compressed. This should be presented in a gradient gel.

In addition to the R. Baer paper, we previously published an article about cell-cycle-dependent regulation of BRCA1 (*PLoS One*. 2010; 5(12): e14484). We found BRCA1 protein peaked in the G2/M phase and degraded after radiation by ubiquitination-proteasome pathway. In comparison with ShScr and ShKLF4 cells, there are no obvious differences in the G2/M phase (Fig. S7). Loss of KLF4 significantly decreases the background expression of BRCA1, suggesting the KLF4-BRCA1 panel is not cell cycle dependent.

We clearly showed the KLF4 proteins could bind to the BRCA1 promoter and enhance BRCA1 promoter transcription (Fig. S6M-P), suggesting the role of KLF4 in regulating BRCA1 transcription. BRCA1 has various posttranslational modifications, such as methylation, sumoylation, phosphorylation, PARylation, and ubiquitination. KLF4 is a transcriptional factor, and we are more interested in the direct function of KLF4 on BRCA1 than in the indirect signals cascade. Studies of BRCA1 protein post translational changes are beyond the scope at this time.

Based on the reviewer's suggestion, we now show the non-compressed BRCA1 bands in Fig. S7G.

Figure S7. KLF4 regulates BRCA1 expression. (G) MDA-MB-231-ShScr and ShKLF4 cells were treated with 5Gy radiation and the cell lysates were collected at indicated time. BRCA1, p21, p53, PARP1, KLF4 and actin were blotted. Knockdown KLF4 decrease BRCA1 and KLF4 expression, but slightly increase p53 expression.

Figure 6. Mechanistic insights into PARylated KLF4 and PARylation-independent KLF4 in DNA damage response and DNA repair. (M) KLF4 physically interacts with BRCA1 upstream promoter region measured by CHIP-PCR. (N) *inset, top*, schematic diagram of the BRCA1 promoter and relative positions of primer sets used in this study. CHIP analysis at the BRCA1 promoter using KLF4-specific or nonspecific control IgG (α -Gal4) in MDA-MB-231 cells. Shown is the enrichment at positions of the BRCA1 locus relative to the TSS, presented as percent recovery of input. (O) *inset, top*, schematic diagram of the BRCA1 promoter cloning primer and the alignment of potential KLF4 binding motif on BRCA1 promoter with KLF4 binding motif on p21 and SLC5A6 promoter. Shown is the wild-type (BRCA1-WT) or mutant (BRCA1-AA) BRCA1 promoter luciferase reporter activity when co-transfect with KLF4 plasmids. KLF4 co-transfection promotes BRCA1-WT but not BRCA1-AA promoter reporter transcription. (P) CHIP analysis of KLF4 binding to the BRCA1 promoter in MDA-MB-231 at $-0.3K$ positions relative to the TSS in untreated and olaparib (10 μ M for 8h) or 5Gy radiation treat cells. No significant difference of KLF4 binds to BRCA1 promoter between untreated and olaparib or radiation treat cells.

Are the effects, particularly on p21 and Bax expression, p53 dependent? Again - considering this question, what is the cell cycle impact? Is there a change in proportion of cells in G1 in response to damage? And how is KLF4 dependent response altered by p53 status?

Our results showed that the effect of PARylation of KLF4 on p21 and Bax expression is p53 independent (Fig. S7A & C).

C

Figure S7. KLF4 regulates BRCA1 expression. (C) p53. p53^{+/+} or p53^{-/-} HCT116 cells were transfected with p53, wild-type (KLF4^{WT}) or mutant KLF4 (KLF4^{YYR/AAA}) were treated with 5Gy or 20Gy radiation, and 48h later the cell viability was measured by CCK-8. KLF4^{WT} suppress cell death independent with p53.

Fig 7R. Can this decrease be rescued by co-depleting 53BP1? Please also note typographical error "KLF4/siRBCA1"

I think the reviewer means Fig 6R. We corrected the typographical error: "KLF4/siRBCA1". We have done the 53BP siRNA experiment and found 53BP siRNA could only slightly reverse the loss of KLF4 (Fig. S7K & L).

Figure S7. KLF4 regulates BRCA1 expression.

(K&L) KLF4^{+/+} and KLF4^{-/-} cells were transfected with pLCN DSB Repair Reporter (DRR) for homologous recombination assay. KLF4^{-/-}-pLCN DSB-GFP cells were transfected with 53BP1 siRNA and then transfected with I-SCE and subjected to flow cytometry for GFP positive cells. (K) shows flow cytometry assay, (L) shows the western blotting of 53BP1 after siRNA transfected.

Fig7 H-R - typographical error "Vetcor".

Thanks, we corrected the error.

With these viability assays there are some confusing results. KLF4^{wt} expression seems to improve in general. Is this BRCA1 dependent (siBRCA1) or can shKLF4 be rescued with si53BP1 co-depletion? HCC1937 is known to be BRCA1 mutant but PARP1i resistant (see for example PMID 25984718) - is the differences in viability altered with wt BRCA1 expression and is it dependent upon p53 status? And most of the tested cell lines have various p53 mutations, do these alter sensitivity and impact of KLF4 status? Typo error. " Vetcor ". Fig7 H-R.

We have done the 53BP siRNA experiment and found 53BP1 siRNA could only slightly reverse the loss of KLF4 (Fig. S7K & L).

We also found for the compound olaparib, HCC1937 is almost equal to the BRCA1 proficient cell line MDA-MB-231 (Fig. 7L-N). The Olaparib sensitivity in Fig. 7L-O is MDA-MB-436 > MDA-MB-468 > MCF7 > HCC1937 > MDA-MB-231. The resistance mechanism of HCC1937 may be that KLF4 is much more accumulated than MDA-MB-436 and MDA-MB-468 (Fig 3A).

Our results indicated that both PARylation-KLF4-P21/Bax (Fig. S7 A & C) and KLF4/BRCA1/HR (Fig. S7F) pathways are independent of p53 status. In addition, in HCT116 p53^{+/+} and p53^{-/-} cells, no obvious change of KLF4 expression was observed (Fig. S7A).

Figure S7. KLF4 regulates BRCA1 expression.

(K&L) KLF4^{+/+} and KLF4^{-/-} cells were transfected with pLCN DSB Repair Reporter (DRR) for homologous recombination assay. KLF4^{-/-}-pLCN DSB-GFP cells were transfected with 53BP1 siRNA and then transfected with I-SCE and subjected to flow cytometry for GFP positive cells. (K) shows flow cytometry assay, (L) shows the western blotting of 53BP1 after siRNA transfected.

Figure 7. Synergism of KLF4 and PARP1 in breast cancer treatment.

(L) Elevated KLF4 expression increases resistance to Olaparib in MDA-MB-468 (BRCA-proficient) cells, while disruption of KLF4 PARylation attenuates KLF4-driven resistance to Olaparib. (M-N) Elevation of mutant KLF4 shows the same effect as WT KLF4 to Olaparib in BRCA1 mutant cell lines MDA-MB-436 (M) and HCC1937 (N) cells. No significant difference between expression of wild-type KLF4 and KLF4 PARylation-deficient mutant was observed in both MDA-MB-436 and HCC1937 cells. (O-P) Modulating KLF4 by knockdown or overexpression affects cellular response of MDA-MB-231 (O) and MCF7 (P) to Olaparib in clonogenic assay.

Figure 3. Abnormal KLF4-PARP1 axis correlates with breast cancer poor prognosis. (A) Expression of PARP1 and KLF4 in mammary gland epithelial cell and various types of breast cancer cell lines.

A

Figure S7. KLF4 regulates BRCA1 expression. (A) PARylation of KLF4 promotes p21 expression is independent on p53. p53^{+/+} or p53^{-/-} HCT116 cells were transfected with p53, wild-type (KLF4^{WT}) or mutant KLF4 (KLF4^{YYR/AAA}). Lysate were collected for detecting the expression of p21 and KLF4 48 hrs after the transfection.

C

Figure S7. KLF4 regulates BRCA1 expression. (C) p53. p53^{+/+} or p53^{-/-} HCT116 cells were transfected with p53, wild-type (KLF4^{WT}) or mutant KLF4 (KLF4^{YYR/AAA}) were treated with 5Gy or 20Gy radiation, and 48h later the cell viability was measured by CCK-8. KLF4^{WT} suppress cell death independent with p53.

F

Figure S7. KLF4 regulates BRCA1 expression. (F) KLF4 promotes BRCA1 in both p53 wild-type or mutant cells. p53^{+/+} or p53^{-/-} HCT116 cells were transfected with wild-type (KLF4^{WT}) or mutant KLF4 (KLF4^{YYR/AAA}). Lysates were collected for detecting the expression of BRCA1 and KLF4.

A

Figure S7. KLF4 regulates BRCA1 expression. (A) PARylation of KLF4 promotes p21 expression is independent on p53. p53^{+/+} or p53^{-/-} HCT116 cells were transfected with p53, wild-type (KLF4^{WT}) or mutant KLF4 (KLF4^{YYR/AAA}). Lysate were collected for detecting the expression of p21 and KLF4 48 hrs after the transfection.

HCC1937 cell line has strong basal expression of p21, while the MDA-MB-436 and MHH-ES1 cell lines do not (Supplementary Fig. S5). Thus, it is possible that the HCC1937 cell line arrests during the cell cycle and does not proliferate at the same rate as the MDA-MB-436 and MHH-ES1 cell lines *in vivo*.

We could not obtain the MHH-ES1 cell line, so it is not shown in Fig. S5. We are unsure what the reviewer is referring to. We did not have *in vivo* experiments of the HCC1937, MDA-MB-436 and MHH-ES1 cell lines. They are beyond our scope at this time.

KLF4 binds to upstream region of BRCA1. This does not require KLF PARylation, while p21 expression depends on PARylation. Expression of BRCA1 rescued HR in absence of KLF4.

Authors then go on to look at clinical relevance and pre-clinical use with 4T1 and MDA231 mouse orthotopic tumor models.

Fig 8 - tumor size appears to be decreased upon combined KLF4 depletion and PARP1i treatment, but still progresses. Can the authors measure levels of apoptosis and p21 in response to treatments? To complement this work 4T1 cell should also be examined *in vitro* as they seem less responsive *in vivo* than MDA231 cells.

Based on the reviewer's suggestion, we carried out new experiments to investigate the levels of apoptosis by Bax and the status of p21 in response to the drug treatment; we made measurements by immunohistochemistry analysis. We stained P21 and Bax in 4T1 xenograft tumors with KLF4 knockdown and Olaparib treatment and found a combination of KLF4 knockdown and Olaparib significantly suppressed P21 expression but increased Bax expression (Fig. S8C). We also stained H & E, KLF4, PARP1, p21 and Bax in MDA-MB-231 xenograft tumors. A combination of ABT-263 or dasatinib with Olaparib treatment decreased KLF4 and p21 expression but increased Bax expression in xenograft tumors (Fig. 8I).

Yes, we agree that 4T1 cells should also be examined *in vitro* as they seem less responsive *in vivo* than MDA231 cells. In fact, we did acquire the results that 4T1 is more olaparib resistance than MDA-MB-231 cells *in vitro* (please see attached figure: *Colongenic assay compare the olaparib sensitivity of 4T1 and MDA-MB-231*).

Figure S8. Expression of KLF4 in 4T1 cells and xenograft tumors. (C) Staining of H & E, Ki67, KLF4, PARP1, p21, Bax and activated caspase 3 in 4T1 xenograft tumors. Scale bars, 100 μ m. Combination of KLF4 knockdown and Olaparib treatment decrease Ki67 positive population cells, decrease p21, but increase Bax and activated caspase-3 in xenograft tumors.

I

Figure 8. Suppression of KLF4 sensitizes triple negative breast tumor to olaparib. (I) Staining of H & E, KLF4, PARP1, p21 and Bax in MDA-MB-231 xenograft tumors. Scale bars, 100 μ m. Combination of ABT-263 or dasatinib with Olaparib treatment decrease KLF4 and p21 expression, but increase Bax expression in xenograft tumors.

Figure. Colongenic assay compare the olaparib sensitivity of 4T1 and MDA-MB-231. MDA-MB-231 and 4T1 cells were plating in 6-well plates and treated with Olaparib at indicated dose for 6 days and then further grow for 14 days. 4T1 shows a bit more resistance to MDA-MB-231

Overall the variations in different drugs used is confusing and not explained throughout the manuscript. It would also help to update the model figure to capture better the relationships of PARP1 related activities and transactivation activities and how this relates to damage response.

We did use doxorubicin as a DNA damage trigger and clinical PARP1-inhibitor Olaparib for most experiments. We compared three types of PARP1 inhibitor—Olaparib, rucaparib and niraparib. The doxorubicin and Olaparib were consistently used in this project.

Our model in Fig. 9 clearly indicated the PARylation-dependent KLF4-P21/Bax pathway and PARylation-independent BRCA1/HR pathway. We tried not to include too many KLF4/PARP1 non-related PARP1 activities, to keep the manuscript streamlined for readers.

Figure 9. Diagram of proposed working model.

When irradiation or DNA damage drugs caused DNA lesion, the PARP1 was activated at the DNA lesion site. PARP-1 is a component of non-homologous end joining (NHEJ) to repair the majority of double strand DNA breaks. Activated PARP1 will further add the PAR chain to transcriptional factor KLF4 and recruit it to the chromatin and then activate p21 or inactivate Bax transcription and finally activate the DNA damage response (DDR). KLF4 also activate the transcriptional BRCA1 in the manner of PARylation independent pathway to enhance homologous recombination (HR). Suppress of KLF4 would impeded DDR and HR, and further has potential synergism with PARP1 inhibitors.

23rd Sep 2020

Dear Yong,

Thank you for the submission of your revised manuscript to EMBO Molecular Medicine. We have now received the enclosed report from two of the three referees who were asked to re-assess it. Unfortunately, after a series of reminders we did not manage to obtain a report from Referee #1. In the interest of time, I prefer to make a decision now rather than further delaying the process. As you will see the referees are overall supportive and I am pleased to inform you that we will be able to accept your manuscript pending the following amendments:

1. Referee #3 still raised a series of - mostly minor - concerns on your work, which we would ask you to address and clarify.

2. In the main manuscript file, please do the following:

- Remove the yellow color font.
- Figure callouts: There is a callout for Fig 4 A-J but only Fig 4 A-G are provided. Please fix.
- Remove "data not shown". As per our guidelines, on "Unpublished Data" the journal does not permit citation of "data not shown". All data referred to in the paper should be displayed in the main or Expanded View figures.
- in legends, provide exact n= and exact p= values, not a range, along with the statistical test used. Some people found that to keep the figures clear, providing an Appendix supplemental table with all exact p-values was preferable. You are welcome to do this if you want to.
- In Material and Methods, provide the antibody dilutions that were used for each antibody.
- in Material & Methods (and in checklist), for animal work, confirm that all experiments were performed in accordance with relevant guidelines and regulations. The manuscript must include a statement in the Materials and Methods identifying the institutional and/or licensing committee approving the experiments. Gender, age and genetic background must be indicated, along with housing conditions.
- in Material & Methods (and in checklist), include a statement that informed consent was obtained from all subjects and that the experiments conformed to the principles set out in the WMA Declaration of Helsinki and the Department of Health and Human Services Belmont Report.

3. Appendix: The Appendix should begin with a short table of contents. The current table of contents is too basic, please add at least number of figs and tables (see example here

[https://www.embopress.org/action/downloadSupplement?](https://www.embopress.org/action/downloadSupplement?doi=10.15252%2Femmm.202012739&file=emmm202012739-sup-0001-Appendix.pdf)

[doi=10.15252%2Femmm.202012739&file=emmm202012739-sup-0001-Appendix.pdf](https://www.embopress.org/action/downloadSupplement?doi=10.15252%2Femmm.202012739&file=emmm202012739-sup-0001-Appendix.pdf)).

Nomenclature needs to be corrected to "Appendix Figure S1" etc. and "Appendix Table S1" etc. in appendix file and callouts in the manuscript text.

4. Funding: Please enter grant number NIH R01CA250110 to the online submission system when you resubmit the manuscript. In the manuscript, please merge Acknowledgements and Funding sections.

5. We would encourage you to include the source data for figure panels that show essential quantitative information. Additional information on source data and instruction on how to label the

files are available at < <https://www.embopress.org/page/journal/17574684/authorguide#sourcedata> >.

6. For more information: There is space at the end of each article to list relevant web links for further consultation by our readers. Could you identify some relevant ones and provide such information as well? Some examples are patient associations, relevant databases, OMIM/proteins/genes links, author's websites, etc...

7. Every published paper now includes a 'Synopsis' to further enhance discoverability. Synopses are displayed on the journal webpage and are freely accessible to all readers. They include a short stand first (maximum of 300 characters, including space) as well as 2-5 one sentence bullet points that summarize the paper. Please write the bullet points to summarize the key NEW findings. They should be designed to be complementary to the abstract - i.e. not repeat the same text. We encourage inclusion of key acronyms and quantitative information (maximum of 30 words / bullet point). Please use the passive voice. Please attach these in a separate file or send them by email, we will incorporate them accordingly.

8. As part of the EMBO Publications transparent editorial process initiative (see our Editorial at <http://embomolmed.embopress.org/content/2/9/329>), EMBO Molecular Medicine will publish online a Review Process File (RPF) to accompany accepted manuscripts.

a. In the event of acceptance, this file will be published in conjunction with your paper and will include the anonymous referee reports, your point-by-point response and all pertinent correspondence relating to the manuscript. Let us know if you do not agree with this.

I look forward to seeing a revised version of your manuscript as soon as possible.

Sincerely,
Jingyi

Jingyi Hou
Editor
EMBO Molecular Medicine

*** Instructions to submit your revised manuscript ***

To submit your manuscript, please follow this link:

Link Not Available

1) a .docx formatted version of the manuscript text (including Figure legends and tables)

2) Separate figure files*

3) supplemental information as Expanded View and/or Appendix. Please carefully check the authors guidelines for formatting Expanded view and Appendix figures and tables at <https://www.embopress.org/page/journal/17574684/authorguide#expandedview>

4) a letter INCLUDING the reviewer's reports and your detailed responses to their comments (as Word file).

5) The paper explained: EMBO Molecular Medicine articles are accompanied by a summary of the articles to emphasize the major findings in the paper and their medical implications for the non-specialist reader. Please provide a draft summary of your article highlighting

6) For more information: There is space at the end of each article to list relevant web links for further consultation by our readers. Could you identify some relevant ones and provide such information as well? Some examples are patient associations, relevant databases, OMIM/proteins/genes links, author's websites, etc...

7) Author contributions: the contribution of every author must be detailed in a separate section.

8) EMBO Molecular Medicine now requires a complete author checklist (<https://www.embopress.org/page/journal/17574684/authorguide>) to be submitted with all revised manuscripts. Please use the checklist as guideline for the sort of information we need WITHIN the manuscript. The checklist should only be filled with page numbers where the information can be found. This is particularly important for animal reporting, antibody dilutions (missing) and exact values and n that should be indicated instead of a range.

9) Every published paper now includes a 'Synopsis' to further enhance discoverability. Synopses are displayed on the journal webpage and are freely accessible to all readers. They include a short stand first (maximum of 300 characters, including space) as well as 2-5 one sentence bullet points that summarise the paper. Please write the bullet points to summarise the key NEW findings. They should be designed to be complementary to the abstract - i.e. not repeat the same text. We encourage inclusion of key acronyms and quantitative information (maximum of 30 words / bullet point). Please use the passive voice. Please attach these in a separate file or send them by email,

we will incorporate them accordingly.

You are also welcome to suggest a striking image or visual abstract to illustrate your article. If you do please provide a jpeg file 550 px-wide x 400-px high.

10) A Conflict of Interest statement should be provided in the main text

11) Please note that we now mandate that all corresponding authors list an ORCID digital identifier. This takes <90 seconds to complete. We encourage all authors to supply an ORCID identifier, which will be linked to their name for unambiguous name identification.

Currently, our records indicate that the ORCID for your account is 0000-0002-8693-3084.

Link Not Available

12) The system will prompt you to fill in your funding and payment information. This will allow Wiley to send you a quote for the article processing charge (APC) in case of acceptance. This quote takes into account any reduction or fee waivers that you may be eligible for. Authors do not need to pay any fees before their manuscript is accepted and transferred to our publisher.

Photos 400-800 DPI

*Additional important information regarding figures and illustrations can be found at <https://bit.ly/EMBOPressFigurePreparationGuideline>

The system will prompt you to fill in your funding and payment information. This will allow Wiley to send you a quote for the article processing charge (APC) in case of acceptance. This quote takes into account any reduction or fee waivers that you may be eligible for. Authors do not need to pay any fees before their manuscript is accepted and transferred to our publisher.

***** Reviewer's comments *****

Referee #2 (Comments on Novelty/Model System for Author):

The author have address all of my concerns

Referee #2 (Remarks for Author):

The author have address all of my concerns.

Referee #3 (Comments on Novelty/Model System for Author):

Generally the authors have used excellent model systems, but occasional switching of cell lines is still a concern.

Referee #3 (Remarks for Author):

Zhou et al present a much revised manuscript addressing many of the questions raised by this and other reviewers. Not only have many experimental details been clarified but a number of additional experiments were performed to address questions raised. Overall the manuscript is still complicated and tries to deal with two different aspects of KLF4 regulation, that which is dependent upon PARylation and another which is not, both in response to DNA damage induction. It is good that the authors have now included cell cycle into their consideration, but this has also raised new concerns. It is also good that that they were able to address the issue of p53 status and indicate that the role of KLF4 in the system they are evaluating is outside the influence of p53 activity or mutation. Beyond this, there remains some minor details that are either missing or are confusing.

Fig S2G-J Indicates loss of KLF4 causes increased damage, particularly after IR and an inability to progress beyond G2 into M (increasing 4n or >4n cells (or increased aneuploidy, but without the FACs data it is not possible to evaluate how this was qualified), but not pH3 in KLF4 depleted cells). Combined with this, later in the manuscript, is the inability to mount a p21 response in the absence of KLF4. Of note, it appears that Fig S2J is a replot of the same data as in Fig6C while Fig 6F

Fig S2K and L complements the points above, with no impact on NHEJ while HR is diminished in absence of KLF4.

There are additional problems with the presentation of this figure as it is noted to be conducted with IR, which is presumably incorrect (typically this IScel based system does not involve the use of IR), but then it is not clear what the "control" and "radiation" samples are indicating.

Figure S7B, it is noted that the authors did 5 Gy 4 hr treatment chase 2hr BrdU (what does this mean? - 5Gy IR, 4hr recovery, then 2hr BrdU chase?). Are the changes significant (+/-IR)? IT is also not possible from this experiment to determine whether there is a G1/S arrest or S-phase checkpoint in the presence of functional KLF4.

It is also still not clear how, with loss of KLF4 activation of p21, and cells presumably (according to the interpretation of the authors) entering S-phase despite DNA damage, that HR is then decreased and NHEJ is unaffected. Even if there is a lack of BRCA1 induced HR, there should be increased replication stress, ATR pathway activation and either death, s-phase arrest or an alternate compensatory DSB repair induced. This aspect of the manuscript still appears under developed and poorly explained. But I also believe that it was this incongruity, lack of cell cycle control and progress into S-phase in the face of damage while a lack of HR is exactly the reasons the authors attempt to cover the two mechanisms they eventually go on to address. Clarifying the reasoning and making this bridge is essential to making this study easier to understand in totality.

Throughout the manuscript there is still a lot of switching of cell models (MEFs, HEK293, MDA231,

U2OS), each used to look a different aspects of KLF4 biology and little continuity. There are some points where the biology observed in one cell line is recapitulated in another, which is good.

Overall the manuscript is significantly improved but there remains some reasonably significant minor concerns.

Response Letter

Dear Editor and Reviewers:

We greatly appreciate the opportunity to address the remaining concerns from the reviewer #3. We have done the amendments and response to Reviewer #3 by point-to-point.

Response to editor: Pages 2-5

Response to reviewer #3: Pages 6-10

Our point-by-point response to the editor and each reviewer's comments are below. To avoid any confusion in the figure numbers, which we have changed in the revised version, all new figure numbers in the description below are colored red.

Response to the editor's comments

Thank you for the submission of your revised manuscript to EMBO Molecular Medicine. First, please accept my apologies for the slow process. We have now received the enclosed report from two of the three referees who were asked to re-assess it. Unfortunately, after a series of reminders we did not manage to obtain a report from Referee #1. In the interest of time, I prefer to make a decision now rather than further delaying the process. As you will see the referees are overall supportive and I am pleased to inform you that we will be able to accept your manuscript pending the following amendments:

1. Referee #3 still raised a series of - mostly minor - concerns on your work, which we would ask you to address and clarify.

We appreciate your comprehensive evaluation of our manuscript and the decision of accept pending! We have now addressed the remaining concerns from the Referee #3 point- by -point, please following response in Page 6-9.

2. In the main manuscript file, please do the following:

- Remove the yellow color font.

The yellow color font which used as tracking has been removed in the revision file.

- Figure callouts: There is a callout for Fig 4 A-J but only Fig 4 A-G are provided. Please fix.

We corrected the callout for Fig 4 A-J to "Fig 4 A-G".

- Remove "data not shown". As per our guidelines, on "Unpublished Data" the journal does not permit citation of "data not shown". All data referred to in the paper should be displayed in the main or Expanded View figures.

The statement of " *Our recent pathological analyses of breast cancer specimens with various biomarkers suggest a connection between KLF4 and several DNA damage-related genes, including BRCA1 and p53 (unpublished data).* " on page 25 has been removed.

- in legends, provide exact n= and exact p= values, not a range, along with the statistical test used. Some people found that to keep the figures clear, providing an Appendix supplemental table with all exact p-values was preferable. You are welcome to do this if you want to.

The exact n= and exact p= values along with the statistical test used has been supplied in figure legends. The appendix p-value for figure 7H-R has been supplied in Appendix table S4.

- In Material and Methods, provide the antibody dilutions that were used for each antibody.

The antibody dilution has been incorporated now.

- in *Material & Methods (and in checklist)*, for animal work, confirm that all experiments were performed in accordance with relevant guidelines and regulations. The manuscript must include a statement in the *Materials and Methods* identifying the institutional and/or licensing committee approving the experiments. Gender, age and genetic background must be indicated, along with housing conditions.

The animal study was approved by the Institutional Animal Care and Use Committee of the Chinese Academy of Medical Sciences Cancer Hospital. We add the statement as well as Gender, age and genetic background of mice.

- in *Material & Methods (and in checklist)*, include a statement that informed consent was obtained from all subjects and that the experiments conformed to the principles set out in the WMA Declaration of Helsinki and the Department of Health and Human Services Belmont Report.

The statement that informed consent was obtained from all subjects and that the experiments conformed to the principles set out in the WMA Declaration of Helsinki and the Department of Health and Human Services Belmont Report has been added.

3. *Appendix: The Appendix should begin with a short table of contents. The current table of contents is too basic, please add at least number of figs and tables (see example here <https://www.embopress.org/action/downloadSupplement?doi=10.15252%2Femmm.202012739&file=emmm202012739-sup-0001-Appendix.pdf>). Nomenclature needs to be corrected to "Appendix Figure S1" etc. and "Appendix Table S1" etc. in appendix file and callouts in the manuscript text.*

We have added the short table of contents. We correct the citation of appendix figures from "Fig. S" or "Table S" to "Appendix Figure S" or "Appendix Table S".

4. *Funding: Please enter grant number NIHR01CA250110 to the online submission system when you resubmit the manuscript. In the manuscript, please merge Acknowledgements and Funding sections.*

We have added the grant number NIHR01CA250110 to the online submission system. We merged the Acknowledgements and Funding sections.

5. *We would encourage you to include the source data for figure panels that show essential quantitative information. Additional information on source data and instruction on how to label the files are available at < <https://www.embopress.org/page/journal/17574684/authorguide#sourcedata> >.*

We have included the source data for Appendix fig S5 in the last revision submission.

6. *For more information: There is space at the end of each article to list relevant web links for further consultation by our readers. Could you identify some relevant ones and provide such information as well? Some examples are patient associations, relevant databases, OMIM/proteins/genes links, author's websites, etc..*

Here are some relevant web links:

1. Author's websites: <https://www.feinberg.northwestern.edu/faculty-profiles/az/profile.html?xid=38433>

2. KLF4 gene: <https://www.ncbi.nlm.nih.gov/gene/9314>
3. PARP1 gene: <https://www.ncbi.nlm.nih.gov/gene/142>

7. Every published paper now includes a 'Synopsis' to further enhance discoverability. Synopses are displayed on the journal webpage and are freely accessible to all readers. They include a short stand first (maximum of 300 characters, including space) as well as 2-5 one sentence bullet points that summarize the paper. Please write the bullet points to summarize the key NEW findings. They should be designed to be complementary to the abstract - i.e. not repeat the same text. We encourage inclusion of key acronyms and quantitative information (maximum of 30 words / bullet point). Please use the passive voice. Please attach these in a separate file or send them by email, we will incorporate them accordingly.

We have included the Fig. 9 as the 'Synopsis' in last revision. Please see following key acronyms. We will attach these in a separate file.

Irradiation or DNA damage drugs caused DNA lesion and active DNA damage response for lesion detection and DNA repair. Double-strand DNA breaks are potential harmful lesions that can be repaired by BRCA1 regulated homologous recombination (HR) and PARP1 regulated non-homologous end joining (NHEJ) pathway.

1. Mass spectrometry identified the PARP1 is a binding partner for KLF4 protein in response to DNA damage.
2. Structure-based modeling and simulations elucidated the mechanism of interaction between KLF4 and PARP1, and unveiled that PBZ domain YYR motif (Y430, Y451 and R452) enables PARylation modification.
3. PARP1-mediated PARylation of KLF4 promotes the recruitment of KLF4 from the nucleus to the chromatin which, in turn, facilitates KLF4-mediated transcriptional function.
4. KLF4 PARylation is required for KLF4-mediated DNA damage response through regulating p21 and Bax, while KLF4-involved regulation of BRCA1-HR is independent from PARylation.
5. Abnormal PARP1-KLF4 axis correlates with poor breast cancer prognosis and combination of KLF4 inactivation and blockade of PARP1 leads to killing BRCA1-proficient triple negative breast cancer tumor.

8. As part of the EMBO Publications transparent editorial process initiative (see our Editorial at <http://embomolmed.embopress.org/content/2/9/329>), EMBO Molecular Medicine will publish online a Review Process File (RPF) to accompany accepted manuscripts.

a. In the event of acceptance, this file will be published in conjunction with your paper and will include the anonymous referee reports, your point-by-point response and all pertinent correspondence relating to the manuscript. Let us know if you do not agree with this.

We agree of this file publish in conjunction with the manuscript.

b. Please note that the Authors checklist will be published at the end of the RPF. I look forward to seeing a revised version of your manuscript as soon as possible.

Thanks, the author check list has been attached.

Referee #3 (Comments on Novelty/Model System for Author):

Generally the authors have used excellent model systems, but occasional switching of cell lines is still a concern.

We also have added detail on cell types and models in all figure legends and/or main manuscript text. This manuscript includes:

1. U2OS is most popular model for the mechanism studies model of DNA damage response and repair. For DNA damage response and repair studies, especially for the NHEJ and HR quantitative reporter assay, we have to switch to NHEJ reporter U2OS-EJ5-GFP and HR reporter U2OS-DR-GFP system. We also used U2OS as an MDA-MB-231 parallel cell lines in many experiments correlated to DNA damage response and repair.
2. The breast cancer cell line MDA-MB-231 is the major model cell. we use this cell lines for breast cancer cell DNA Damage, cell cycle, soft agar clone formation as well as cell viability assay; finally, we use this cell line for in vivo mouse model for therapeutic studies. In parallel of MDA-MB-231, we also include other breast cancer cell line such as BRCA1 proficient cell line MDA-MB-468, MCF7, and BRCA1 mutant cell lines HCC1937 and MDA-MB-436 cells for breast cancer cell DNA Damage, cell cycle, soft agar clone formation as well as cell viability assay.
3. Other than cancer status, to study the PARP1-KLF4 axis in normal condition *in vitro* and *in vivo* in response to DNA damage, we use the AAV-Cre induced adult mice KLF4 gut conditional knock out mice for *in vivo* studies and the KLF4 knockout MEF cells for *in vitro* studies. In *in vitro* studies, we use KLF4 knockout MEF in cytogenetic analysis and cell viability assay. KLF4 knockout MEF cells and PARP1 knockout MEFs are also used to test the function of addback of KLF4 and PARP1 wildtype and mutant genes.
4. For biochemistry studies, we use the standard cell 293T cells to complete protein-protein binding and protein modification assay.
5. DNA damage response and DNA repair are highly correlated with senescence. We use the BTR model, a temperature-induced senescence model by RasV12 in MEFs. We use this cell line to dissect PARP1-KLF4 function in senescence.

In DNA damage and DNA repair field, in addition to MEFs cells, U2OS is principally suggested and required to be used for DNA damage response study due to its advantage in genomic integrity. While cancer cells such as MDA-MB-231, MDA-MB-468, MCF7, HCC1937 MDA-MB-436 serves good model for examination of drug response, their feature of p53 mutation and other caveats make inaccuracy on DNA damage response measurement. Thus, we need to utilize various cells when we address issue of DNA damage, DNA repair as well as therapeutic drug response. To make our description of experimental design clear, we have now added details on cell types and models to make the transition smooth.

Referee #3 (Remarks for Author):

Zhou et al present a much revised manuscript addressing many of the questions raised by this and other reviewers. Not only have many experimental details been clarified but a number of additional experiments were performed to address questions raised. Overall the manuscript is still complicated and tries to deal with two different aspects of KLF4 regulation, that which is dependent upon PARylation and another which is not, both in response to DNA damage induction. It is good that the authors have now included cell cycle into their consideration, but this has also raised new concerns. It is also good that that they were able to address the issue of p53 status and indicate that the role of KLF4 in the system they are evaluating is outside the influence of p53 activity or mutation. Beyond this, there remains some minor details that are either missing or are confusing.

Fig S2G-J Indicates loss of KLF4 causes increased damage, particularly after IR and an inability to progress beyond G2 into M (increasing 4n or >4n cells (or increased aneuploidy, but without the FACs data it is not possible to evaluate how this was qualified), but not pH3 in KLF4 depleted cells). Combined with this, later in the

manuscript, is the inability to mount a p21 response in the absence of KLF4. Of note, it appears that Fig S2J is a replot of the same data as in Fig6C while Fig 6F.

For the KLF4 affect the population of 4n or >4n cells, we have published before (Please **adaptive Fig5** in *Nat Commun. 2015 Sep 30;6:8419. doi: 10.1038/ncomms9419*). By the profile of PI staining, we calculate the amount of 4n or >4n cells.

Fig 6C is a repot of Fig S2J with additional group of KLF4^{YYR/AAA}. We put the Fig S2J is in the appendix figure and keep Fig 6C in the main figure.

Adaptive from Figure 5 in *Nat Commun. 2015 Sep 30;6:8419. doi: 10.1038/ncomms9419*.

Figure removed

Fig S2K and L complements the points above, with no impact on NHEJ while HR is diminished in absence of KLF4. There are additional problems with the presentation of this figure as it is noted to be conducted with IR, which is presumably incorrect (typically this IScel based system does not involve the use of IR), but then it is not clear what the "control" and "radiation" samples are indicating.

Yes, the I-Scel based system could quantitate HR efficiency without radiation, but with radiation could boost the HR efficiency (please see **Adaptive Figure 1** in *PLoS One . 2015 Mar 31;10(3):e0122582. doi: 10.1371/journal.pone.0122582*). Here the "control" indicates without radiation. Radiation means cells were treated with 5Gy after I-Scel transfected and wait for 24h for endpoint assay. We add the explanation in the figure legend.

Adaptive from Figure 1 in *PLoS One . 2015 Mar 31;10(3):e0122582. doi: 10.1371/journal.pone.0122582*.

Figure removed

Figure S7B, it is noted that the authors did 5 Gy 4 hr treatment chase 2hr BrdU (what does this mean? - 5Gy IR, 4hr recovery, then 2hr BrdU chase?). Are the changes significant (+/-IR)? IT is also not possible from this experiment to determine whether there is a G1/S arrest or S-phase checkpoint in the presence of functional KLF4.

In figure S7B, we did 5Gy radiation, then 4hr recovery, and then chase with 2hr BrdU. We could see without KLF4, accumulation of more S phase cells in both Sham and radiation treatment in compare to KLF4^{+/+} MEFs, which indicates KLF4 contribute to G1/S checkpoint. Addback wildtype KLF4 but not mutant KLF4 (KLF4^{YYR/AAA}) limited S phase cells. In response to radiation, we could see decrease of S phase cells in both KLF4^{+/+}, KLF4^{-/-}, KLF4^{-/-}/KLF4^{WT} and KLF4^{-/-}/KLF4^{YYR/AAA} MEFs. However, S phase of KLF4^{+/+} and KLF4^{-/-}/KLF4^{WT} MEFs are much less than KLF4^{-/-}/KLF4^{YYR/AAA} and KLF4^{-/-}, which implicated that KLF4 is involved in radiation induced G1/S arrest.

It is also still not clear how, with loss of KLF4 activation of p21, and cells presumably (according to the interpretation of the authors) entering S-phase despite DNA damage, that HR is then decreased and NHEJ is unaffected. Even if there is a lack of BRCA1 induced HR, there should be increased replication stress, ATR pathway activation and either death, s-phase arrest or an alternate compensatory DSB repair induced. This aspect of the manuscript still appears under developed and poorly explained. But I also believe that it was this incongruity, lack of cell cycle control and progress into S-phase in the face of damage while a lack of HR is exactly the reasons the authors attempt to cover the two mechanisms they eventually go on to address. Clarifying the reasoning and making this bridge is essential to making this study easier to understand in totality.

As shown in figure S7B, we agree that loss of KLF4 activation of p21, and cells presumably entering S-phase despite DNA damage. We agree that in the model of MEFs, their potential loss of KLF4 leads to be increased replication stress, ATR pathway activation and either death, s-phase arrest or an alternate compensatory DSB repair induced. However, during this manuscript revision, Cintia Checa-Rodríguez *et al.* (*DNA Repair (Amst)*. 2020 Oct;94:102902) reported that in U2OS cells, knockdown KLF4 has no obvious effect on cell cycle (please see adapted figure S1 of *DNA Repair (Amst)*. 2020 Oct;94:102902), which is different from KLF4 knockout MEFs, but significantly decrease HR repair efficiency (please see adapted figure 1 of *DNA Repair (Amst)*. 2020 Oct;94:102902), which indicated that KLF4 regulated Cell cycle checkpoint is not the main contributor for HR repair regulation. Interesting, this paper found the methylation of KLF4 by PRMT5 is critical for HR repair. Actually, we reported that PRMT5 methylation is dominant KLF4 stabilization, which in consist indicated that KLF4 protein expression highly correlated with HR repair. Due to the YYR/AAA mutant, which disrupt KLF4 transcriptional function, could fully recovery HR repair, we conclude that KLF4 regulate BRCA1 expression to control HR pair is the major pathway. We agree that we need further studies to dissect the more precise mechanism of KLF4 in cell cycle, S phase function as well as HR repair in different cell lines and different models.

Adaptive from Figure 1 from *DNA Repair (Amst)*. 2020 Oct;94:102902. doi:10.1016/j.dnarep.2020.102902.

Figure removed

Adaptive from Figure S1 *DNA Repair (Amst)*. 2020 Oct;94:102902. doi:10.1016/j.dnarep.2020.102902.

Figure removed

Throughout the manuscript there is still a lot of switching of cell models (MEFs, HEK293, MDA231, U2OS), each used to look a different aspect of KLF4 biology and little continuity. There are some points where the biology observed in one cell line is recapitulated in another, which is good.

Overall, the manuscript is significantly improved but there remain some reasonably significant minor concerns.

We used the MEFs for the study of the PARP1-KLF4 axis in normal condition *in vitro* including in cytogenetic analysis and cell viability assay. KLF4 knockout MEF cells and PARP1 knockout MEFs are also used to test the function of addback of KLF4 and PARP1 wildtype and mutant genes. The function of KLF4 in MEFs have been repeated in U2OS cell lines.

The HEK293 cells have utilized as model cell for were used for overexpression, immunoprecipitation and protein-protein binding motif mapping in all fields including DNA damage and DNA repair study. In this project, while we used the HEK293 as initial protein binding analysis, all results obtained from the HEK293 cells were eventually repeated in U2OS and MDA231 cells.

The U2OS is most popular model for the mechanism studies model of DNA damage response and repair due to its quality of genomic integrity. We also used U2OS as an MDA-MB-231 parallel cell lines in many experiments correlated to DNA damage response and repair.

The MDA-MB-231 is the major BRCA1 proficient breast cancer cell model. We used this cell lines for breast cancer cell DNA Damage, cell cycle, soft agar clone formation as well as cell viability assay. In parallel of MDA-MB-231, we also include other breast cancer cell line such as BRCA1 proficient cell line MDA-MB-468, MCF7, and BRCA1 mutant cell lines HCC1937 and MDA-MB-436 cells for breast cancer cell DNA Damage, cell cycle, soft agar clone formation as well as cell viability assay. One of goals for this work is to assess the impact of PARP1-KLF4 axis in PARP inhibitor-based cancer treatment. The above cells are necessary for the drug response experiment.

19th Oct 2020

Dear Yong,

Please find enclosed the final reports on your manuscript. We are pleased to inform you that your manuscript is accepted for publication and is now being sent to our publisher to be included in the next available issue of EMBO Molecular Medicine.

We would like to remind you that as part of the EMBO Publications transparent editorial process initiative, EMBO Molecular Medicine will publish a Review Process File online to accompany accepted manuscripts. If you do NOT want the file to be published or would like to exclude figures, please immediately inform the editorial office via e-mail.

Please read below for additional IMPORTANT information regarding your article, its publication and the production process.

Congratulations on your interesting work,
Jingyi

Jingyi Hou
Editor
EMBO Molecular Medicine

Follow us on Twitter @EmboMolMed
Sign up for eTOCs at embopress.org/alertsfeeds

***** Reviewer's comments *****

*** ** IMPORTANT INFORMATION *** **

SPEED OF PUBLICATION

The journal aims for rapid publication of papers, using the advance online publication "Early View" to expedite the process: A properly copy-edited and formatted version will be published as "Early View" after the proofs have been corrected. Please help the Editors and publisher avoid delays by providing e-mail address(es), telephone and fax numbers at which author(s) can be contacted.

Should you be planning a Press Release on your article, please get in contact with embomolmed@wiley.com as early as possible, in order to coordinate publication and release dates.

LICENSE AND PAYMENT:

All articles published in EMBO Molecular Medicine are fully open access: immediately and freely

available to read, download and share.

EMBO Molecular Medicine charges an article processing charge (APC) to cover the publication costs. You, as the corresponding author for this manuscript, should have already received a quote with the article processing fee separately. Please let us know in case this quote has not been received.

Once your article is at Wiley for editorial production you will receive an email from Wiley's Author Services system, which will ask you to log in and will present you with the publication license form for completion. Within the same system the publication fee can be paid by credit card, an invoice, pro forma invoice or purchase order can be requested.

Payment of the publication charge and the signed Open Access Agreement form must be received before the article can be published online.

PROOFS

You will receive the proofs by e-mail approximately 2 weeks after all relevant files have been sent to our Production Office. Please return them within 48 hours and if there should be any problems, please contact the production office at embopressproduction@wiley.com.

Please inform us if there is likely to be any difficulty in reaching you at the above address at that time. Failure to meet our deadlines may result in a delay of publication.

All further communications concerning your paper proofs should quote reference number EMM-2020-12391-V3 and be directed to the production office at embopressproduction@wiley.com.

Thank you,

Jingyi Hou
Editor
EMBO Molecular Medicine

Corresponding Author Name: Yong Wan
 Journal Submitted to: EMBO Molecular Medicine
 Manuscript Number: EMM-2020-12391